# Olfactory responses of *Drosophila* are encoded in the organization of projection neurons

Kiri Choi*, Won Kyu Kim*, Changbong Hyeon*

School of Computational Sciences, Korea Institute for Advanced Study, Seoul, Republic of Korea

**Abstract** The projection neurons (PNs), reconstructed from electron microscope (EM) images of the *Drosophila* olfactory system, offer a detailed view of neuronal anatomy, providing glimpses into information flow in the brain. About 150 uPNs constituting 58 glomeruli in the antennal lobe (AL) are bundled together in the axonal extension, routing the olfactory signal received at AL to mushroom body (MB) calyx and lateral horn (LH). Here we quantify the neuronal organization in terms of the inter-PN distances and examine its relationship with the odor types sensed by *Drosophila*. The homotypic uPNs that constitute glomeruli are tightly bundled and stereotyped in position throughout the neuropils, even though the glomerular PN organization in AL is no longer sustained in the higher brain center. Instead, odor-type dependent clusters consisting of multiple homotypes innervate the MB calyx and LH. Pheromone-encoding and hygro/thermo-sensing homotypes are spatially segregated in MB calyx, whereas two distinct clusters of food-related homotypes are found in LH in addition to the segregation of pheromone-encoding and hygro/thermo-sensing homotypes. We find that there are statistically significant associations between the spatial organization among a group of homotypic uPNs and certain stereotyped olfactory responses. Additionally, the signals from some of the tightly bundled homotypes converge to a specific group of lateral horn neurons (LHNs), which indicates that homotype (or odor type) specific integration of signals occurs at the synaptic interface between PNs and LHNs. Our findings suggest that before neural computation in the inner brain, some of the olfactory information are already encoded in the spatial organization of uPNs, illuminating that a certain degree of labeled-line strategy is at work in the *Drosophila* olfactory system.

## Editor's evaluation

Choi et al. explore how olfactory information flows across the three major neuropils in the *Drosophila* brain – the antennal lobe (AL), mushroom Body (MB), and the lateral horn (LH). They use the two connectomes of adult *Drosophila* and 'inter-PN distances' to do this. Using this neuroanatomy based approach, they find support for a labeled-line strategy, which they subsequently test for with synaptic connectivity data for a subset of PNs. They find that while some labelled lines may exist, PNs generally participate in multi-channel integration at the MB and LH. This manuscript will be of interest to neuroscientists interested in olfactory processing and to those working on connectomic-level circuit analysis.

## Introduction

Anatomical details of neurons obtained based on a full connectome of the *Drosophila* hemisphere reconstructed from electron microscope (EM) image datasets (**Bates et al., 2020**; **Scheffer et al., 2020**) offer the wiring diagram of the brain, shedding light on the origin of brain function. Out of the

*For correspondence:
ckiri0315@kias.re.kr (KC);
wonkyukim@kias.re.kr (WKK);
hyeoncb@kias.re.kr (CH)

**Competing interest:** The authors declare that no competing interests exist.

**Figure 1.** A schematic of the *Drosophila* olfactory system. uPNs comprising each glomerulus in AL collect input signals from ORNs of the same receptor type and relay the signals to MB calyx and LH. uPNs in MB calyx synapse onto KCs; and uPNs in LH synapse onto LHNs.

immense amount of data, we study the second-order neurons, known as the projection neurons (PNs) of the olfactory system. It is the PNs that bridge the olfactory receptor neurons (ORNs) in the antenna and maxillary palp to higher olfactory centers where neural computation occurs for *Drosophila* to sense and perceive the environment (*Hallem and Carlson, 2004a*). The three neuropils, namely the antennal lobe (AL), mushroom body (MB) calyx, and lateral horn (LH), are the regions that abound with an ensemble of axonal branches of PNs and synapses (*Figure 1*). PNs can be classified as uniglomerular and multiglomerular PNs based on their structure and connectivity to other PNs. The uniglomerular PNs (uPNs) in AL constitute glomeruli that collect olfactory signals from ORNs of the same receptor type (*Gao et al., 2000*; *Couto et al., 2005*). uPNs innervating MB calyx and LH relay the signals further inside the brain through synaptic junctions with the Kenyon cells (KCs) and lateral horn neurons (LHNs), respectively. Multiglomerular PNs (mPNs), on the other hand, innervate multiple glomeruli, often contributing to the inhibitory regulation of signals relayed from ORNs to third-order olfactory neurons (*Berck et al., 2016*). PNs can functionally be categorized into either excitatory (cholinergic) or inhibitory (GABAergic), where a many GABAergic PNs tend to bypass MB calyx while innervating multiple glomeruli in AL (and hence are mPNs) (*Schultzhaus et al., 2017*; *Shimizu and Stopfer, 2017*).

Since the seminal work by *Cajal, 1911*, who recognized neurons as the basic functional units of the nervous system, there have been a series of attempts at classifying neurons using different representations of neuronal morphologies and at associating the classified anatomies with their electrophysiological responses and functions (*Uylings and van Pelt, 2002*; *Scorcioni et al., 2008*; *Jefferis et al., 2007*; *Seki et al., 2010*; *Gillette and Ascoli, 2015*; *Lu et al., 2015*; *Li et al., 2017*; *Kanari et al., 2018*; *Mihaljević et al., 2018*; *Gouwens et al., 2019*; *Laturnus et al., 2020*). Systematic and principled analyses of neuronal anatomy would be a prerequisite for unveiling a notable link between the PN organization and olfactory representations. Several different metrics involving spatial projection

patterns (*Jefferis et al., 2007*), electrophysiological properties (*Seki et al., 2010*; *Gouwens et al., 2019*), topological characteristics (e.g. morphometrics) (*Uylings and van Pelt, 2002*; *Scorcioni et al., 2008*; *Lu et al., 2015*; *Mihaljević et al., 2018*; *Gouwens et al., 2019*), intersection profiles (*Gouwens et al., 2019*), and NBLAST scores (*Jeanne et al., 2018*; *Zheng et al., 2018*; *Bates et al., 2020*; *Scheffer et al., 2020*) have been utilized in the past. More recently, machine learning approaches have been popularized as a tool for classification tasks (*Vasques et al., 2016*; *Buccino et al., 2018*; *Mihaljević et al., 2018*; *Zhang et al., 2021*).

Among a multitude of information that can be extracted from the neural anatomy associated with uPNs, the inter-PN organization draws our attention. To compare spatial characteristics of uPNs across each neuropil and classify them based on the odor coding information, we confine ourselves to uPNs innervating all three neuropils, most of which are cholinergic and follow the medial antennal lobe tract (mALT). Within this scope, we first calculate inter-PN distance matrices in each neuropil and study them in reference to the glomerular types (homotypes) to discuss how the inter-PN organization changes as the PNs extend from AL to MB calyx and from AL to LH.

In this study, we utilize two representative EM-based reconstruction datasets for the analysis (the latest FAFB *Bates et al., 2020* and the hemibrain datasets *Scheffer et al., 2020*). The FAFB dataset specifically encompasses the *Drosophila* olfactory system, while the hemibrain dataset aims for a reconstruction of the entire right hemisphere of the *Drosophila* brain. The results based on the two datasets are largely consistent and interchangeable, which generalizes our findings.

We have conducted statistical analyses to unravel potential associations between the uPN organization and the behavioral responses of *Drosophila* to external stimuli encoded by glomerular homotypes, finding that certain odor types and behavioral responses are linked to a characteristic inter-neuronal organization. The map of synaptic connectivity between uPNs and the third-order neurons (KCs and LHNs in MB calyx and LH, respectively) complements the functional implication of the association between the inter-PN organization and olfactory processing. A 'labeled-line design' in olfaction is generally considered to exhibit a chain of neurons dedicated to encoding a single olfactory feature with no direct integration with other features as the signal is passed onto higher-order neurons. While we do not demonstrate the full architecture of labeled-line design in the *Drosophila* olfactory system as the signals from odor-sensing by ORNs are passed down to the inner brain for perception, our analysis shows that homotypic uPNs encoding particular odor types not only maintain their spatially localized and bundled structure throughout all three neuropils but also display synaptic connections that converge to a narrow subset of third-order neurons. The *Drosophila* olfactory system leverages the efficiency of the labeled-line design in sensory information processing (*Min et al., 2013*; *Howard and Gottfried, 2014*; *Andersson et al., 2015*; *Galizia, 2014*).

## Results

### Spatial organization of neurons inside neuropils

#### The inter-PN distance $d_{\alpha\beta}$

First, we define a metric with which to quantify the spatial proximity between neurons. Specifically, the inter-PN distance $d_{\alpha\beta}$ represents the average taken over the minimum Euclidean distances between two uPNs $\alpha$ and $\beta$, such that $d_{\alpha\beta}$ is small when two uPNs are tightly bundled together (see *Equation 1* and *Figure 2—figure supplement 1A*). Although metrics such as the NBLAST score (*Costa et al., 2016*) and others (*Kohl et al., 2013*) can be used to study the PN organization, these metrics take both the morphological similarity and the spatial proximity into account. The distance $d_{\alpha\beta}$ only measures the pairwise distance but not the dot product term (which measures the similarity of two neuronal morphologies), whereas the NBLAST score considers both. Therefore, while the distance $d_{\alpha\beta}$ is computationally comparable to the NBLAST score, it only measures the spatial proximity between two neurons. We notice that the features of the uPN organization captured by the NBLAST distance are not necessarily aligned with $d_{\alpha\beta}$ (see *Figure 2—figure supplement 1B*). The two distances are correlated but with significant dispersion, indicating that these two metrics are not the same. Since we are solely interested in the spatial proximity (or co-location) between two uPN innervations but not the morphological similarity between them (which the NBLAST score accounts for, a point also noted by *Zheng et al., 2018*), we deliberately chose the metric $d_{\alpha\beta}$ instead of the NBLAST score for our analyses.

The distances $d_{\alpha\beta}$ (*Equation 1*) between all the possible pairs ($\alpha$ and $\beta$) of 135 uPNs from the FAFB dataset are visualized in the form of a matrix (*Figure 2*). We perform hierarchical clustering on the distance matrix for uPNs in each neuropil (see the outcomes of $d_{\alpha\beta}$-based clustering analysis in *Figure 2—figure supplement 2* and Materials and methods for the details). Individual clusters from the hierarchical clustering of uPNs in MB calyx and LH from the FAFB dataset are visualized in *Figure 3* and *Figure 4* with the colors denoting the odor types encoded by the individual uPNs, which will be discussed in detail later.

## Spatial proximity-based clustering results

In MB calyx, the hierarchical clustering divides the uPNs from the FAFB dataset into 10 clusters (*Figure 3*). Clusters $C_2^{\mathrm{MB}}$ and $C_{10}^{\mathrm{MB}}$ largely encompass the dorsal region and clusters $C_6^{\mathrm{MB}}$ and $C_7^{\mathrm{MB}}$ encompass the ventral region of the neuropil. The cluster $C_7^{\mathrm{MB}}$ shows a characteristic biforked pattern projecting to the lateral and medial regions. The cluster $C_3^{\mathrm{MB}}$ also exhibits the same structural pattern but is composed of a tight bundle of uPNs that are part of DL2d and DL2v (both of which encodes food-related odors). The cluster $C_8^{\mathrm{MB}}$ is located between the biforked innervation pattern of clusters $C_6^{\mathrm{MB}}$ and $C_7^{\mathrm{MB}}$, and predominantly innervates the posterior region. Lastly, clusters $C_1^{\mathrm{MB}}$, $C_4^{\mathrm{MB}}$, and $C_5^{\mathrm{MB}}$, innervate the anterior region of MB calyx, spatially separated from other uPNs.

In LH, 11 clusters are identified in the FAFB dataset (*Figure 4*). The cluster $C_3^{\mathrm{LH}}$ is the largest, which mainly innervates the dorsal posterior region of LH. Clusters $C_4^{\mathrm{LH}}$, $C_5^{\mathrm{LH}}$, $C_6^{\mathrm{LH}}$, and $C_9^{\mathrm{LH}}$ display variable biforked projection patterns along the coronal plane, enveloping the boundary of the cluster $C_3^{\mathrm{LH}}$. This creates a spatial pattern where a large blob of uPNs ($C_I^{\mathrm{LH}}$) are surrounded by a claw-like structure ($C_O^{\mathrm{LH}}$) (*Figure 4*, inset). Clusters $C_1^{\mathrm{LH}}$, $C_2^{\mathrm{LH}}$, and $C_7^{\mathrm{LH}}$ innervate the anterior-ventral region and display clear segregation from the other uPNs. Another group composed of clusters $C_{10}^{\mathrm{LH}}$ and $C_{11}^{\mathrm{LH}}$ innervates the posterior-ventral-medial region.

We use Pearson's $\chi^2$-test (see Materials and methods for the details) to assess the likelihood of dependence between the $d_{\alpha\beta}$-based clustering outputs for MB calyx, LH, and the glomerular labels (homotypes) statistically significant correlations are found in terms of both the p-value and the Cramér's $V$ (see *Appendix 1—table 1* and Methods for a detailed explanation of the meaning behind the p-value and the Cramér's $V$), the latter of which is analogous to the correlation coefficient for the $\chi^2$-test. The mutual information between the same set of nominal variables, which is calculated to verify our $\chi^2$-tests (see Materials and methods), offers a similar conclusion (see Appendix 1 and *Appendix 1—table 3*).

We also categorize the spatial organization of uPNs in reference to the glomerular labels. The homotypic uPNs constituting a tightly bundled glomerulus in AL manifest themselves as the block diagonal squares in the $d_{\alpha\beta}$-matrix (*Figure 2*). This is apparent in the dendrogram constructed from the distance matrix for the uPNs at AL (*Figure 2—figure supplement 3*), where uPNs sharing the same glomerular label are grouped under a common branch, thereby demonstrating the spatial proximity between uPNs forming the same glomerulus. The $d_{\alpha\beta}$-matrix indicates that such organizations are also preserved in MB calyx and LH. However, clear differences are found in the off-diagonal part of $d_{\alpha\beta}$ matrices (*Figure 2*).

The same hierarchical clustering analysis performed on the hemibrain dataset results in 14 clusters for uPN innervation in MB calyx and 13 clusters in LH. Despite the differences in the number of clusters, we find that spatial and structural characteristics of individual clusters observed from the FAFB dataset are well translated and comparable to those from the hemibrain dataset (see the clustering result in *Figure 8—figure supplement 1*). Furthermore, various statistical tests used in this study (e.g. Pearson's $\chi^2$-test) on the hemibrain dataset lead to the same conclusion (see Appendix 1, *Appendix 1—table 2*, and *Appendix 1—table 4*).

## The degrees of bundling, packing, and overlapping

To conduct a quantitative and concise analysis of $d_{\alpha\beta}$ matrices, we define the mean intra- and inter-homotypic uPN distances, $\bar{d}_{\mathrm{intra},X}$ and $\bar{d}_{\mathrm{inter},X}$ (see Methods for detailed formulation). The $\bar{d}_{\mathrm{intra},X}$ is the average distance between uPNs in the same homotype and measures the degree of uPNs in the homotype $X$ being bundled. Therefore, a smaller $\bar{d}_{\mathrm{intra},X}$ signifies a tightly bundled structure of $X$-th homotypic uPNs (see *Figure 5—figure supplement 1* for raw $\bar{d}_{\mathrm{intra},X}$ values). Similarly, $\bar{d}_{\mathrm{inter},X}$, which measures the degree of packing (or segregation), is defined as the average distance between

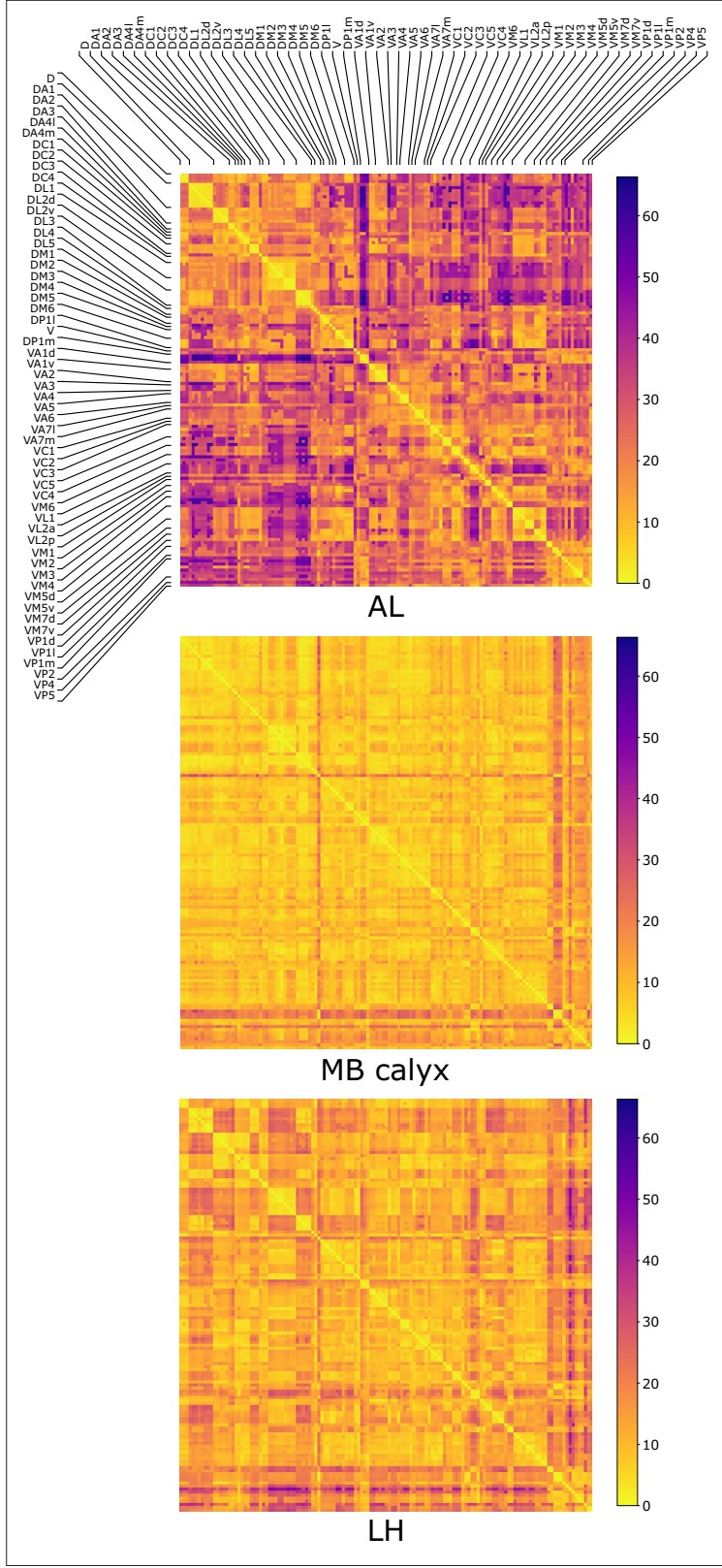

**Figure 2.** The three matrices representing the pairwise distances $d_{\alpha\beta}$ in units of $\mu$m between individual uPN in AL, MB calyx, and LH. The matrices are calculated based on uPNs available in the FAFB dataset. The diagonal blocks reflect the homotypic uPNs comprising the 57 glomerular homotypes defined in the FAFB dataset (*Bates et al., 2020*), labeled at the edges.

*Figure 2 continued on next page*

*Figure 2 continued*

The online version of this article includes the following figure supplement(s) for figure 2:

**Figure supplement 1.** An illustration of inter-PN distance calculation and comparison with the NBLAST distance.

**Figure supplement 2.** Two 135×135 matrices representing the inter-neuronal distances in (**A**) MB calyx and (**B**) LH that are reorganized based on the clustering results.

**Figure supplement 3.** The dendrograms of $d_{\alpha\beta}$-based clustering on uPNs innervation in (**A**) AL, (**B**) MB calyx, and (**C**) LH.

---

the neurons comprising the $X$-th homotype and neurons comprising other homotypes. Thus, a small value of $\bar{d}_{\mathrm{inter},X}$ signifies tight packing of heterotypic uPNs around $X$-th homotype, while a large value indicates that the homotypic uPNs comprising the homotype $X$ are well segregated from other homotypes (see *Figure 5—figure supplement 1* for raw $\bar{d}_{\mathrm{inter},X}$ values).

The degrees of bundling averaged over all homotypes ($\bar{d}_{\mathrm{intra}} = N_X^{-1} \sum_X^{N_X} \bar{d}_{\mathrm{intra},X} \approx 4\ \mu\mathrm{m}$) are comparable over all three neuropils (blue dots in *Figure 5A and B*). On the other hand, from $\bar{d}_{\mathrm{inter}}$, which is defined as the mean inter-homotype distance averaged over all $X$ s, we find that homotypic uPNs are well segregated from others in AL as expected, whereas spatial segregation among homotypes is only weakly present in MB calyx (orange dots in *Figure 5A and B* and the cartoon of *Figure 5C*). We also observe that the $\bar{d}_{\mathrm{intra}}$ and $\bar{d}_{\mathrm{inter}}$ are comparable for the two different datasets. A minor difference is observed in $\bar{d}_{\mathrm{intra}}$, indicating a slightly tighter bundling structure for the hemibrain dataset.

Next, we take the ratio of mean intra- to inter-PN distances of $X$-th homotype as $\lambda_X$ to quantify the degree of overlapping around $X$-th homotype (see Materials and methods). The term 'overlapping' is specifically chosen to describe the situation where different homotypes are occupying the same space. A large value of $\lambda_X$ (particularly $\lambda_X > 0.4$) suggests that the space occupied by the uPNs of the $X$-th homotype is shared with the uPNs belonging to other homotypes. The value $\lambda(= N_X^{-1} \sum_X^{N_X} \lambda_X)$ averaged over all the homotypes (red in *Figure 5A and B*) suggests that the extent of overlapping between uPNs is maximal in MB calyx and minimal in AL (*Figure 5C*).

*Figure 6A and B*, *Figure 7*, and *Figure 7—figure supplement 1* show individual values of $\lambda_X$ for all homotypes in the three neuropils. We identify the following features: (i) In AL, $\lambda_X \leq 0.4$ for all homotypes except DL5 (a homotype encoding aversive odors), indicating that homotypic uPNs are tightly bundled and segregated from uPNs in other glomeruli. The same trend is observable in the hemibrain dataset (*Figure 6B*), but with $\lambda_{\mathrm{DL5}} \leq 0.4$.; (ii) In MB calyx, a large portion ($\approx 65\%$) of $\lambda_X$'s exceed 0.4 and even the cases with $\lambda_X > 1$ are found (VC5, DL5), implying that there is a substantial amount of overlap between different homotypes. In the hemibrain dataset, $\approx 76\%$ of $\lambda_X$'s exceed 0.4.; (iii) Although not as significant as those in AL, many of uPNs projecting to LH are again bundled and segregated in comparison to those in MB calyx (see *Figure 7B*). (iv) The scatter plot of $\lambda_X$ between MB calyx and LH (*Figure 7C*) indicates that there exists a moderate positive correlation ($r = 0.642, p < 0.0001$) between $\lambda_X$ at MB calyx and LH. This implies that a higher degree of overlapping in MB calyx carries over to the uPN organization in LH. The result from the hemibrain dataset is similar ($r = 0.677$, $p < 0.0001$, see *Figure 7—figure supplement 1*).

The entire neuron morphologies of uPNs from two homotypes with a small ($X = \mathrm{DL3}$, which largely responds to pheromones) and a large ($X = \mathrm{DL5}$) $\lambda_X$ s in LH are visualized along with the other uPNs (gray) (*Figure 6C*). The homotype DL3, which seldom overlaps with others in AL ($\lambda_{\mathrm{DL3}} \approx 0.07$) and LH ($\lambda_{\mathrm{DL3}} \approx 0.17$), displays an increased overlapping in MB calyx ($\lambda_{\mathrm{DL3}} \approx 0.31$). Therefore, DL3 is tightly packed in AL and LH, whereas it is relatively dispersed in MB calyx. Meanwhile, the homotype DL5 displays a significant dispersion in all three neuropils, although the dispersion is the smallest in AL ($\lambda_{\mathrm{DL5}} \approx 0.74$) compared to that in MB calyx ($\lambda_{\mathrm{DL5}} \approx 1.1$) and LH ($\lambda_{\mathrm{DL5}} \approx 1.5$).

There are minor variations between the FAFB and the hemibrain datasets in terms of $\bar{d}_{\mathrm{intra},X}$, $\bar{d}_{\mathrm{inter},X}$, and $\lambda_X$, and they likely arise from the factors such as a minor mismatch in the glomerulus label annotations that sometimes affects the number of uPNs constituting a given homotype, and the difference in the number of uPNs between two datasets as a result of our selection criterion. Regardless, still present in both datasets are the spatial and organizational trends described above. Taken together, the organization of olfactory uPNs varies greatly in the three neuropils. The clear homotype-to-homotype segregation in AL no longer holds in MB calyx. Instead, the $d_{\alpha\beta}$-based clustering suggests the presence of clusters made of multiple different homotypic uPNs (*Figure 5C*). For some homotypes, the

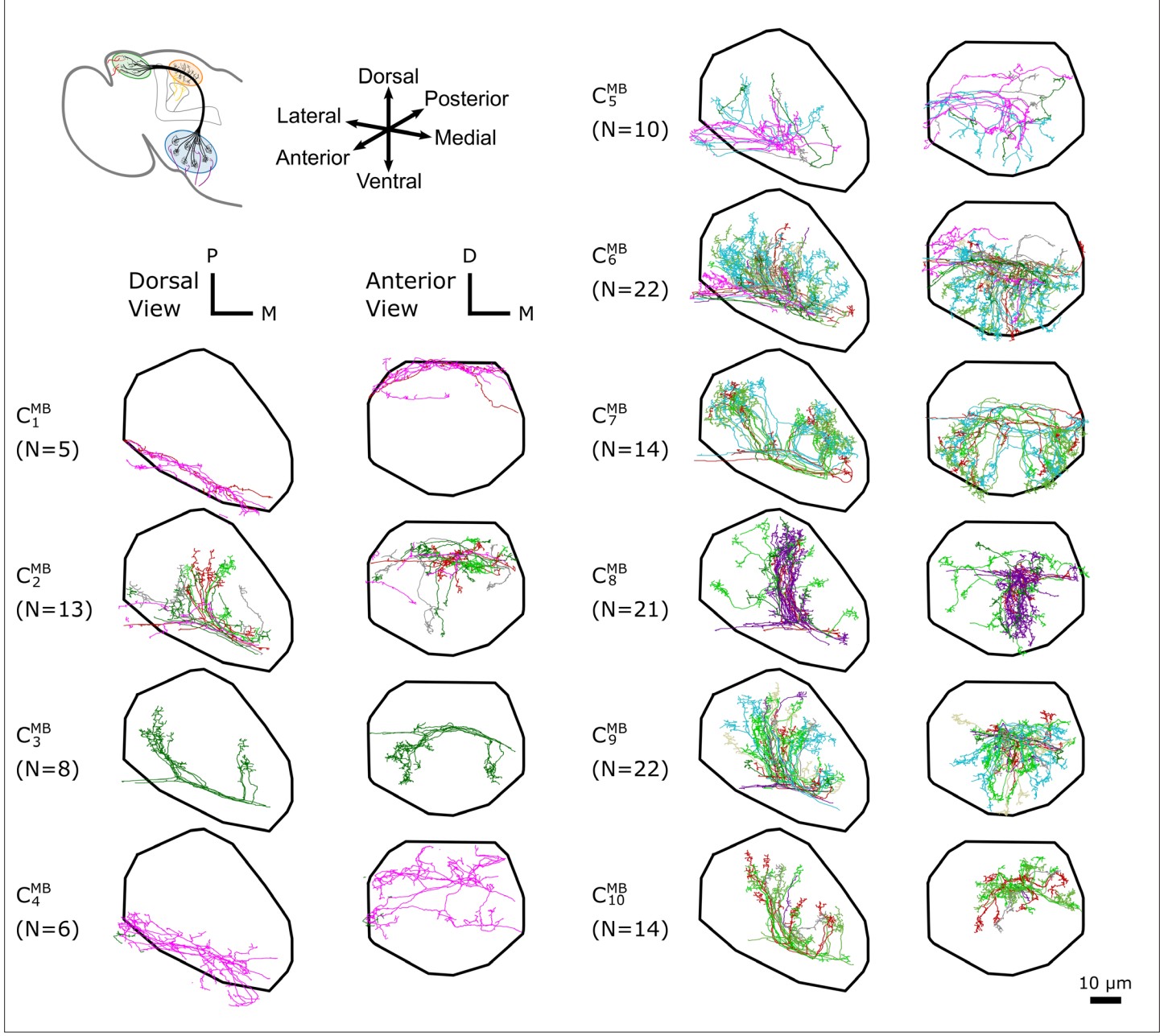

**Figure 3.** The $d_{\alpha\beta}$-based clustering on uPNs based on the FAFB dataset in MB calyx resulting in 10 clusters. The individual uPNs are color-coded based on the encoded odor types (Dark green: decaying fruit, lime: yeasty, green: fruity, gray: unknown/mixed, cyan: alcoholic fermentation, red: general bad/unclassified aversive, beige: plant matter, brown: animal matter, purple: pheromones, pink: hygro/thermo) (*Mansourian and Stensmyr, 2015*; *Bates et al., 2020*). The first and second columns illustrate the dorsal and the anterior view, respectively (D: dorsal, M: medial, P: posterior). The black line denotes the approximate boundary of MB calyx.

well-segregated organizations in AL are recovered when they reach LH (compare *Figure 7A* and *Figure 7B*).

## Relationship between neuronal organization and olfactory features

Now we explore how the structural features identified from our clustering outputs are associated with odor types and valences (behavioral responses). As briefly mentioned earlier, the color codes in *Figure 3*, *Figure 4*, *Figure 6*, and *Figure 7* depict odor types encoded by corresponding homotypic uPNs, which follow the same categorical convention used by *Mansourian and Stensmyr, 2015* and

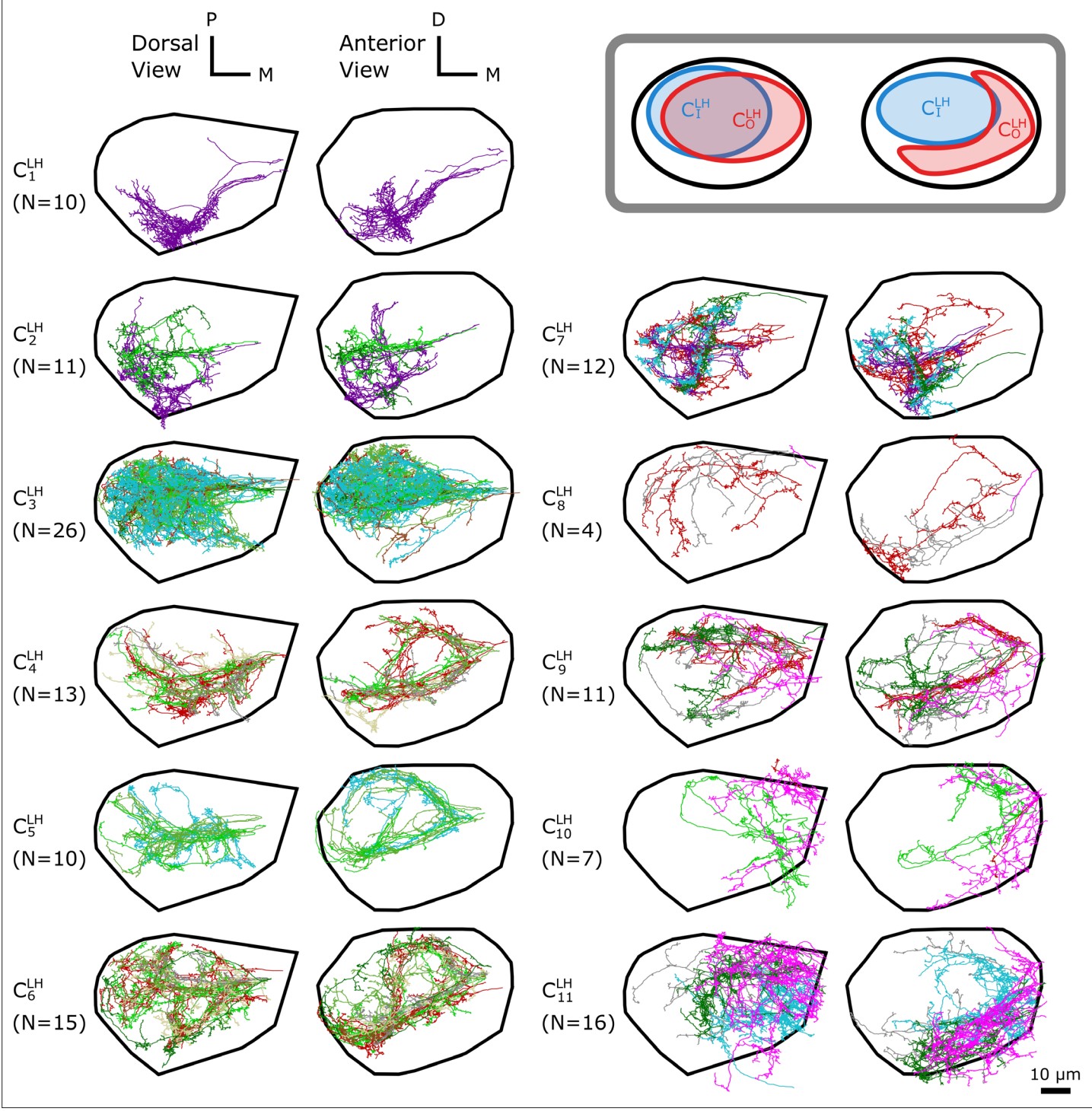

**Figure 4.** The $d_{\alpha\beta}$-based clustering on uPNs based on the FAFB dataset in LH resulting in 11 clusters. (inset) A cartoon illustrating the relative position between clusters $C_I^{LH} = C_3^{LH}$ and $C_O^{LH} = C_{4,5,6,9}^{LH}$. The individual uPNs are color-coded based on the encoded odor types (Dark green: decaying fruit, lime: yeasty, green: fruity, gray: unknown/mixed, cyan: alcoholic fermentation, red: general bad/unclassified aversive, beige: plant matter, brown: animal matter, purple: pheromones, pink: hygro/thermo). The first and second columns illustrate the dorsal and the anterior view, respectively (D: dorsal, M: medial, P: posterior). The black line denotes the approximate boundary of LH.

The online version of this article includes the following figure supplement(s) for figure 4:

**Figure supplement 1.** 15 clusters from the $d_{\alpha\beta}$-based clustering on the entire PN innervation in LH including those that do not innervate all three neuropils such as GABAergic mPNs.

*Figure 4 continued on next page*

*Figure 4 continued*

**Figure supplement 2.** Comparison of innervation pattern of PNs in LH between the uPNs innervating all three neuropils (gray, most of which follow mALT) and those that bypass MB calyx (black, most of which follow mlALT) for the FAFB dataset.

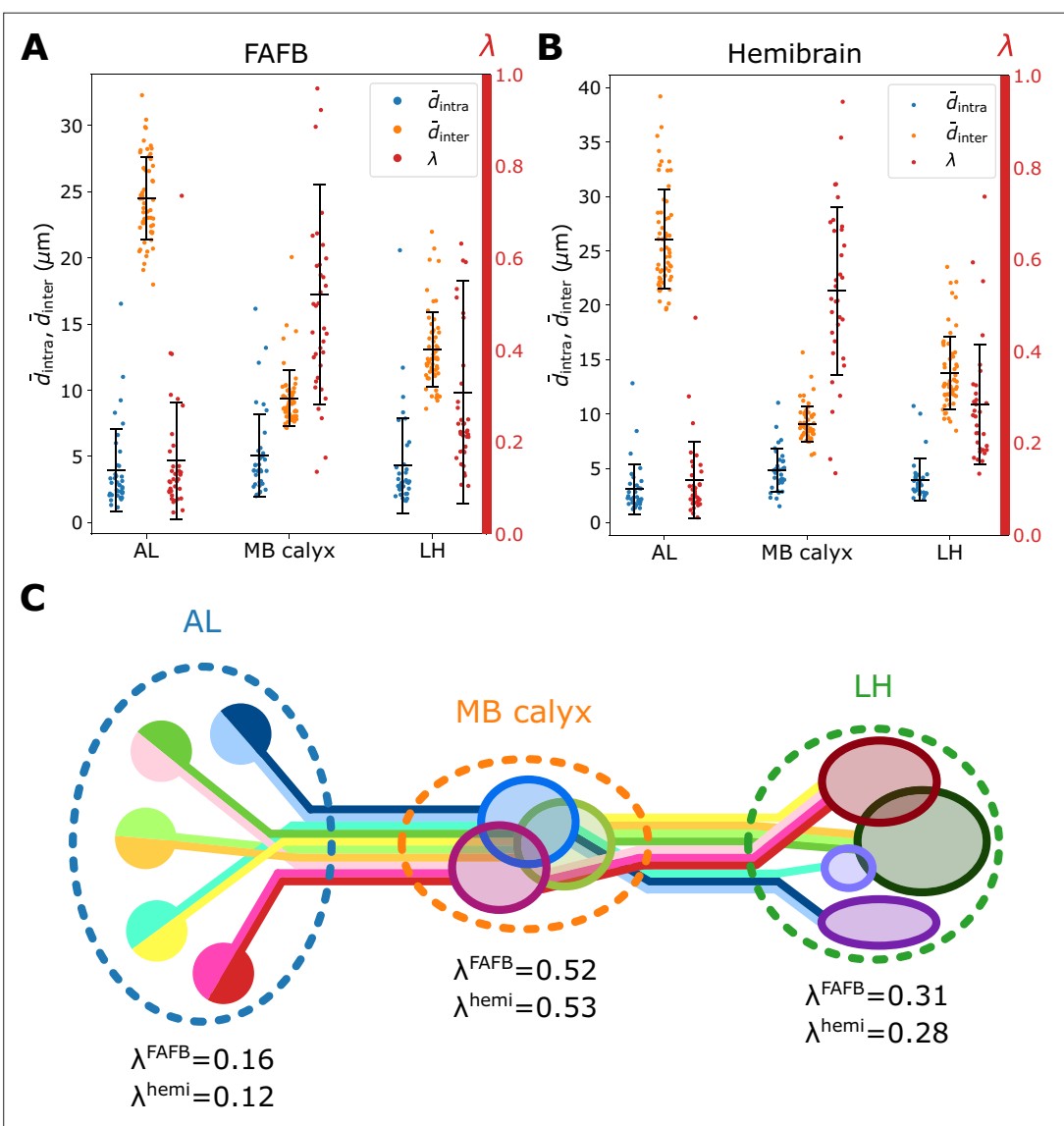

**Figure 5.** Organization of homotypic uPNs in the three neuropils. Plots of $\bar{d}_{\text{intra}}$ (blue, degree of bundling), $\bar{d}_{\text{inter}}$ (orange, degree of packing), and the ratio between the two distances $\lambda$ (red, degree of overlapping) calculated based on (**A**) the FAFB dataset and (**B**) the hemibrain dataset. Error bars depict the standard deviation. (**C**) Diagram illustrating the overall organization of uPNs at each neuropil. Homotypic uPNs are tightly bundled and segregated in AL. Several groups of homotypic uPNs form distinct heterotypic spatial clusters at higher olfactory centers, extensively overlapping in MB calyx (see *Figure 3*).

The online version of this article includes the following figure supplement(s) for figure 5:

**Figure supplement 1.** Comparison of the intra- ($\bar{d}_{\text{intra},X}$) and inter-PN ($\bar{d}_{\text{inter},X}$) distances of $X$-th homotype.

**Figure supplement 2.** A plot depicting $\bar{d}_{\text{intra}}$ (blue, degree of bundling), $\bar{d}_{\text{inter}}$ (orange, degree of packing), and the ratio between the two distances $\lambda$ (red, degree of overlapping) of 15 homotypes without (left) and with (right) 27 additional uPNs added to the FAFB dataset, which are mostly GABAergic and follow mlALT.

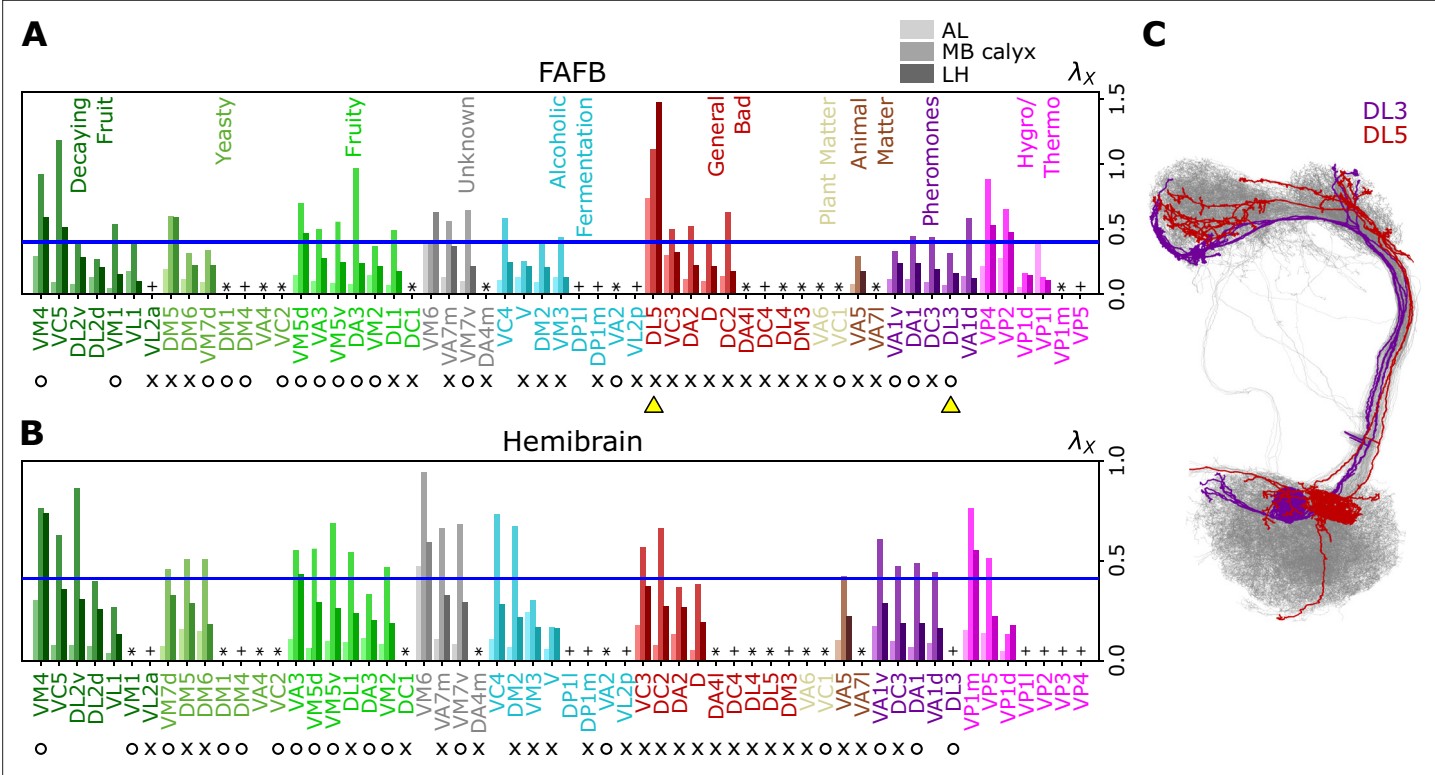

**Figure 6.** Degree of overlapping between inter-homotypic uPNs, $\lambda_X$ ($X = \text{VM4, VC5}, \ldots, \text{VP5}$). The degree of overlapping ($\lambda_X$) for $X$-th homotype in AL, MB calyx, and LH (from lighter to darker colors) calculated from the uPNs based on (**A**) the FAFB dataset and (**B**) the hemibrain dataset. The homotype label is color-coded based on the odor types associated with the glomerulus obtained from the literature and is sorted based on the value of $\lambda_X$ for each odor type at LH. Asterisks (*) mark homotypes composed of a single uPN while plus (+) mark homotypes composed of a single uPN under our selection criterion but are actually a multi-uPN homotype, whose intra-homotype uPN distance is not available. O (attractive) and X (aversive) indicate the putative valence information collected from the literature. The blue horizontal line denotes $\lambda_X = 0.4$. (**C**) Two homotypes taken from the FAFB dataset, DL3 (purple) and DL5 (red), which are indicated by yellow triangles in (**A**), are highlighted along with other uPNs (gray).

The online version of this article includes the following figure supplement(s) for figure 6:

**Figure supplement 1.** Selected morphologies of uPN innervation at each neuropil for single uPN homotypes and multiple uPN homotypes.

*Bates et al., 2020*. The O and X represent the putative valence, which indicates whether *Drosophila* is attracted to or repelled from the activation of specific homotypic uPNs. For example, DA2 responds to geosmin, a chemical generated from harmful bacteria and mold, which evokes a strong repulsion in *Drosophila* (*Stensmyr et al., 2012*). Similarly, VM3 is suggested to encode repulsive odors, while VM2 and VM7d encode attractive odors (*Mansourian and Stensmyr, 2015*; *Bates et al., 2020*). Overall, the following information is acquired from the literature (*Hallem et al., 2004b*; *Galizia and Sachse, 2010*; *Mansourian and Stensmyr, 2015*; *Badel et al., 2016*; *Bates et al., 2020*) and labeled accordingly:

- DA1, DA3, DL3, DM1, DM4, VA1v, VA2, VA3, VC1, VC2, VM1, VM2, VM4, VM5d, VM5v, VM7d, and VM7v (17 homotypes) encode attractive (O) odor.
- D, DA2, DA4l, DA4m, DC1, DC2, DC3, DC4, DL1, DL4, DL5, DM2, DM3, DM5, DM6, DP1m, V, VA5, VA6, VA7l, VA7m, VC3, VL2a, VL2p, and VM3 (25 homotypes) encode aversive (X) odor.
- The remaining homotypes are characterized as either unknown, non-preferential, or conflicting valence information.

Collecting the glomerular types of tightly bundled homotypic uPNs with $\lambda_X < 0.4$ in LH (*Figure 6*, *Figure 7*, and *Figure 7—figure supplement 1*), we explore the presence of any organizational trend.

1. In LH, out of 37 homotypes composed of multiple uPNs ($2 \leq n \leq 8$) based on our selection criterion, 29 homotypes (DL2v, DL2d, VM1, VL1, DM6, VM7d, VA3, VM5v, DA3, VM2, DL1, VA7m, VC3, VM7v, VC4, V, DM2, VM3, DA2, D, DC2, VA5, VA1v, DA1, DC3, DL3, VA1d, VP1d,

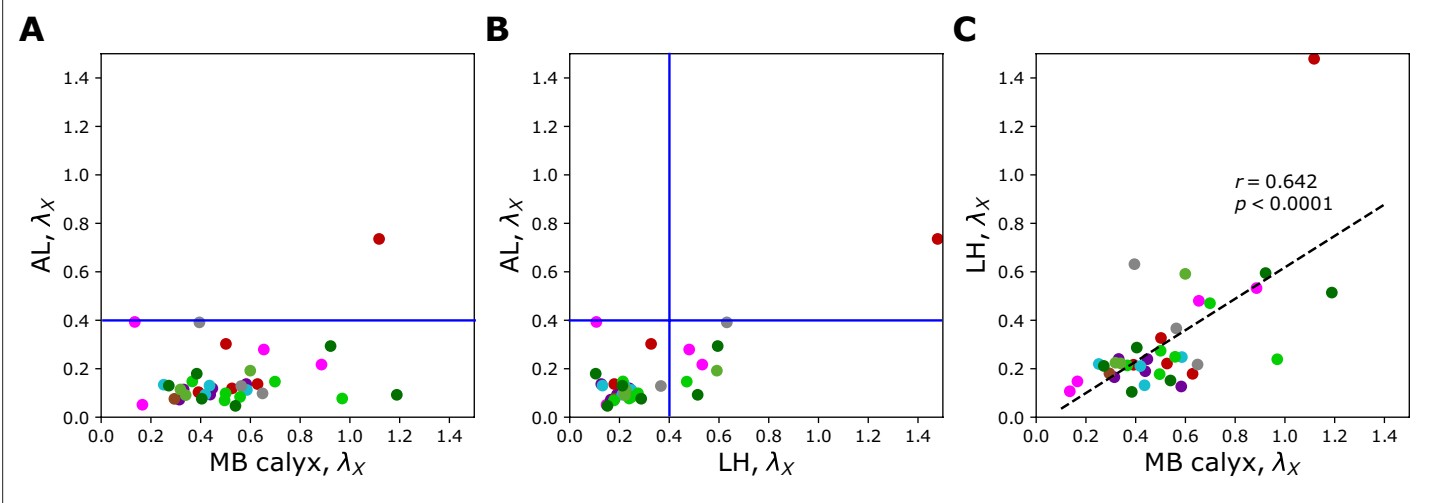

**Figure 7.** Scatter plots depicting the relationships between $\lambda_X$ s at two different neuropils calculated from the uPNs based on the FAFB dataset; (**A**) AL versus MB calyx, (**B**) AL versus LH, and (**C**) LH versus MB calyx. The color code is the same as in **Figure 6**. The blue lines in (**A**) and (**B**) denote $\lambda_X = 0.4$.

The online version of this article includes the following figure supplement(s) for figure 7:

**Figure supplement 1.** The same graph as **Figure 7** based on the hemibrain dataset.

and VP1l) satisfy the condition of $\lambda_X < 0.4$. In the hemibrain dataset, a couple of homotypes (VM1, DL5, DL3, and VP1l) are suggested to be single uPN homotypes based on our selection criterion.

2. Homotypes VA1v, DA1, DC3, DL3, and VA1d (colored purple in **Figure 3**, **Figure 4**, **Figure 6**, and **Figure 7**) encode pheromones involved with reproduction (**Grabe et al., 2016**; **Bates et al., 2020**; **Dweck et al., 2015**), and VM4, VM1, VM7d, DM1, DM4, VC2, VM5d, VA3, VM5v, DA3, and VM2 encode odors presumed to be associated with identifying attractive food sources (**Couto et al., 2005**; **Semmelhack and Wang, 2009**; **Mohamed et al., 2019**; **Bates et al., 2020**) (see **Figure 6A**). A previous work (**Grosjean et al., 2011**) has identified a group of glomeruli that co-process food stimuli and pheromones via olfactory receptor gene knock-in coupled with behavioral studies. The list of homotypes mentioned above is largely consistent with those glomeruli reported by **Grosjean et al., 2011**.

3. Homotypes DM6, DM2, VM3, VL2p, DA2, and D are likely associated with aversive food odors. DA2 responds to bacterial growth/spoilage; VL2p, DM2, and VM3 to the alcoholic fermentation process; DM6 and D to flowers (**Galizia and Sachse, 2010**; **Bates et al., 2020**).

4. Many homotypes responding to odors which can be described as kairomones, a type of odors emitted by other organisms (**Kohl et al., 2015**), are part of the 29 homotypes with $\lambda_X < 0.4$. This includes the pheromone encoding groups (VA1v, DA1, DC3, DL3, and VA1d) and others such as DA2, VC3, and VA5, which respond to geosmin, 1-hexanol, and 2-methyl phenol, respectively (**Hallem et al., 2004b**; **Galizia and Sachse, 2010**).

**Figure 8** recapitulates the cluster information from $d_{\alpha\beta}$ -based analysis along with homotypes, odor types (color-codes), and putative valence (attractive (O) and aversive (X) odors). Some points are worth making:

1. Even though uPNs innervating MB calyx exhibit large $\lambda_X$ s, the hierarchical clustering grouped homotypic uPNs together. This suggests the homotypic uPNs are still proximal in MB calyx, indicating the reduction in $d_{\text{inter}}$ is what is driving the increase in overlapping. This is already shown through $\bar{d}_{\text{intra}}$ in **Figure 5A, B** and is supported by our statistical tests (see **Appendix 1—table 1** and **Appendix 1—table 3**). The same is true for LH. The grouping of homotypic uPNs is also observable from the hemibrain dataset (**Figure 8—figure supplement 1**).

2. In the FAFB dataset, 13 out of 57 homotypes are made of a single uPN ($n = 1$, the asterisked glomeruli in **Figure 6A** and **Figure 8**), which tend to be characterized by comparatively dense branched structures (see **Figure 6—figure supplement 1**), suggestive of homotypic uPN number dependence for the neuron morphology. Among the 13 homotypes, 7 encode aversive stimuli (X), 4 encode attractive stimuli (O), and 2 have no known valence information (see **Appendix 1—table 5**). In the hemibrain dataset, 7 encode aversive stimuli (X), 5 encode

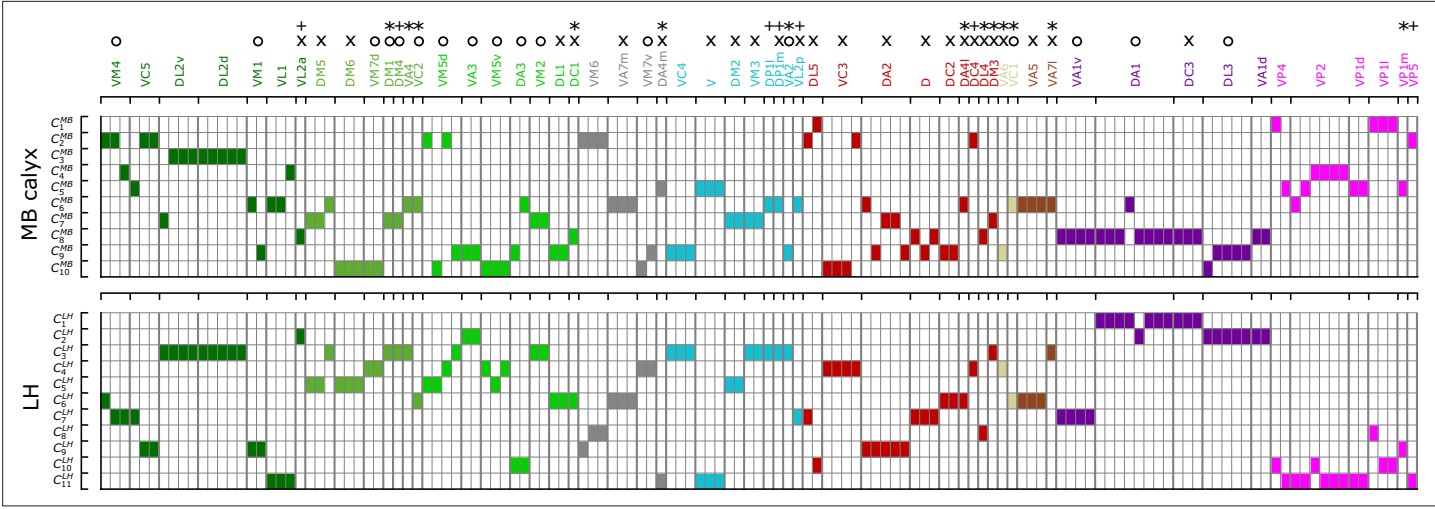

**Figure 8.** A diagram summarizing how the clusters of uPNs in MB calyx (10 clusters) and LH (11 clusters) from the FAFB dataset are associated with the odor types (Dark green: decaying fruit, olive: yeasty, green: fruity, cyan: alcoholic fermentation, red: general bad/unclassified aversive, beige: plant matter, brown: animal matter, purple: pheromones, gray: unknown, pink: hygro/thermo). Asterisks (*) mark homotypes composed of a single uPN while plus (+) mark homotypes composed of a single uPN under our selection criterion but are actually a multi-uPN homotype, whose intra-homotype uPN distance is not available. O and X represent the putative valence information collected from the literature (O: attractive, X: aversive).

The online version of this article includes the following figure supplement(s) for figure 8:

**Figure supplement 1.** The same graph as *Figure 8* based on the hemibrain dataset.

attractive stimuli (O), and 1 has no known valence information (see *Appendix 1—table 6*). The relative prevalence of single-uPN homotypes encoding aversive stimuli is noteworthy.

3. In LH, the cluster $C_1^{\text{LH}}$, located in the anterior-ventral region of the neuropil, is composed only of pheromone-encoding homotypic uPNs, DA1 and DC3. The cluster $C_2^{\text{LH}}$ is also mostly composed of pheromone-encoding homotypic uPNs, DL3 and VA1d (*Figure 4* and *Figure 8*), which is consistent with the results by *Jefferis et al., 2007*. In MB calyx, the majority of the uPNs encoding pheromones, except DL3, are grouped into the cluster $C_8^{\text{MB}}$ (see *Figure 3* and *Figure 8*). A similar trend is observed in the hemibrain dataset, although the arbitrary cluster labels differ (see clusters $C_4^{\text{LH}}$, $C_8^{\text{LH}}$, and $C_{10}^{\text{MB}}$ in *Figure 8—figure supplement 1*).

4. Hygro/thermo-sensing homotypes such as VP2 and VP4 are spatially segregated from other odor-encoding uPNs, which is observable through clusters composed predominantly of hygro/thermo-sensing homotypes (see *Figure 8* and *Figure 8—figure supplement 1*). In MB calyx, these neurons rarely project to anterior region and are distributed along the base of the neuropil. This is in line with previous literature (*Li et al., 2020*). In LH, they are clustered in the posterior-ventral-medial region, hardly innervating the neuropil but covering the medial side of the neuropil (*Figure 3* and *Figure 4*).

5. Along with the clusters of uPNs visualized in *Figure 3* and *Figure 4*, of particular note are the clusters formed by a combination of several homotypic uPNs. A large portion of uPNs innervating LH that encodes potentially aversive responses are grouped into clusters $C_4^{\text{LH}}$, $C_5^{\text{LH}}$, $C_6^{\text{LH}}$, and $C_9^{\text{LH}}$, which envelop the cluster $C_3^{\text{LH}}$ where mostly food-related homotypes converge (*Figure 4*). In the hemibrain dataset, these correspond to $C_{10}^{\text{LH}}$ and $C_{11}^{\text{LH}}$ for the aversive responses and $C_6^{\text{LH}}$ and $C_{13}^{\text{LH}}$ for the food-related homotypes (*Figure 8—figure supplement 1*).

Given that the synaptic communications with KCs and LHNs are critical for neural computation in the inner brain, the specific type of uPN organization in each neuropil should be of great relevance. Indeed, it has been suggested that the spatial convergence, segregation, and overlapping of different homotypic uPNs within neuropil influence the information processing in higher olfactory centers (*Grosjean et al., 2011*).

According to previous studies (*Jefferis et al., 2007*; *Liang et al., 2013*; *Kohl et al., 2013*; *Fişek and Wilson, 2014*), uPN innervation in LH and LHNs are highly stereotyped in terms of connectivity and response. Homotypic uPNs are spatially organized in AL, and to a certain degree, in LH, based on the odor type and valence information (*Min et al., 2013*; *Huoviala et al., 2020*). The presence of tightly bundled anatomy of homotypic uPNs ($\lambda_X < 0.4$) in both AL and LH (*Figure 7B* and

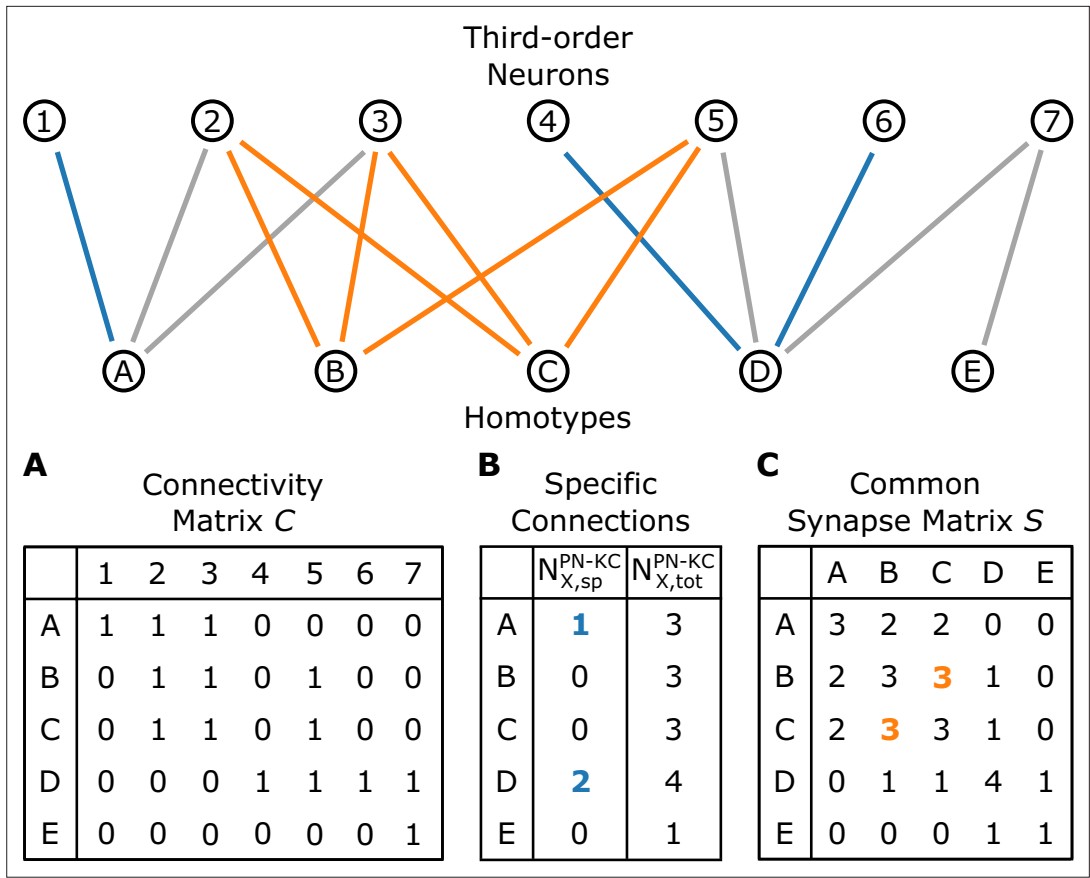

**Figure 9.** A schematic illustrating the connectivity between homotypes ($X = A, B, \ldots, E$) and third-order neurons ($i = 1, 2, \ldots, 7$). (**A**) The connectivity matrix $C$, where $\mathcal{C}_{X,i} = 1$ when any uPNs in the $X$-th homotype and $i$-th third-order neuron synapses and $\mathcal{C}_{X,i} = 0$ otherwise. (**B**) The number of $X$-th homotype-specific connections ($N_{X,\text{sp}}$) and the total number of third-order neurons synapsed to any uPNs in the $X$-th homotype. (**C**) The common synapse matrix ($\mathcal{S}$) whose element specifies the number of third-order neurons commonly connected between two homotypes. The homotype A is connected to three third-order neurons 1, 2, and 3 ($N_{A,\text{tot}} = 3$). Neuron 1 is not synapsing with any other homotype but A, and hence $N_{A,\text{sp}} = 1$; similarly, $N_{D,\text{sp}} = 2$ (the blue lines depict specific connections). The signals from the two homotypes B and C are shared by the third-order neurons 2, 3, and 5; therefore, $\mathcal{S}_{BC} = 3$ in the common synapse matrix $\mathcal{S}$.

The online version of this article includes the following figure supplement(s) for figure 9:

**Figure supplement 1.** The raw connectivity matrices.

---

*Figure 7—figure supplement 1B*) may imply that the *Drosophila* olfactory system dedicates a part of the second-order neural circuit on behalf of the 'labeled-line' design, which enables the organism to sense urgent chemical stimuli at the early stage of information processing without going through more sophisticated neural computation in the inner brain (*Howard and Gottfried, 2014*; *Andersson et al., 2015*; *Min et al., 2013*).

## Labeled-line design of the higher order olfactory neurons

The concept of labeled-line design is widely considered at work at the ORN-PN interface (AL) as the signal generated from specific olfactory receptors converges to a single glomerulus (*Vosshall et al., 2000*; *Couto et al., 2005*; *Fishilevich and Vosshall, 2005*). A potential labeled-line strategy or separated olfactory processing of aversive odors encoded by DA2 has been extensively discussed (*Stensmyr et al., 2012*; *Seki et al., 2017*; *Huoviala et al., 2020*). It has been shown that pheromone-encoding homotypes in LH (*Jefferis et al., 2007*; *Ruta et al., 2010*; *Kohl et al., 2013*; *Frechter et al., 2019*; *Bates et al., 2020*; *Das Chakraborty and Sachse, 2021*) are at work in specific third-order olfactory neurons. So far, we have shown that the labeled-line design is present in the architecture of

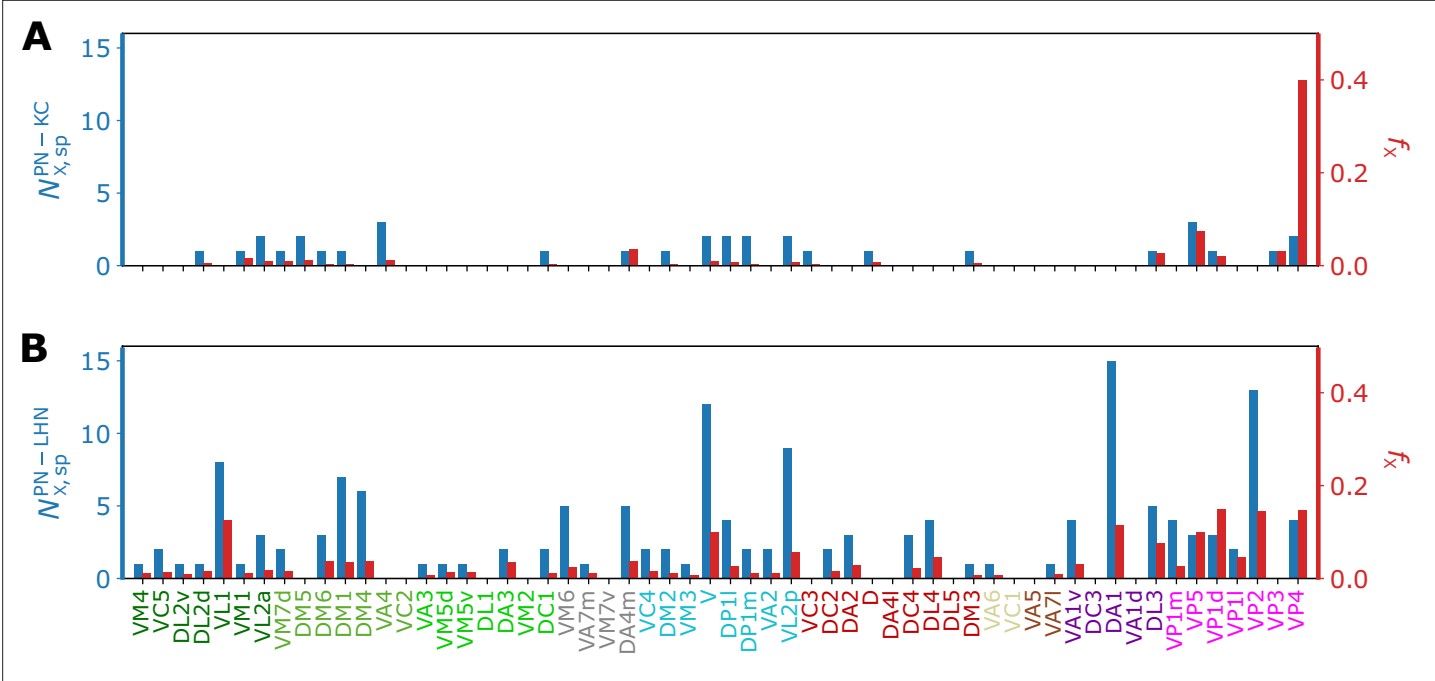

**Figure 10.** Bar graphs depicting the number of KCs/LHNs that synapse with a specific homotype $X$ ($N_{X,\mathrm{sp}}$, blue) and the percentage of KCs/LHNs that synapse with a specific homotype $X$ ($f_X = N_{X,\mathrm{sp}}/N_{X,\mathrm{tot}}$, red) at (**A**) PN-KC and (**B**) PN-LHN interfaces, with the synaptic weight of $N = 3$ used as the threshold.

The online version of this article includes the following figure supplement(s) for figure 10:

**Figure supplement 1.** Homotype specific connections with the synaptic connectivity threshold of $N = 8$.

higher olfactory centers of second-order neurons, that is, MB calyx and LH, where homotypic uPNs are tightly bundled together despite the lack of glomerular structure. In this section, we will conduct a comprehensive analysis of the synaptic connectivity between PNs and third-order olfactory neurons (KCs and LHNs) using three demonstrations. We ask (i) whether the labeled-line strategy implied in the uPN organization is translated over to the third-order olfactory neurons, (ii) to what extent the signals encoded by different homotypic uPNs are integrated at synaptic interfaces with the third-order neurons, and (iii) whether the spatial properties of pre-synaptic neurons (PNs) play any role in signal integration by the third-order neurons.

## Homotype-specific connections

For the analysis of the interface between homotypic uPNs and third-order neurons, we study the connectivity matrices $\mathcal{C}^{\mathrm{PN-KC}}$ and $\mathcal{C}^{\mathrm{PN-LHN}}$ (see *Figure 9*, *Figure 9—figure supplement 1*), which are extracted from the hemibrain dataset (*Scheffer et al., 2020*). The $\mathcal{C}^{\xi}$ ($\xi = $ PN-KC or PN-LHN) is a binary matrix ($\mathcal{C}^{\xi}_{X,i} = 0$ or $1$ dictating the connectivity) of synaptic connectivity between $X$-th homotypic uPNs and $i$-th third-order neuron (KC or LHN). It is observed that most of the KCs and LHNs *integrate* information from multiple homotypes, but that there are also a small number of KCs and LHNs that synapse only with a single homotype (*Figure 10*).

The 'homotype-specific' connections, defined as the number of third-order neurons that only synapses with a specific homotype but not with the others (see *Figure 10* and Methods for more information) can be quantified in terms of the total number of third-order neurons in contact with $X$-th homotypic uPNs, and it can be obtained by counting the non-zero elements of the matrix $\mathcal{C}$ with fixed $X$. For the case of the PN-KC interface, this number can be obtained from $N^{\mathrm{PN-KC}}_{X,\mathrm{tot}} = \sum_{i=1}^{1754} \mathcal{C}^{\mathrm{PN-KC}}_{X,i}$. Specifically, *Figure 10A* shows $N^{\mathrm{PN-KC}}_{X,\mathrm{sp}}$ and those normalized by $N^{\mathrm{PN-KC}}_{X,\mathrm{tot}}$ ($f_X = N^{\mathrm{PN-KC}}_{X,\mathrm{sp}}/N^{\mathrm{PN-KC}}_{X,\mathrm{tot}}$, see Materials and methods for the detailed algorithms behind the calculation), for all homotypes ($X = $ VM4, VC5, ..., VP4). Compared to those in KCs, the 'homotype-specific' connections are much more prevalent in LHNs (*Figure 10*). Certain homotypic uPNs, in particular, the

hygro/thermo-sensing homotypes are connected to the LHNs which are dedicated to process the signals from hygro/thermo-sensing homotypes ($\geq 10\%$ of PN-LHN connections made by homotypes).

To address the concern with potential false positives in the detected synapses, we reexamine our results based on the synaptic connectivity with a higher threshold ($N = 8$). *Figure 10—figure supplement 1* demonstrates that the homotype-specific connections tend to increase under a more stringent synapse selection criterion, especially in LH. This is most notable in homotypes DM1, DM4, DP1l, and VM6. The existence of these 'homotype-specific' third-order neurons suggests that a subset of olfactory processing may rely on the labeled-line strategy that extends beyond the layer of second-order neurons to the higher brain center.

## Third-order neurons mediate signal integration

*Figure 11A, B* show the 'common synapse matrices' representing the number of commonly connected third-order neurons between two homotypes $X$ and $Y$ ($\mathcal{S}_{XY}^{\eta}$ with $\eta$ = KC or LHN), which provide glimpses into the extent of signal integration mediated by KCs and LHNs (see *Figure 9C* and the caption for how these matrices are constructed from the connectivity matrix).

1. Overall, the number of synaptic connections between uPNs and KCs is greater than that between uPNs and LHNs ($\mathcal{S}_{XY}^{\text{KC}} > \mathcal{S}_{XY}^{\text{LHN}}$, see *Figure 11*).
2. In MB calyx, the signals from food-related odors-encoding homotypes (e.g. Yeasty, Fruity, or Alcoholic Fermentation odor types) are shared by a large number of KCs, which constitute a few large clusters in $\mathcal{S}^{\text{KC}}$ matrix, depicted in red ($\mathcal{S}_{XY}^{\text{KC}} \gtrsim 35$) and indicated by the blue arrows on the top in *Figure 11A*. Some KCs process signals almost exclusively from the hygro/thermo-sensing homotypes without sharing any signal from other homotypes ($\mathcal{S}_{XY}^{\text{KC}} = 0$ for the cases of $X$ and $Y$ homotype pairs without any signal integration, which are depicted in black in *Figure 11*). There are also homotypes with significantly less number of overall synaptic connections to KCs, dictated by the diagonal element of the matrix $\mathcal{S}^{\text{KC}}$ (see *Figure 11—figure supplement 2A*). In comparison with $\mathcal{S}^{\text{LHN}}$, the $\mathcal{S}^{\text{KC}}$ suggests a stronger but less organized signal integration between heterotypic uPNs by KCs and lends support to the previous literature pointing to the random synapsing of KCs with uPNs at MB calyx (*Caron et al., 2013*; *Stevens, 2015*; *Eichler et al., 2017*; *Zheng et al., 2020*).
3. $\mathcal{S}^{\text{LHN}}$, on the other hand, demonstrates LHN-mediated signal integration localized to subsets of homotypes. When we collect LHNs connected to a particular homotype and check which other homotypes these LHNs are also synapsing (thereby analyzing the scope of signal integration happening at LH), we find a strong tendency of signals from pheromone and hygro/thermo-sensing uPNs to be integrated within the given odor/signal type (*Figure 11*). The fact that the pheromone-encoding and hygro/thermo-sensing homotypes share the synaptic connections to LHNs among themselves are demonstrated as the homotype-specific block patterns along the diagonal of the $\mathcal{S}^{\text{LHN}}$ matrix (see purple and pink arrows on the side in *Figure 11B*). The $\mathcal{S}^{\text{LHN}}$ matrix also shows that signals from various food-related odor encoding homotypes, such as DP1l, DP1m, VA2, and VL2p or DM1, DM4, and VA4 are also integrated (see blue arrows in *Figure 11B*). Many of these homotypes encode signals originating from esters, which is intriguing given the ester-encoding LHN cluster shown by *Frechter et al., 2019*. The results suggest that certain odor types are processed through common channels of LHNs that are largely dedicated to encoding a particular odor type.

A more stringent selection criterion for synaptic connectivity does not affect our conclusion on the signal integration by the third-order olfactory neurons (*Figure 11—figure supplement 2*). The only notable change is the general increase in the cases with no integration ($\mathcal{S}_{XY} = 0$) in $\mathcal{S}^{\text{LHN}}$, especially for hygro/thermo-sensing homotypes. Thus, the extent of signal integration from homotypic uPNs to KCs and LHNs summarized in $\mathcal{S}^{\text{KC}}$ and $\mathcal{S}^{\text{LHN}}$ is robust.

## Spatial proximity-based versus connectivity-based clustering

Next, we study the relationship between spatial proximity-based clustering and connectivity-based clustering results. Upon visual inspection, the connectivity-based clustering at MB calyx (*Figure 12A* on the right) appears less structured than the spatial proximity-based clustering (*Figure 12A* on the left). Specifically, many homotypic uPNs are grouped under a common branch in the tree structure obtained from the spatial proximity-based clustering, whereas such a feature is largely absent in the output of the connectivity-based clustering. Therefore, the spatially well-clustered uPNs at

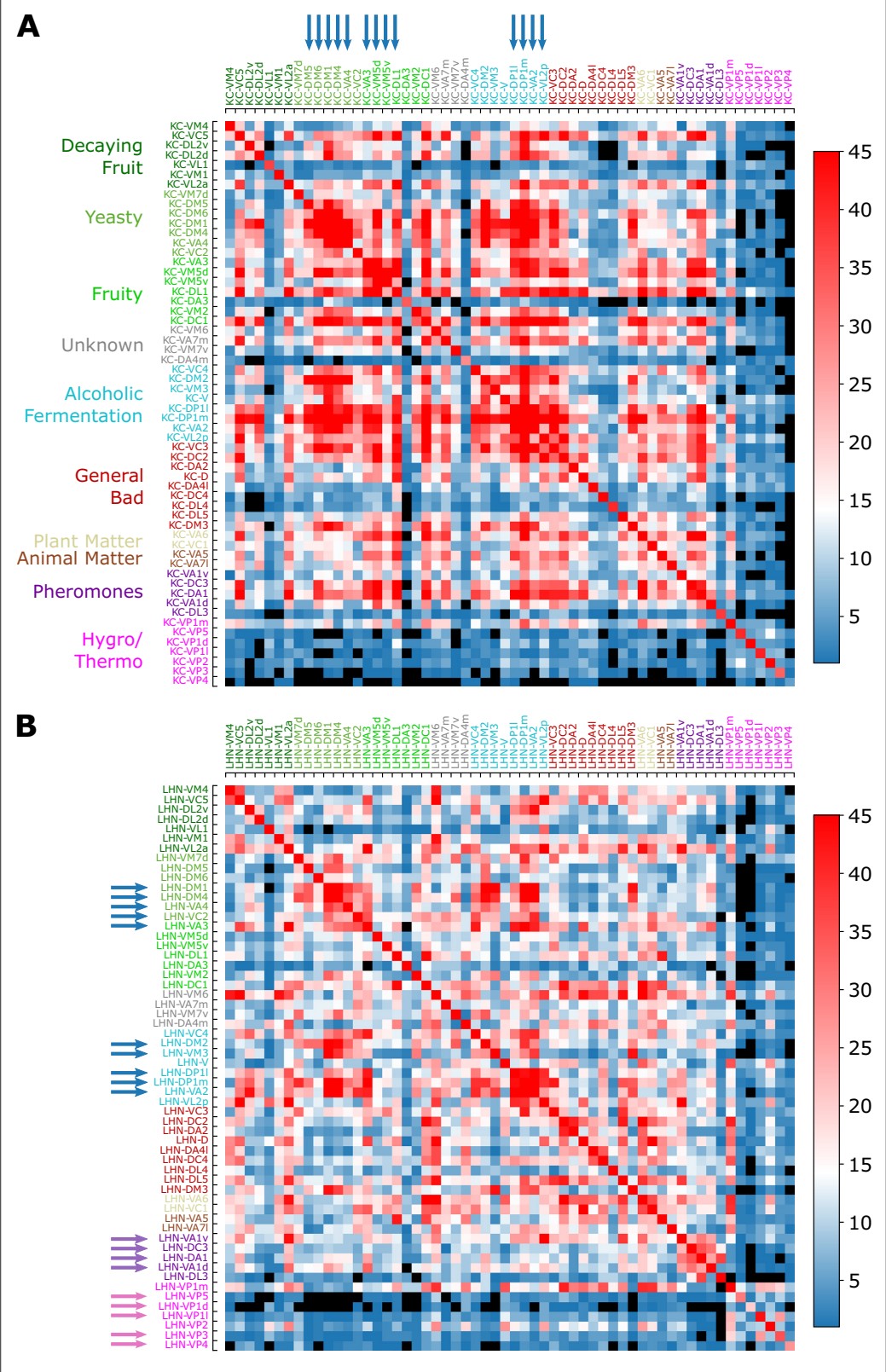

**Figure 11.** Common synapse matrices (**A**) $\mathcal{S}^{\mathrm{KC}}$ and (**B**) $\mathcal{S}^{\mathrm{LHN}}$, each of which represents the extent of signal integration from homotypic uPNs to KCs and LHNs. The color code represents $\mathcal{S}_{XY}$, which is the number of the third-order neurons (KCs or LHNs) synapsing with both homotypes $X$ and $Y$. The black color is used when there is no third-order neuron-mediated signal integration ($\mathcal{S}_{XY} = 0$) happening between two homotypes $X$ and $Y$. See

*Figure 11 continued on next page*

*Figure 11 continued*

**Figure 9C** and its caption for how the common synapse matrices are calculated from the connectivity matrices provided in **Figure 9—figure supplement 1**.

The online version of this article includes the following figure supplement(s) for figure 11:

**Figure supplement 1.** The total number of third-order neurons connected to each homotype.

**Figure supplement 2.** Common Synapse matrices (**A**) $\mathcal{S}^{\mathrm{KC}}$ and (**B**) $\mathcal{S}^{\mathrm{LHN}}$ calculated with the synaptic connectivity threshold of $N = 8$, each of which represents the extent of signal integration from homotypic uPNs to KCs and LHNs.

---

MB calyx (or *stereotyped structure*) do not necessarily translate into structured connectivity patterns (or *stereotyped connectivity*), which is consistent with the notion of randomized PN-KC connections (**Caron et al., 2013**; **Stevens, 2015**; **Eichler et al., 2017**; **Zheng et al., 2020**). In stark contrast to the outcomes for MB calyx, most homotypic uPNs are grouped in the connectivity-based clustering for LH (**Figure 12B**). This suggests that the spatially proximal uPNs synapse with a similar group of LHNs. The stereotyped organization and stereotyped connectivity of uPNs in LH have been suggested before (**Jefferis et al., 2007**; **Liang et al., 2013**; **Kohl et al., 2013**; **Fişek and Wilson, 2014**), and we demonstrate such stereotypies are, in reality, expressed throughout LH over all uPNs. In LH, spatial and organizational characteristics of uPNs are well translated to connectivity to LHNs.

A quantitative comparison of two trees based on statistical tests lends support to the notion that the spatial organization of uPNs can be indicative of connective properties, most evident in LH (see Appendix 2 for Baker's Gamma index, entanglement, and cophenetic distance correlation).

## Discussion

The inter-PN organization revealed in this study and its association with odor type/valence are reminiscent of the generally accepted notion that *the form determines the function* in biology. Previously observed stereotypes of neurons in the *Drosophila* olfactory system were largely based on the differentiation between pheromones and non-pheromones (**Ruta et al., 2010**; **Kohl et al., 2013**; **Frechter et al., 2019**; **Das Chakraborty and Sachse, 2021**), the whole-cell patch-clamp recording (**Seki et al., 2017**), and imaging studies suggestive of stimulus-dependent arrangement of neurons in LH (**Marin et al., 2002**; **Wong et al., 2002**; **Jefferis et al., 2007**). Our results are generally consistent with the previous studies, which suggest that a level of stereotypy in uPN organization in MB calyx and LH is universal throughout *Drosophila*, which can be captured through different metrics and methodologies. In line with **Lin et al., 2007**, our study finds that homotypes DL2v and DL2d constitute a bilateral cluster in MB calyx ($C_3^{\mathrm{MB}}$), and that the dual organization of uPNs is present in MB calyx and LH, such that homotypes DC2, DL1, and VA5 are sorted into the same cluster in LH while sharing similar innervation pattern in MB calyx. Our clustering results in LH share similarities with the NBLAST score-based LH clusters (**Bates et al., 2020**). The uPNs that ended up in the same cluster or nearby clusters, such as homotypes DM1, DM3, DM4, VA4, and VM3 in the cluster $C_3^{\mathrm{LH}}$, are also grouped in the NBLAST score-based clustering analysis (**Bates et al., 2020**). We find a significant correlation of $d_{\alpha\beta}$ with NBLAST score (see **Figure 2—figure supplement 1**) despite the fact that two metrics prioritize different aspects of neuronal anatomy.

Our inter-PN distances and clustering results suggest the spatial organization of uPNs differs greatly in each neuropil (**Figure 5**). Some of the tightly bundled organization of uPN homotypes are well preserved throughout the neuropils despite the lack of glomerulus in MB calyx and LH. The spatial segregation between different homotypes is, however, practically not present in MB calyx, leading to a high degree of overlapping. Therefore, the heterogeneity of homotypes at the PN-KC synaptic interface may physically assist the randomized sampling known to exist between uPNs and KCs (**Caron et al., 2013**; **Stevens, 2015**; **Eichler et al., 2017**; **Zheng et al., 2020**).

Our analysis suggests that LH is compartmentalized into four regions: (1) Posterior-dorsal region primarily occupied by food-related uPNs; (2) Anterior-ventral region occupied by pheromone-encoding uPNs; (3) Biforked bundle surrounding posterior-dorsal region largely occupied by food-related uPNs with an aversive response; (4) Posterior-ventral-medial region occupied by hygro/thermo-sensing uPNs. Previous attempts at identifying regions of odorant space in LH revealed compatible results. The three domains (LH-PM, LH-AM, and LH-AL) suggested by **Strutz et al., 2014** seem to be a

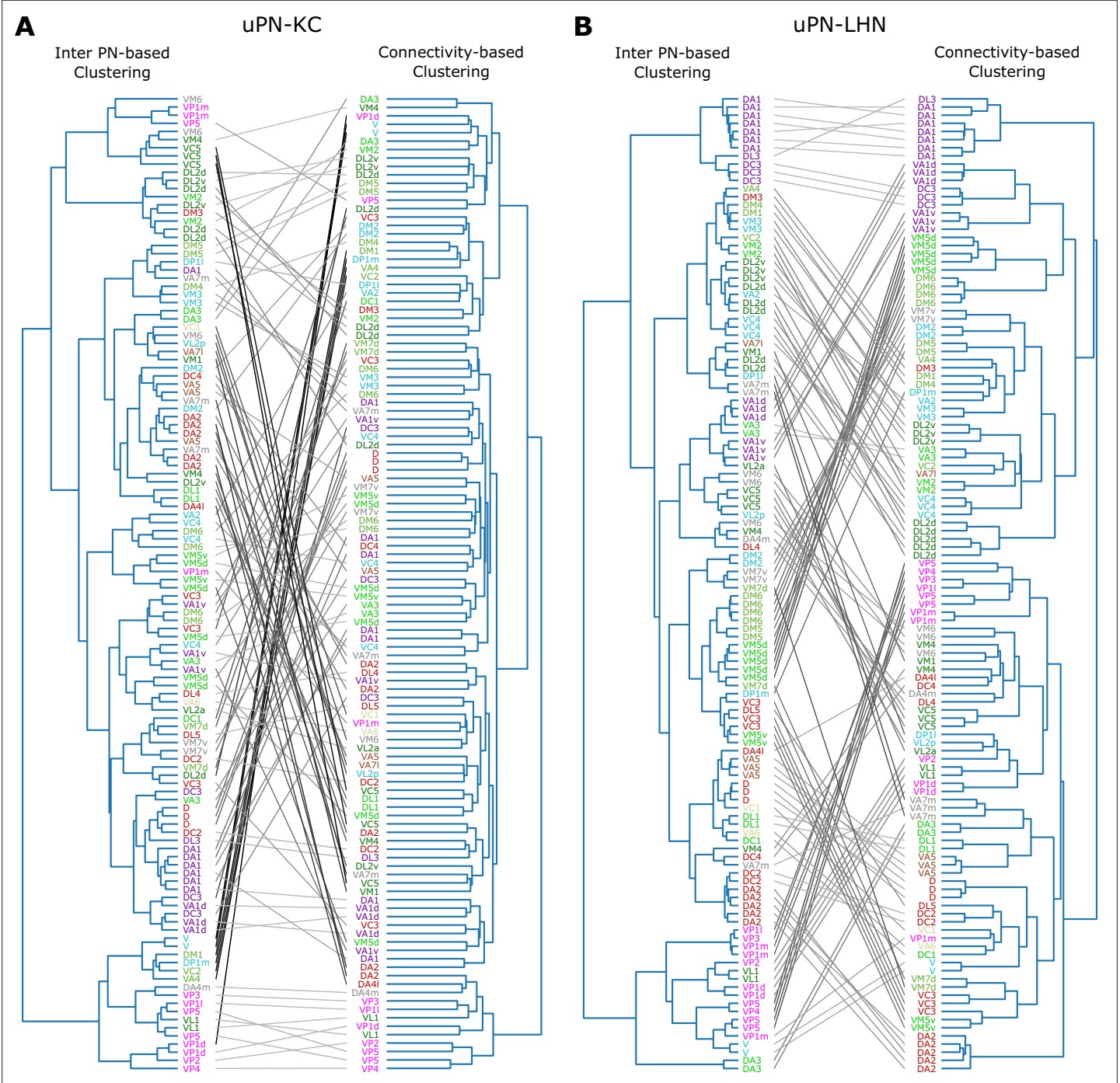

**Figure 12.** Tanglegrams comparing the tree structures generated from the inter-PN distances-based (left) and the connectivity-based clustering (right) (**A**) between uPNs and KCs, and (**B**) between uPNs and LHNs. The same uPNs in the two tree structures are connected with lines, which visualize where the uPNs clustered by one method end up in the clustering results of another. The labels for uPNs are representative of the homotype and are color-coded based on the encoded odor types (Dark green: decaying fruit, lime: yeasty, green: fruity, gray: unknown/mixed, cyan: alcoholic fermentation, red: general bad/unclassified aversive, beige: plant matter, brown: animal matter, purple: pheromones, pink: hygro/thermo).

different combination of our clustering result (LH-PM and LH-AM correspond to the posterior-dorsal region and LH-AL corresponds to a combination of anterior-ventral region and the biforked bundle). Although not perfect, the study of the axo-axonic communities in LH yields results with comparable characteristics (*Bates et al., 2020*), understandably due to the necessity of inter-neuronal proximity to form synapses. For example, the community 12 by *Bates et al., 2020* is predominantly composed

of homotypes VP1l and DL5, which resembles our cluster $C_{10}^{\mathrm{LH}}$. The community 6 contains a mixture of homotypes VA5, VC1, D, DA4l, DC2, DA3, and VA7m, which is reminiscent of our cluster $C_{6}^{\mathrm{LH}}$.

Many homotypic uPNs that are spatially localized in LH can be associated with key survival features and a strong innate response (*Seki et al., 2017*). In this sense, the stereotyped localization of pheromone-encoding uPNs in $C_{8}^{\mathrm{MB}}$, $C_{1}^{\mathrm{LH}}$, and $C_{2}^{\mathrm{LH}}$ is of great interest. Our study not only lends support to the existing studies pointing to the labeled-line strategy in the *Drosophila* olfactory system but also suggests that there may exist an even more sophisticated level of spatial organization, which supersedes the pheromone versus non-pheromone segregation. Interestingly, while the spatial organization of uPNs in LH has a basis on the functionality of the odor type encoded, it does not seem to be directly translated to segregated chemical features seen in LHNs (*Frechter et al., 2019*). The apparent divergence observed at the PN-LHN interface, coupled with strongly stereotyped connectivity may contribute to a higher resolution of odor categorization.

## The uPN organizations from FAFB and hemibrain datasets are consistent

Our analyses of both the FAFB and the hemibrain dataset (*Scheffer et al., 2020*) find that the results from both datasets are generally consistent. For example, $\bar{d}_{\mathrm{intra}}$, $\bar{d}_{\mathrm{inter}}$, and $\lambda$ analyzed based on two different datasets are almost identical (see *Figure 5A and B*). $\bar{d}_{\mathrm{intra,X}}$, $\bar{d}_{\mathrm{inter,X}}$, and $\lambda_X$ show slight differences due to a mismatch between the FAFB and the hemibrain dataset (on glomerulus labels and the number of uPNs based on our selection criterion) leading to a different number of uPNs per homotype (*Figure 6A and B* and *Figure 5—figure supplement 1*), but the correlation between $\lambda_X$ s at MB calyx and LH are still observed (*Figure 7C* and *Figure 7—figure supplement 1C*). Most importantly, the clustering results are similar, where many spatial clusters in both datasets share the same set of homotypes. Additionally, odor type-dependent spatial properties are retained (*Figure 8* and *Figure 8—figure supplement 1*), with all statistical tests supporting our hypothesis. In conclusion, the outcomes from our analyses of the two EM datasets lend support to the previous claims of stereotypy in the *Drosophila* brain and neuronal structures (*Jenett et al., 2012*; *Jeanne et al., 2018*; *Schlegel et al., 2021*).

## Odor signal processing and labeled-lines

Our study suggests that while the primary connectivity motif of third-order olfactory neurons indeed integrates signals, there still exist several labeled lines. The synaptic connections at the PN-KC interface in MB calyx are largely integrative and randomized - with an exception of hygro/thermo-sensing homotypes that display stereotypy even in terms of the connectivity to the KCs. A similar observation has been made by *Li et al., 2020*, who employed NBLAST score to identify a structural segregation between odor-encoding and hygro/thermo-sensing homotypes. They found that specific KC types are preferentially targeted by hygro/thermo-sensing homotypes. *Marin et al., 2020*, who carried out connectome analysis specific to hygro/thermo-sensing homotypes, also identified that lateral accessory calyx (lACA), the anterior-dorsal part of the calyx, are primarily targeted by hygro/thermo-sensing homotypes (analogous to our clusters $C_{1}^{\mathrm{MB}}$ and $C_{4}^{\mathrm{MB}}$ in *Figure 3*), and found that a number of KCs are dedicated to encoding signals from these homotypes. The uPNs in LH are spatially segregated, which translates to connectivity in three different levels. First, certain LHNs are dedicated to encoding signals from a specific homotype. The number of these 'homotype-specific' LHNs varies across the homotype and can make up a significant portion of PN-LHN connections depending on the homotype (*Figure 10*). Second, synaptic connectivity maps between uPNs and LHNs indicate odor type-dependent integration occurs at LH (*Figure 11B*). Channels of LHNs predominantly encoding specific odor types are observed; one primarily integrates responses from certain food-related homotypes, one integrates pheromone-encoding homotypes, and another integrates hygro/thermo-sensing homotypes. Third, homotypic uPNs share similar connectivity to LHNs, unlike those in MB calyx. The signals relayed from the spatially well-organized (or tightly bundled) homotypes are localized into a specific group of LHNs, thereby forming a 'homotype-specific' connectivity motif (*Figure 10*, *Figure 11*, and *Figure 12*).

In our study of the labeled-line strategy, we made several interesting observations, which are worth comparing with the concept of 'fovea' introduced by *Zheng et al., 2020*. A 'fovea' delineates deviations between experimentally observed connectivity matrices and connectivity under the assumption

of random synapses in MB calyx, specifically for certain food-related uPNs (*Zheng et al., 2020*). A group of common KCs predominantly sampling 'food-related' uPNs manifest themselves in the common synapse matrix $\mathcal{S}^{\text{KC}}$ (see the group of homotypes comprising the clusters, highlighted by the blue arrows in *Figure 11A*). A subset of homotypic uPNs under the food-related 'fovea' reported by *Zheng et al., 2020* are also spatially clustered (e.g. DM1, DM4, DP1m, DP1l, VA2, and VA4). While most of these homotypes are spatially proximal (the vast majority of the uPNs are located in clusters $C_6^{\text{MB}}$ and $C_7^{\text{MB}}$), some homotypes under the food-related 'fovea' such as VA2 are sampled from spatially disparate clusters. Thus, it is likely that factors other than the spatial organization of uPNs in neuropils contribute to creating the 'fovea'. Interestingly, the spatial proximity of pheromone-encoding homotypes in MB calyx may suggest the existence of pheromone-encoding 'fovea,' but most uPNs in these homotypes do not converge in connectivity-based clustering with an exception of VA1d. In fact, we suspect the spatial organization of pheromone-encoding homotypes in MB calyx, which is placed at the center of the neuropil, to facilitate the observed randomization of connections by increasing the accessibility of KCs to these homotypes. There is, however, a potential hygro/thermo 'fovea,' where homotypes such as VP1d and VP2 are spatially clustered together and the signals from these homotypes are relayed by the same set of KCs. Curiously, VL1 is part of this hygro/thermo 'fovea' (*Figure 12A*).

## Multiglomerular PNs are spatially distinctive

Apart from uPNs primarily explored in this study, a host of local neurons (LNs) and multiglomerular PNs (mPNs) also constitute sophisticated neural circuits to regulate the signals received from ORNs (*Sudhakaran et al., 2012*; *Bates et al., 2020*), playing a significant role in the olfactory signal processing (*Olsen et al., 2010*; *Jeanne and Wilson, 2015*; *Seki et al., 2017*). A large portion of these mPNs is GABAergic and inhibitory (*Berck et al., 2016*; *Tobin et al., 2017*), whereas the role of interneurons can be both inhibitory and excitatory (*Wilson et al., 2004*; *Turner et al., 2008*). Electrophysiological measurements indicate that mPNs are narrowly tuned to a specific set of odor stimuli (*Berck et al., 2016*), which is significant given that PNs are generally thought to be more broadly tuned than presynaptic ORNs (*Wilson et al., 2004*). Several PNs do not follow the typical mALT, but mediolateral antennal lobe (mlALT) or lateral antennal lobe tracts (lALT) instead, thereby bypassing innervation through one of the higher olfactory centers (*Schultzhaus et al., 2017*; *Zheng et al., 2018*; *Bates et al., 2020*). As stated previously, we confined ourselves to uPNs innervating all three neuropils to compare the spatial organization across neuropils for each uPN. As a result, 28 uPNs present in the FAFB dataset are not explored in our study. In MB calyx, only two uPNs constituting VP3 were dropped as a result of our selection criterion, which ended up in an almost identical clustering output once hierarchical clustering was performed on the entire 137 uPNs that innervate MB calyx. Two missing uPNs were grouped into clusters $C_4^{\text{MB}}$ and $C_6^{\text{MB}}$, along with other hygro/thermo-sensing homotypes. On the other hand, the addition of 27 uPNs constituting 15 homotypes innervating LH but not MB calyx created four new clusters when hierarchical clustering was performed (*Figure 4—figure supplement 1*). The additional uPNs changed the content of the individual clusters; that is, the tree-cutting algorithm broke down a few clusters that became larger due to the additional uPNs. Furthermore, when we calculate the $\bar{d}_{\text{intra}}$, $\bar{d}_{\text{inter}}$, and $\lambda$ in LH for the 15 homotypes that included the 27 uPNs, we find that the $\bar{d}_{\text{intra}}$ values increased when the 27 uPNs were included (see *Figure 5—figure supplement 2*). This suggests that the previously removed uPNs, most of which follow mlALT, are significantly different in terms of spatial and organizational characteristics and thus should be analyzed separately. Out of 27 additional uPNs in LH, 21 were in mlALT, 5 were in trans-lALT, and 1 was in mALT. *Figure 4—figure supplement 2* illustrates how these 27 uPNs innervate LH which demonstrates the reason behind increased $\bar{d}_{\text{intra}}$ values. These 27 uPNs are mostly GABAergic (21 are labeled as GABAergic, 1 as cholinergic, and 4 as unknown neurotransmitter type), covering 84% of all GABAergic uPNs available in the FAFB dataset. These uPNs innervate LH differently from other uPNs in the same homotype that follow mALT (see homotypes such as DA1, DC4, DL2d, DL2v, DP1l, VA1d, VA1v, VL2a, VL2p, and VP5 in *Figure 4—figure supplement 2*). Morphologically, inhibitory GABAergic neurons are often considered 'smooth' and aspiny (*Douglas et al., 1989*; *Bopp et al., 2014*; *Gouwens et al., 2019*), which are discernible from *Figure 4—figure supplement 2*.

## The single-uPN homotypes may have different morphological properties

It is of great interest that many of the single-uPN homotypes, characterized by densely branched morphology, encode signals with aversive responses. Direct transmission of the associated signals across the three neuropils via a single PN might simplify the overall processing of the olfactory signals as well as reduce the energetic cost. Similarly, the morphological characteristics of uPN innervation at each neuropil are intriguing. Even though a structural difference exists between the single-uPN and multi-uPN homotypes, all uPN innervations within neuropil share a similar morphology regardless of the homotype (see *Figure 6—figure supplement 1*; *Choi et al., 2022*). A localized morphological diversity within a neuron may be a characteristic aspect of pseudo-unipolar neurons like uPN and suggests a fundamentally multi-scale characteristic of neuron morphology.

The *Drosophila* brain EM reconstruction project has evolved to its near completion since the EM image dataset was first released (*Dorkenwald et al., 2022*). The reconstruction of the majority of the *Drosophila* central brain as well as the corresponding connectome with detailed information of the individual synapses has become publicly available (*Scheffer et al., 2020*). Our analysis of the second-order neurons inside the *Drosophila* olfactory system may be translated to other parts of the nervous system in *Drosophila* as well as different organisms including the central nervous system (CNS) of humans. For the mammalian olfactory system, the details of analyses must be adapted, however, since the wiring scheme is much more complex than that of an insect (*Maresh et al., 2008*). For example, multiple glomeruli encoding the same olfactory signal exist in humans (*Mombaerts et al., 1996*). When analyzing the spatial properties, this can be accounted for by prioritizing the individual glomerulus over the homotypes. Then, homotypic PNs forming different glomeruli may be compared or averaged if one were to consider the homotype-dependent characteristics. According to the neurotransmitter map from a recent study (*Dolan et al., 2019*), sophisticated processes beyond neuronal anatomy are apparently at work in the olfactory signal processing. Thus, functional studies incorporating odor response profiles in PNs (*Badel et al., 2016*) and ORNs (*Münch and Galizia, 2016*; *Bak et al., 2018*) would supplement our findings. The extension of our study to the other regulatory interneurons and mPNs, morphological studies of second-order neurons, and spatial analysis of third-order neurons will be of great interest for a better understanding of the olfactory signal processing beyond the implication of the neural anatomy and connectivity studied here.

## Materials and methods

### Data preparation

We used the neuron morphology reconstruction of 346 *Drosophila* olfactory neurons from the FAFB dataset (*Bates et al., 2020*) traced from EM images. The neurons were extracted from the right hemisphere of the female *Drosophila*. Out of 346 neurons in the FAFB dataset, 164 neurons were uPNs. One uPN in the dataset (neuron ID = 1356477 forming VP3) did not have an associated reconstruction (.swc file) available and was therefore ignored. For this study, uPNs that innervate all three neuropils were chosen because our aim is (1) to compare spatial characteristics of the uPN innervation across each neuropil and (2) to classify each uPN based on the odor encoding information. Thus, out of 164 uPNs, a total of 135 uPNs constituting 57 homotypes were collected under this criteria, resulting in mostly cholinergic uPNs that follow mALT. Rest of the uPNs that did not innervate all three neuropils are collected for the supplementary analysis. In MB calyx, a total of 137 PNs are identified with two PNs constituting VP3 that do not innervate all three neuropils. On the other hand, in LH, a total of 162 PNs are identified, indicating that 27 PNs constituting 15 homotypes do not innervate all three neuropils. The morphological information of each neuron is stored as a set of 3D coordinates with the connectivity specified with the parent nodes. Complete reconstruction of neuron morphology was made by connecting data points based on their parent-child relationship.

The hemibrain dataset (*Scheffer et al., 2020*) was taken from the neuPrint database (*Clements et al., 2020*), where we collected from the right hemisphere of the female *Drosophila* a total of 120 uPNs forming 58 glomeruli based on the same criterion we used for the FAFB dataset (uPNs that innervate all three neuropils). Unlike the FAFB dataset, the neurons in the hemibrain dataset are labeled with regions of interest (ROI), which are used to query uPNs conforming to our selection criterion. The discrepancy in the number of uPNs between the two datasets most likely resulted from the

difference between the neuropil boundary we used and the region defined by the hemibrain dataset. In fact, we find that the total number of uPNs in both datasets is comparable, with 164 uPNs in the FAFB dataset and 162 uPNs in the hemibrain dataset. The two datasets also had a minor mismatch in the glomerulus label annotations, sometimes affecting the number of uPNs constituting a given homotype. Among the 120 uPNs from the hemibrain dataset, five uPNs had ambiguity in terms of their glomerulus labels, which is presumably due to poorly formed glomerular structures. For these uPNs, we adopted the glomerulus labels of the FAFB dataset with the matching hemibrain neuron IDs.

Additionally, a recent community-led effort identified three glomeruli in both databases with conflicting glomerulus labels, which have been a source of confusion (*Schlegel et al., 2021*). After an extensive study, the community agreed to rename the glomeruli in both datasets labeled as VC3l, VC3m, and VC5 as VC3, VC5, and VM6, respectively (*Schlegel et al., 2021*). Thus, we have manually incorporated these labels into our analyses for both the FAFB and the hemibrain dataset.

Next, we systematically demarcated the regions of AL, LH, and MB calyx. The density of data points projected to each axis was used for the identification since the neuropils are featured with a much higher density of data points than the rigid backbone connecting them. The boundaries defining each neuropil were systematically chosen from local minima that separate neuropils from rigid backbones. Due to the unique structure of uPNs, sometimes the projection along a given axis cannot fully differentiate two neuropils. To resolve this issue, projections along each axis were sampled while rotating the data points along the reference axes at $5°$ increments to obtain multiple snapshots. The densities were analyzed to choose the optimal degrees of rotation along the reference axes that could best segment the neuropils. We used the smallest average and deviation value of density at the local minima as the criteria to choose the optimal rotation. The process has been repeated for each neuropil to identify a set of boundaries along multiple transformed axes with various degrees of rotations that optimally confine each neuropil. This information has been combined to create a set of conditions per neuropil for segmentation. The resulting neuropils were confirmed through visual inspection. We compared our neuropil segmentation boundaries with neuropil volume surface coordinates provided by *Ito et al., 2014* via CATMAID (*Saalfeld et al., 2009*) and found the boundaries are comparable (data not shown). An overview of the segmentation process is available in *Figure 13*.

The odor type and odor valence information were extracted from various literature (*Hallem et al., 2004b*; *Galizia and Sachse, 2010*; *Stensmyr et al., 2012*; *Mansourian and Stensmyr, 2015*; *Badel et al., 2016*; *Bates et al., 2020*) and we closely followed the categorical convention established by *Mansourian and Stensmyr, 2015* and *Bates et al., 2020*. However, we note that the categorization of a uPN under a specific odor category may overshadow the complete spectrum of odorants a uPN might encode, especially if the uPN encodes ORs that are broadly tuned. Therefore, we focused on the well-separated pheromone/non-pheromone encoding types and valence information.

To test our labeled-line hypothesis, the connectivity information between uPNs and higher olfactory neurons such as KCs and LHNs was necessary. Since only the hemibrain dataset contains detailed connectivity information, all of our connectivity studies are done using uPNs, KCs, and LHNs queried from the hemibrain dataset. We chose KCs and LHNs that made at least three synaptic connections with any of the 120 uPNs from the hemibrain dataset. This resulted in 1754 KCs and 1295 LHNs, creating bipartite connectivity matrices at each neuropil.

## Inter-PN distance

The 'distance' $d_{\alpha\beta}$ between two neurons, $\alpha$ and $\beta$, with different lengths ($N_\alpha \leq N_\beta$) is quantified by calculating.

$$d_{\alpha\beta}^2 = \frac{1}{N_\alpha} \sum_{i=1}^{N_\alpha} \min\left[(\boldsymbol{r}_i^\alpha - \boldsymbol{r}_j^\beta)^2\right], \tag{1}$$

where $\boldsymbol{r}_i^\alpha$ is an i-th coordinate forming the neuron $\alpha$. *Equation 1* is evaluated over all pairs of $\boldsymbol{r}_i^\alpha$ and $\boldsymbol{r}_j^\beta$ with $j = 1, \ldots, N_\beta$ that gives rise to the minimum value. This means that when $N_\alpha \leq N_\beta$, for every i-th coordinate in the neuron $\alpha$ ($\boldsymbol{r}_i^\alpha$), we find j-th coordinate in the neuron $\beta$ ($\boldsymbol{r}_j^\beta$) that is the closest to $\boldsymbol{r}_i^\alpha$. Then, the spatial proximity of a given pair of neurons is assessed by the $d_{\alpha\beta}$ that denotes the average of all the minimum Euclidean distances between the pair of coordinates.

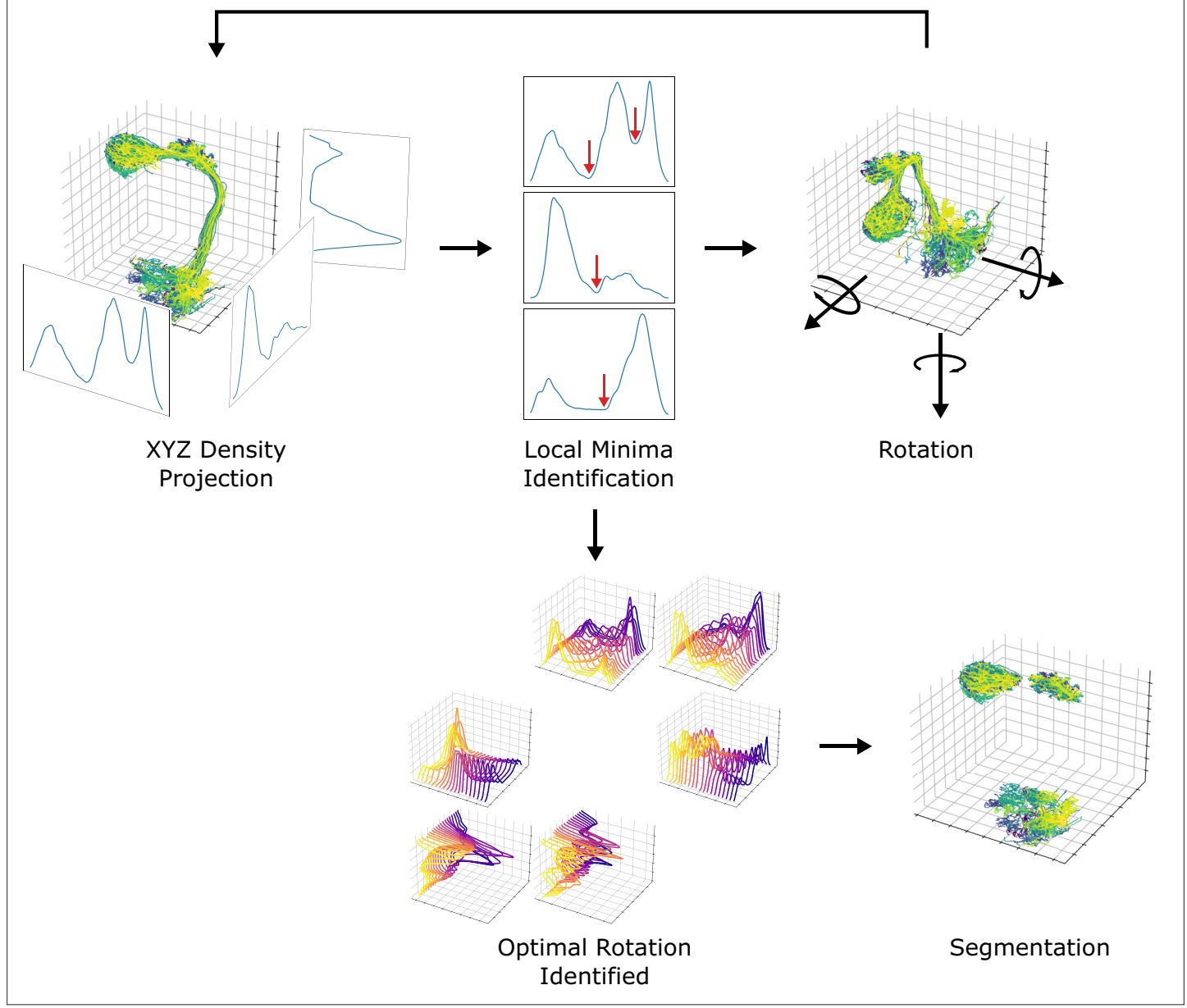

**Figure 13.** A diagram depicting the neuropil segmentation process. The data points from skeletal reconstruction are projected to each axis to generate distributions from which local minima are obtained. The process is repeated while rotating the uPNs along each axis. A collection of histograms and corresponding local minima are surveyed to generate a set of optimal rotations and boundaries for individual neuropil. The resulting parameters are combined to form a collection of conditions to segment each neuropil.

## The degree of bundling, packing, and overlapping

We define the mean intra- and inter-homotype distances as.

$$\bar{d}_{\text{intra},X} \equiv \frac{1}{N} \sum_{\alpha,\beta \in X}^{N} d_{\alpha\beta} \tag{2}$$

and

$$\bar{d}_{\text{inter},X} \equiv \frac{1}{N} \sum_{\alpha \in X, \beta \notin X}^{N} d_{\alpha\beta}, \tag{3}$$

**Table 1.** The optimal number of clusters of uPNs in the FAFB dataset determined by employing the dynamic hybrid cut tree method, elbow method, gap statistics, and maximum average silhouette coefficient.

|  | Dynamic hybrid | Elbow | Gap | Silhouette |
|---|---|---|---|---|
| AL | 19 | 14 | 8 | 54 |
| MB calyx | 10 | 11 | 7 | 2 |
| LH | 11 | 9 | 7 | 2 |

where $X$ denotes a homotype and $N$ is the total number of uPN pairs to be averaged. The $\bar{d}_{\text{intra},X}$ is calculated over all the pairs of uPNs in the $X$-th homotype, quantifying the tightness of bundling of uPNs constituting the $X$-th homotype. On the other hand, $\bar{d}_{\text{inter},X}$ is calculated over the pairs of uPNs between $\alpha$-th uPN belonging to the $X$-th homotype and $\beta$-th uPN in the $Y$-th homotype ($Y \neq X$), such that it measures the extent of packing of uPNs around the $X$-th homotype. The degree of overlapping for the $X$-th homotype, $\lambda_X$, is defined as the ratio of average intra- and inter-homotype distances,

$$\lambda_X = \frac{\bar{d}_{\text{intra},X}}{\bar{d}_{\text{inter},X}}, \tag{4}$$

which represents how clearly the $X$-th homotype is segregated from other homotypes in a given space. A large value of $\lambda_X$ ($\lambda_X \gg 1$) implies that the space spanned by the $X$-th homotype is not clearly discerned from other homotypes.

## Spatial clustering of projection neurons

Hierarchical/agglomerative clustering was used to cluster the uPN innervation at each neuropil using the pairwise $d_{\alpha\beta}$ matrices. First, the linkage was decided based on the pairwise distance matrix built with the Farthest Point Algorithm (or 'complete' method), where uses the maximum distance between neurons to define the distance between two clusters. This criterion is used to build hierarchical relations (or nested clusters) in a bottom-up approach where each neuron is treated as a cluster at the beginning. The result is a fixed tree structure of individual neurons from which the finalized clusters are formed using an optimal tree-cutting algorithm. In the dendrogram from AL (*Figure 2—figure supplement 3*), homotypic uPNs are grouped together with high accuracy, suggesting our distance metric $d_{\alpha\beta}$ is adequate. We tested various tree-cutting criteria such as elbow method, gap statistics, maximum average silhouette coefficient, and dynamic hybrid cut tree method (*Langfelder et al., 2008*) to determine the optimal number of clusters. Among them, we selected the dynamic hybrid cut tree method, since it performed the best in giving the cluster number closest to the number of different odor types (which is 10) (*Table 1*). We deployed the dynamic hybrid cut tree method with the minimum cluster size of 4 neurons for the tree-cutting, following the neuron clustering procedure used by *Gouwens et al., 2019*.

## Pearson's $x^2$-test of independence

The association between two categorical variables is assessed using Pearson's $\chi^2$-test. For the test, a contingency table, which lists the categorical frequency of two variables, is created. For example, $O_{ij}$ of the $i$- and $j$-th element of the contingency table shown below is the frequency counting the putative valence $i = 1$ (attractive), 2 (aversive), 3 (unknown), and the number of uPNs in one of the 10 clusters in MB calyx with $j = 1$ ($C_1^{\text{MB}}$), 2 ($C_2^{\text{MB}}$), ... , 10 ($C_{10}^{\text{MB}}$).

|  | $C_1^{\text{MB}}$ | $C_2^{\text{MB}}$ | $C_3^{\text{MB}}$ | $C_4^{\text{MB}}$ | $C_5^{\text{MB}}$ | $C_6^{\text{MB}}$ | $C_7^{\text{MB}}$ | $C_8^{\text{MB}}$ | $C_9^{\text{MB}}$ | $C_{10}^{\text{MB}}$ | Total |
|---|---|---|---|---|---|---|---|---|---|---|---|
| Attractive | 0 | 4 | 0 | 1 | 0 | 5 | 4 | 11 | 11 | 8 | 44 |
| Aversive | 1 | 2 | 0 | 0 | 4 | 12 | 9 | 8 | 8 | 3 | 47 |
| Unknown | 4 | 7 | 8 | 5 | 6 | 5 | 1 | 2 | 3 | 3 | 44 |
| Total | 5 | 13 | 8 | 6 | 10 | 22 | 14 | 21 | 22 | 14 | 135 |

Then the $\chi^2$ value is evaluated based on the table using.

$$\chi^2 = \sum_{i=1}^{R} \sum_{j=1}^{C} \frac{(O_{ij} - E_{ij})^2}{E_{ij}}, \tag{5}$$

where $R$ and $C$ are the numbers of rows and columns, and $O_{ij}$ and $E_{ij}$ are the observed and expected frequencies of the event in the $i$-th row and $j$-th column, respectively. $E_{ij}$ is calculated from $O_{ij}$ as.

$$E_{ij} = Np_{i\cdot}p_{\cdot j}, \tag{6}$$

where $p_{i\cdot} = \sum_{j}^{C} O_{ij}/N$ and $p_{\cdot j} = \sum_{i}^{R} O_{ij}/N$ with $N$ being the total count. Thus, $E_{ij}$ is the frequency expected by assuming that the two categorical data are statistically independent. Pearson's $\chi^2$ test aims to check whether there is a significant difference between $O_{ij}$ and $E_{ij}$.

In the $\chi^2$-test, the p-values are estimated using $f_k(x)$, the $\chi^2$-distribution with the degree of freedom $k = (R-1)(C-1)$. If the test returns a $\chi^2$ value that gives rise to a p-value smaller than the defined significance level ($\alpha = 0.01$), the null hypothesis of independence between the two data sets should be rejected. As a result, the distribution of the categorical data is deemed significantly different from a randomly generated distribution, which concludes that the association between two sets of data is statistically significant.

For the above contingency table with $k = 18$, which leads to $\chi^2 \approx 66.1$ (*Equation 5*), we get a p-value much smaller than the significance level ($\alpha = 0.01$), $p = 1 - \int_0^{\chi^2} f_{k=18}(x)dx \approx 2.016 \times 10^{-7} \ll \alpha = 0.01$.

When Pearson's $\chi^2$ statistics are available, one can calculate Cramér's $V$ with bias correction, a measure of association between two categorical variables, as follows.

$$V = \sqrt{\frac{\phi'^2/N}{\min\left(R'-1, C'-1\right)}}, \tag{7}$$

where $\phi'^2 = \max\left(0, \chi^2/N - (R-1)(C-1)/(N-1)\right)$, $R' = R - (R-1)^2/(N-1)$, and $C' = C - (C-1)^2/(N-1)$. Similar to the Pearson correlation coefficient, the value $V$ ranges between 0 and 1 where 0 indicates no correlation and 1 indicates a complete correlation between two categorical variables.

## Mutual information

Mutual information ($I$) is used to verify the significance of association between nominal variables observed in Pearson's $\chi^2$-test for independence. The $I$ measures the information transfer or the similarity between two data. The concept can be extended to clustering outputs to check how two different clustering labels from the same data are similar to each other. Traditionally, the $I$ between two jointly discrete variables $A$ and $B$ is given by.

$$I(A; B) = \sum_{i=1}^{n_A} \sum_{j=1}^{n_B} P(A_i, B_j) \log\left[\frac{P(A_i, B_j)}{P(A_i)P(B_j)}\right], \tag{8}$$

where $n_A$ (or $n_B$) is the number of clusters in $A$ (or $B$). Numerically, the $I$ between two clustering outputs $A$ and $B$ is calculated by evaluating $P(A_i) = N_{A_i}/N$, $P(B_i) = N_{B_i}/N$, and $P(A_i, B_j) = N_{A_i \cap B_j}/N$ where $N$ is the total count and $N_{A_i \cap B_j}$ is the number of elements common in both clusters $A_i$ and $B_j$.

The significance was assessed by comparing the observed $I$ with the distribution of $I$ s from randomly sampled variables. Specifically, the cluster label was randomly sampled 1000 times to generate a distribution of $I$ under the assumption of independence. The value of observed $I$ is considered significant if the approximated p-value is below 0.01 (p< 0.01).

## Analysis of synaptic interfaces

We conducted three different analyses on the synaptic interfaces of uPNs with the third-order neurons (KCs or LHNs) from the hemibrain dataset.

(i) The 'homotype-specific' connections ($N_{X,\text{sp}}^{\xi}$ with $\xi = $ PN-KC or PN-LHN) are obtained by counting the number of third-order neurons that synapse with a homotype $X$ but do not synapse with any other homotypes, the information of which is provided by the binarized connectivity matrix $\mathcal{C}$. The total

number of synaptic connections for a homotype $X$ is simply the sum of the row of the connectivity matrix $\mathcal{C}$ ($N_{X,\text{tot}}^{\xi} = \Sigma_{i=1}^{N_\xi} \mathcal{C}_{Xi}$).

(ii) To generate the $\mathcal{S}$ matrices, we counted the number of third-order neurons synapsing with a given homotype $X$ that also synapses with other homotypes.

(iii) The tanglegram study required a hierarchical clustering of uPNs based on their connectivity to third-order neurons. The distances between uPNs in the connectivity matrix $\mathcal{C}$ represent the similarity of the connectivity patterns to third-order neurons between two uPNs. We utilized the metric of cosine distance, which is widely used for analyzing the connectivity matrix (*Bates et al., 2019*; *Bates et al., 2020*; *Li et al., 2020*; *Eschbach et al., 2020*; *Schlegel et al., 2021*). The cosine distance is defined as.

$$d_{\cos} = 1 - \frac{u \cdot v}{|u|\,|v|}, \tag{9}$$

where $u$ and $v$ are two vectors to be compared. After calculating the distances, we performed hierarchical clustering by Ward's criterion, which minimizes the variance of merged clusters, to generate the tree structure. The results of hierarchical clustering using the spatial proximity ($d_{\alpha\beta}$) and connectivity ($d_{cos}$) are compared using a tanglegram (*Figure 12*) after untangling two trees using the 'step-1side' method (*Galili, 2015*).

## Acknowledgements

We thank Dr. Ji Hyun Bak for helpful discussions. This study was supported by KIAS Individual Grants CG077001 (KC), CG076002 (WKK), and CG035003 (CH). We thank the Center for Advanced Computation in KIAS for providing the computing resources.

## Additional information

### Funding

| Funder | Grant reference number | Author |
| --- | --- | --- |
| KIAS individual grant | CG077001 | Kiri Choi |
| KIAS individual grant | CG076002 | Won Kyu Kim |
| KIAS individual grant | CG035003 | Changbong Hyeon |

The funders had no role in study design, data collection and interpretation, or the decision to submit the work for publication.

### Author contributions

Kiri Choi, Conceptualization, Data curation, Software, Formal analysis, Funding acquisition, Validation, Investigation, Visualization, Methodology, Writing - original draft, Writing – review and editing; Won Kyu Kim, Conceptualization, Funding acquisition, Validation, Investigation, Visualization, Methodology, Writing – review and editing; Changbong Hyeon, Conceptualization, Resources, Supervision, Funding acquisition, Validation, Investigation, Visualization, Methodology, Writing - original draft, Project administration, Writing – review and editing

### Author ORCIDs

Kiri Choi ⓘ http://orcid.org/0000-0002-0156-8410
Won Kyu Kim ⓘ http://orcid.org/0000-0002-6286-0925
Changbong Hyeon ⓘ http://orcid.org/0000-0002-4844-7237

### Decision letter and Author response

Decision letter https://doi.org/10.7554/eLife.77748.sa1
Author response https://doi.org/10.7554/eLife.77748.sa2

# Additional files

## Supplementary files

- Transparent reporting form
- Source data 1. Python scripts used in this study.

## Data availability

All data generated during this study and Python script are available in *Drosophila* Olfaction-main.zip included as the supporting file. They are also available at https://github.com/kirichoi/DrosophilaOlfaction, (copy archived at swh:1:rev:91dd60f4231a58590e2571e72b660c5dfee261b6).

The following previously published datasets were used:

| Author(s) | Year | Dataset title | Dataset URL | Database and Identifier |
|---|---|---|---|---|
| Bates AS, Schlegel P, Roberts RJV, Drummond N, Tamimi IFM, Turnbull R, Zhao X, Marin EC, Popovici PD, Dhawan S, Jamasb A, Javier A, Capdevila LS, Li F, Rubin GM, Waddell S, Bock DD, Costa M, Jefferis GSXE | 2020 | Complete Connectomic Reconstruction of Olfactory Projection Neurons in the Fly Brain | https://fafb.catmaid.virtualflybrain.org/ | FAFB, catmaid |

*Continued*

| Author(s) | Year | Dataset title | Dataset URL | Database and Identifier |
|---|---|---|---|---|
| Scheffer LK, Xu CS, Januszewski M, Lu Z, Takemura SY, Hayworth KJ, Huang GB, Shinomiya K, Maitlin-Shepard J, Berg S, Clements J, Hubbard PM, Katz WT, Umayam L, Zhao T, Ackerman D, Blakely T, Bogovic J, Dolafi T, Kainmueller D, Kawase T, Khairy KA, Leavitt L, Li PH, Lindsey L, Neubarth N, Olbris DJ, Otsuna H, Trautman ET, Ito M, Bates AS, Goldammer J, Wolff T, Svirskas R, Schlegel P, Neace E, Knecht CJ, Alvarado CX, Bailey DA, Ballinger S, Borycz JA, Canino BS, Cheatham N, Cook M, Dreher M, Duclos O, Eubanks B, Fairbanks K, Finley S, Forknall N, Francis A, Hopkins GP, Joyce EM, Kim S, Kirk NA, Kovalyak J, Lauchie SA, Lohff A, Maldonado C, Manley EA, McLin S, Mooney C, Ndama M, Ogundeyi O, Okeoma N, Ordish C, Padilla N, Patrick CM, Paterson T, Phillips EE, Phillips EM, Rampally N, Ribeiro C, Robertson MK, Rymer JT, Ryan SM, Sammons M, Scott AK, Scott AL, Shinomiya A, Smith C, Smith K, Smith NL, Sobeski MA, Suleiman A, Swift J, Takemura S, Talebi I, Tarnogorska D, Tenshaw E, Tokhi T, Walsh JJ, Yang T, Horne JA, Li F, Parekh R, Rivlin PK, Jayaraman V, Costa M, Jefferis GS, Ito K, Saalfeld S, George R, Meinertzhagen IA, Rubin GM, Hess HF, Jain V, Plaza SM | 2020 | A connectome and analysis of the adult Drosophila central brain | https://doi.org/10.25378/janelia.12818645.v1 | figshare, 10.25378/janelia.12818645.v1 |

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

## Appendix 1

### Monte Carlo approach to independence test

In this section, we describe an alternative method to the independence test inspired by the Monte Carlo significance test (*Hope, 1968*) to further support our Pearson's $\chi^2$-test of independence. The procedure is as follows: (1) For a given contingency table, randomize the observation such that the marginal sum of each row remains the same as the observed contingency table. That is, for each row, randomize the vector with integers while the sum of the vector stays the same as the observed contingency table. This procedure randomly shuffles the distribution of the clusters while keeping the distribution of a particular categorical variable intact. (2) Calculate the $\chi^2$ value from the randomized contingency table. (3) Repeat steps 1 and 2 for 1000 times to generate a distribution of the $\chi^2$ values. (4) Obtain the mean and the standard deviation of $\chi^2$ values. The distribution of $\chi^2$ values is approximately normal. (5) If the $\chi^2$ value from the observed contingency table is more than $4\sigma$ different from the distribution, we consider the observed $\chi^2$ value statistically significant and reject the null hypothesis. Whenever we ran a Pearson's $\chi^2$-test, we performed the above procedure alongside. The output of this procedure supported whichever conclusion we drew from Pearson's $\chi^2$-test.

### Identifying the agreement between two categorical data via mutual information

We verified our Pearson's $\chi^2$-test of independence of two categorical variables by calculating the mutual information $I$ (see Methods). In the FAFB dataset, the mutual information between glomerular labels and $d_{\alpha\beta}$-based clustering output in MB calyx was equal to $I(\text{glo}; C^{\text{MB}}) = 1.892$, which is significantly (more than $4\sigma$) different from the mean of randomly sampled $I$ distribution, $I(\text{glo}; C^{\text{MB}})_{\text{rand}} = 1.386 \pm 0.035$. This result is consistent with our $\chi^2$-test, as the mutual information of the observed variables is significantly larger than the mutual information under the assumption of random sampling, suggestive of a statistically significant association between glomerular labels and MB calyx cluster labels. In LH, the mutual information between glomerular labels and $d_{\alpha\beta}$-based clustering output was $I(\text{glo}; C^{\text{LH}}) = 2.128$ which deviated $4\sigma$ or more from the mean of the randomly sampled $I$ distribution, $I(\text{glo}; C^{\text{LH}})_{\text{rand}} = 1.466 \pm 0.035$.

The same method is applied to confirm that a statistically significant association exists between odor type and the clustering outputs, with $I(\text{odor}; C^{\text{MB}}) = 0.819$ and $I(\text{odor}; C^{\text{LH}}) = 0.963$, all of which differ by more than $4\sigma$ from the means of the randomly sampled $I$ distributions, $I(\text{odor}; C^{\text{MB}})_{\text{rand}} = 0.337 \pm 0.044$ and $I(\text{odor}; C^{\text{LH}})_{\text{rand}} = 0.372 \pm 0.043$. For odor valence, we obtain $I(\text{val}; C^{\text{MB}}) = 0.277$ and $I(\text{val}; C^{\text{LH}}) = 0.326$, where both $I(\text{val}; C^{\text{MB}})$ and $I(\text{val}; C^{\text{LH}})$ differ significantly from the means of the randomly sampled $I$ distributions, $I(\text{val}; C^{\text{MB}})_{\text{rand}} = 0.073 \pm 0.026$ and $I(\text{val}; C^{\text{LH}})_{\text{rand}} = 0.081 \pm 0.026$. Overall, the conclusion drawn from the association study based on mutual information is identical to Pearson's $\chi^2$-test.

**Appendix 1—table 1.** Pearson's $\chi^2$ tests of independence of variables in the FAFB dataset. $C^Z$ indicates cluster labels from $d_{\alpha\beta}$-based clustering in $Z$ neuropil. Cramér's V values are displayed on each cell and the corresponding p-values are shown in parentheses.

| $C^{\text{LH}}$ | Glomerular Labels | Odor Type | Odor Valence |
|---|---|---|---|
| $C^{\text{MB}}$ 0.502 (1.149E-36) | 0.610 (1.255E-27) | 0.401 (3.303E-21) | 0.425 (2.016E-07) |
| $C^{\text{LH}}$ | 0.671 (2.266E-40) | 0.416 (1.980E-22) | 0.455 (2.586E-08) |

**Appendix 1—table 2.** Pearson's $\chi^2$ tests of independence of variables in the hemibrain dataset. $C^Z$ indicates cluster labels from $d_{\alpha\beta}$-based clustering in $Z$ neuropil. Cramér's V values are displayed on each cell and the corresponding p-values are shown in parentheses.

| $C^{\text{LH}}$ | Glomerular Labels | Odor Type | Odor Valence |
|---|---|---|---|
| $C^{\text{MB}}$ 0.495 (6.635E-34) | 0.577 (3.461E-25) | 0.425 (9.400E-18) | 0.463 (6.283E-07) |
| $C^{\text{LH}}$ | 0.685 (1.523E-40) | 0.502 (6.072E-29) | 0.521 (2.932E-09) |

**Appendix 1—table 3.** Mutual information (observed mutual information (top), randomly sampled mutual information (bottom) in each cell) from the association study using the FAFB dataset. $C^Z$ is cluster labels from $d_{\alpha\beta}$-based clustering at $Z$ neuropil. The observed mutual information differs from the randomly sampled mutual information by more than $4\sigma$.

| $C^{LH}$ | | Glomerular Labels | Odor Type | Odor Valence |
|---|---|---|---|---|
| $C^{MB}$ | 1.076 | 1.892 | 0.819 | 0.277 |
| | 0.397±0.045 | 1.386±0.035 | 0.337±0.044 | 0.073±0.026 |
| $C^{LH}$ | | 2.128 | 0.963 | 0.326 |
| | | 1.466±0.035 | 0.372±0.043 | 0.081±0.026 |

**Appendix 1—table 4.** Mutual information (observed mutual information (top), randomly sampled mutual information (bottom) in each cell) from the association study using the hemibrain dataset. $C^Z$ is cluster labels from $d_{\alpha\beta}$-based clustering at $Z$ neuropil. The observed mutual information differs from the randomly sampled mutual information by more than $4\sigma$.

| $C^{LH}$ | | Glomerular Labels | Odor Type | Odor Valence |
|---|---|---|---|---|
| $C^{MB}$ | 1.371 | 2.244 | 1.036 | 0.336 |
| | 0.710±0.048 | 1.783±0.033 | 0.527±0.047 | 0.124±0.035 |
| $C^{LH}$ | | 2.344 | 1.211 | 0.434 |
| | | 1.717±0.034 | 0.493±0.048 | 0.116±0.033 |

**Appendix 1—table 5.** Statistics of homotypes composed of a single uPN (or multiple uPNs) in the FAFB dataset and the corresponding putative valence.

| | Aversive | Attractive | Unknown | Total |
|---|---|---|---|---|
| Single uPN Homotypes Count | 7 | 4 | 2 | 13 |
| Multiple uPN Homotypes Count | 18 | 13 | 13 | 44 |
| Total | 25 | 17 | 15 | 57 |

**Appendix 1—table 6.** Statistics of homotypes composed of a single uPN (or multiple uPNs) in the hemibrain dataset and the corresponding putative valence.

| | Aversive | Attractive | Unknown | Total |
|---|---|---|---|---|
| Single uPN Homotypes Count | 7 | 5 | 1 | 13 |
| Multiple uPN Homotypes Count | 18 | 12 | 15 | 45 |
| Total | 25 | 17 | 16 | 58 |

## Appendix 2

### Testing the labeled-line hypothesis

We detail the analyses performed on the tanglegram and the respective outputs (*Figure 12*). First, we applied the dynamic hybrid cut tree method on the dendrogram generated from connectivity and conducted Pearson's $\chi^2$ test. The results are shown in *Table 1*. The p-values for the connectivity-based clustering between uPNs and LHNs for glomerular labels, odor types, and odor valence were very small. For the connectivity between uPNs and KCs, we see a moderate to no association for the given categorical variables (*Appendix 2—table 1*).

The similarity between two tree structures from spatial proximity-based and connectivity-based clustering at a given synaptic interface is measured in several different ways to provide a comprehensive comparison. First, we quantified the similarity using Baker's Gamma index (*Baker, 1974*), which is a measure of rank correlation (or ordinal relation) calculated from concordant and discordant pairs given by.

$$G_{\text{Baker}} = \frac{N_{\text{con}} - N_{\text{dis}}}{N_{\text{con}} + N_{\text{dis}}}, \tag{10}$$

where $N_{\text{con}}$ is the number of concordant pairs (the ordering of elements in two trees match) and $N_{\text{dis}}$ is the number of discordant pairs (the ordering of elements in two trees do not match). Baker's Gamma index ranges from -1 to 1 where 0 represents the ordering of two trees is completely dissimilar and 1 or -1 indicate the ordering of two trees match. We find $G_{\text{Baker}}^{\text{MB}} = 0.286$ and $G_{\text{Baker}}^{\text{LH}} = 0.219$ (which we double-checked using both the in-house code and 'dendextend' package in R). Baker's Gamma index for LH is very similar to the one obtained by *Bates et al., 2020* ($G_{\text{Baker}}^{\text{LH}} = 0.21$), who conducted a similar study using the NBLAST score and connectivity. However, the fact that $G_{\text{Baker}}^{\text{MB}} > G_{\text{Baker}}^{\text{LH}}$ when the tanglegram of MB calyx is seemingly more incoherent (*Figure 12A*) raises a question of whether Baker's Gamma index alone is enough to describe the tanglegram.

Apart from the ordinal relations between two sets of leaves, we employed two additional metrics to compare the two trees: (1) entanglement, a measure spanning from 0 to 1 quantifying the number of lines crossing, and (2) cophenetic distance correlation, a measure spanning from 0 to 1 quantifying how similar the two branching structures are. The entanglement between two trees for MB calyx was 0.35 (higher entanglement), while the entanglement for LH was 0.26 (lower entanglement), which agrees with *Figure 12*. To calculate cophenetic distance correlation, we measured the pairwise cophenetic distances within each tree and calculated the Pearson correlation coefficient. The cophenetic distance between two leaves in the dendrogram is equal to the minimum distance (or height) to the branching point that contains both leaves. The Pearson correlation coefficient between cophenetic distances of the spatial proximity-based and connectivity-based tree structures was $r = -0.032$ ($p > 0.001$) for MB calyx and $r = 0.236$ ($p \ll 0.001$) for LH, reflecting the less disrupted tree structure in LH compared to MB calyx.

**Appendix 2—table 1.** Pearson's $\chi^2$ tests of independence of variables on the connectivity-based clustering results.

Cramér's V values are displayed on each cell and the corresponding p-values are shown in parentheses. Bold entries are used to specify statistically significant results.

| | Glomerular Labels | Odor Type | Odor Valence |
|---|---|---|---|
| $C^{\text{PN}-\text{KC}}$ | 0.433 (2.472E-08) | 0.316 (9.978E-09) | 0.271 (0.012) |
| $C^{\text{PN}-\text{LHN}}$ | 0.765 (1.410E-67) | 0.630 (1.519E-48) | 0.604 (4.055E-12) |

