## [Editor Report]

Choi et al. explore how olfactory information flows across the three major neuropils in the *Drosophila* brain – the antennal lobe (AL), mushroom Body (MB), and the lateral horn (LH). They use the two connectomes of adult *Drosophila* and 'inter-PN distances' to do this. Using this neuroanatomy based approach, they find support for a labeled-line strategy, which they subsequently test for with synaptic connectivity data for a subset of PNs. They find that while some labelled lines may exist, PNs generally participate in multi-channel integration at the MB and LH. This manuscript will be of interest to neuroscientists interested in olfactory processing and to those working on connectomic-level circuit analysis.

---

## [Decision Letter]

**Decision letter after peer review:**

Thank you for sending your article entitled "Olfactory responses of *Drosophila* are encoded in the organization of projection neurons" for peer review at *eLife*. Your article is being evaluated by 3 peer reviewers, one of whom is a member of our Board of Reviewing Editors, and the evaluation is being overseen by K VijayRaghavan as the Senior Editor.

While appreciating the authors' work presented in this manuscript there were two major concerns that came up during the consultations:

1. Given that two EM datasets exist for the fly brain, determining the generalisability of what the authors describe should have been done.

2. Making claims about labelled line representation of odours or not using morphometric data alone is not justified when connectivity data exists for flies.

Based on this, and other comments, the essential revisions that were discussed were the following:

The writing: The text needs to be made more accessible to the reader, perhaps by providing more intuitive explanations, biological context, and emphasise their findings more than they currently do (which emphasises methods used more). We would also like to see their analyses in the context of what's been done before.

Reproducibility: Since the EM data in this analysis is based on a single fly, we recommend testing whether the authors' findings will be consistent across individuals. So, could the authors test their analysis on the hemibrain alongside the extant FAFB?

Number of uPNs used: The authors use 111 uPNs for their analysis whereas the number of uPNs is 164. (This might be why they obtain only 31 glomeruli). It's not clear why the rest were excluded. Could the authors conduct their analyses by including all cholinergic uPs (including those involved in hygro and thermo sensation)? (We assume the authors have omitted GABAergic neurons from their analyses because they do not innervate the MB? Including it would provide important biological insight, but if not, their reasoning needs to be stated.)

Hierarchical clustering: We are concerned that the inferences rely on a few clusterings that don't appear to be very robust. This is usually okay, but a lot of the paper is built around this. So, could the authors reassess their hierarchical clustering parameters and the normalisation for the λ value and state them explicitly for the reader?

Labelled line hypothesis: The claims made in this study about the labelled line hypothesis cannot be inferred from morphometrics alone. For this, the authors would need to compare these findings with connectivity. This could even be done at a small scale to understand whether their morphological groups also have more specific or segregated connectivity within the LH / AB.

(A possible pipeline for this could be: Use neuprint-python to pull connection data for each chosen PN and subset it to the select brain regions. They could then perform hierarchical clustering on that, and devise groupings that can be compared to the morphological grouping visually using a tanglegram. They might then compare the two using Baker's Γ Index or similar.)

In addition to this, please read the other (not essential) recommendations and if any of them are feasible to respond to, please do so.

*Reviewer #1 (Recommendations for the authors):*

The olfactory PNs have been studied using numerous approaches. In this study, the authors use a parameter called 'inter PN distance' to assess the relationship between all the uniglomerular PNs in the hemibrain connectome in the AL, MB, and LH. As odour information from the sensory neurons is brought into the AL via OSNs classes that innervate individual glomeruli, PNs that innervate the same glomeruli are considered homotypic. Using the inter PN distance and downstream statistical analyses, the authors infer that odour information is discretely represented in the AL and less so in the LH; however, in the MB, there is considerable overlap in odour representation.

The manuscript is well written and presented.

- PN information representation across the AL, MB, and LH has been explored before. While the authors do mention this, it would be useful if they discussed what each one of those approaches inferred about olfactory representation, and why they felt their approach was better. In the same vein, it would be nice if they finally concluded by comparing their findings with the others.

- The manuscript is statistically dense. It will help readers if the authors provided some intuitive explanations along the way – for example, for inter-PN distance, among others.

- Many of the legends are not very explanatory. Could the authors please elaborate on them? It would also help to have pointers in the figures to focus the attention of the reader. For example, the legend for S2 can be expanded. I can't tell which homotypic PNs are merging into a single cluster. There are similar issues in other figures as well – could the authors please add visual aids such as arrows in the figures?

*Reviewer #2 (Recommendations for the authors):*

Choi et al. competently quantify known aspects of excitatory, uniglomerular projection neuron (uPN) morphology, in order to demonstrate how neurons of this well-studied cell class differ in the three main brain regions in which they arborise. Classically, these three regions support three different functions: siloed olfactory channel processing (antennal lobe), ethological odour classification (lateral horn), and associative memory (mushroom body). They find that uPNs of the same cell type are morphologically similar in each region, but the nature of that similarity, and their similarity with other uPN cell types, differ as appropriate for the function of the neuron in that region. While not novel, their quantitive description contributes to the field. Their 'major' conclusion on labeled line processing is, however, insufficiently substantiated given that the olfactory system of the fly is post-connectomic and this question is best taken up with connectionist analysis. They also only use a single dataset from a single fly (though a second is available) and do not explore how robust their results are to other clustering approaches/decisions, making it unclear how general their description is.

The strength of this work is that it is able to use quantitative methods to demonstrate features of *Drosophila* olfactory projection neuron (PN) anatomy. To date, these features have been known and used in classification work on the PNs (e.g. Tanaka et al. 2012; Bates et al. 2020; Zheng et al. 2018; Bates et al. 2020; Jefferis et al. 2007). They have also been shown through (PN->target) connection analyses (Eichler et al. 2017; Schlegel et al. 2021). The findings are therefore largely not novel but are a good quantitative description and a nice synthesis in one place. This work would be cited in the field for the following point: within a cell type, PNs tile their input glomerulus with their dendrites, intermingle their axons with other PN cell types in the mushroom body, and overlap with each other, and to a lesser degree other PN cell types, in the lateral horn. The authors' other main conclusions would take more to support.

In this work, the authors first (A) choose the neurons they will examine, and then perform two main analyses, (B) pair-wise morphology similarity analysis within three major projection regions for these neurons and then (C) an analysis of 'homotypic' PN cell types and how similarity within and between the cell type varies between the three projection areas. They then (D) conclude that their results are indicative of 'form determining function' and 'labelled line' encoding at play in the *Drosophila* olfactory system, where 2nd and 3rd order neurons meet.

On A: The authors use data from the FAFB connectome. These are manually reconstructed neuronal skeletons, built using CATMAID. The dataset at present offers a sparse connectome, though a denser one using the same EM data is currently being built by the Flywire project (Dorkenwald et al. 2022), open to anyone for contributions. A dense connectome for the olfactory system already exists in a separate EM dataset, thanks to the hemibrain project (Scheffer et al. 2020). One of the limitations of the present work is that it assumes that the data it uses from FAFB (PNs on a single hemisphere of a single fly) are representative of all flies, and does not discuss this assumption (which is a fair enough assumption given how morphologically stereotyped the system appears to be (Jenett et al. 2012; Schlegel et al. 2021; Jeanne et al. 2018) but should be discussed; the degree of connection stereotypy is still unsettled, and there could be variation in how well neurons of a homotype may fasciculate across brains). The authors could better support their statements by re-running their analysis on the hemibrain data and comparing it to their CATMAID FAFB data. It would greatly strengthen their study to show that the patterns they observe hold across two EM data sets. (Light-level co-registered datasets also exist via the flycircuit project (Shih et al. 2015; Chiang et al. 2011; Costa et al. 2016), though registration offsets likely limit the effectiveness of the authors' present analyses – the authors could consider it).

The authors' analysis focuses on 111 PNs. This is close to the number of uniglomerular projection neurons (uPNs) in the right hemisphere. The actual count from Zheng et al. is 114. The updated count for the same dataset is 164 (Bates et al. 2020). The decision to not use a full complement of uPNs is not adequately justified. (The authors state that they dropped three neurons that are not PNs, including the APL – those neurons are not in the numbers I give above, and the Zheng et al. archiver must have given them the wrong label.) In addition, uPNs have two functionally important sub-divisions – there are excitatory uPNs (cholinergic) and inhibitory uPNs (GABAergic). This distinction, and how the two could differ in terms of which clusters they fall into, is significant because it might reveal how different odour channels can synergise with, or antagonise one another. I think the GABAergic uPNs are left out because they have no MB arbors, but the existence of these different PN groups and the choice to exclude some biological classes are not made clear in the main text.

Indeed, 6 neurons are said to be dropped because 'they did not project to any of the neuropils'. This does not seem like a great reason for dropping these cells completely from the paper. If the point the authors wish to make is that there are labeled lines among the olfactory PNs, surely the ones that stray off into other brain regions are the most evidential! While they do not say it directly, the authors are looking at cholinergic, olfactory uniglomerular PNs and excluding other PN classes – which is fine, but could be made clear. (There are a total of 347 PNs covering all classes, on the right hemisphere in FAFB).

On B: The authors use an all-by-all pair-wise morphology similarity metric to construct a score matrix and then perform hierarchical clustering. The metric is not adequately explained. In reviewing the methods, the metric appears, by the author's own note, to be similar to NBLAST (Costa et al. 2016). There are many different and valid reasons for using different metrics, but it is not clear to me why the authors did not use NBLAST, and how their own method differs. Perhaps results from both would be similar, and it does not really matter – or maybe they found that their own metric is more performant in this situation? The authors, I should be clear, do not claim to have invented a 'better' metric – but some explanation of their choice is warranted. The terms in the author's equation are not adequately explained, but it seems to be a distance score calculated between all points between two neurons in Euclidean space, that decays exponentially with distance, which is common practice (Schlegel et al. 2016; Strutz et al. 2014; Kohl et al. 2013; Masse et al. 2012). Some explanation to that effect should appear in the main text. (I opine that since there are now many studies examining PN morphology with similar metrics, it would be easier if the same algorithms were applied when there is not a good reason to do otherwise, so that results can be directly compared, NBLAST is currently the most popular choice).

In examining their scores, the authors do not give their clustering method, just stating 'a method of hierarchical clustering was used'. Drawing conclusions from hierarchical clustering is very sensitive to the cut height used. It would be nice if the authors showed the result of their clustering as, for example, a dendrogram with the cut height they used indicated by a horizontal line in the main sequence (this is currently S2). Their cut height was determined by a silhouette method for determining optimal clusters. Other common, valid methods include the elbow and gap statistic methods – perhaps in supplement, the authors could give or summarise how the result would have been using those, or in the methods, explain why they could not? How robust is their analysis to different clustering methods, different numbers of clusters, etc? The details of an unsupervised clustering analysis are often skimmed over in papers because we trust the authors were able to tweak things to bring their biological conclusions to the foreground, and that is fine. However, since this paper is all about the results of the authors' clustering analysis, the detail and how robust their analysis is, are very important. Their analysis yields 39 clusters in AL, 3 clusters in MB calyx, and 4 clusters in LH, and there is a nice visualisation in Figure S6 that could use some vertical lines to show the shown cluster numbers. However, those 4 LH clusters could easily be broken down further into smaller groups that look meaningful to a neuroanatomist. The authors use a Pearson's Chi-squared test to observe that their clusters are statistically different from a random sample of PNs, which is great. The authors could re-word this segment though, with simpler language, to let the reader qualitatively know what they are doing and why.

The authors do not consider their MB clusters in-depth or explain what one would expect from the field. The literature suggests that PN-KC connections in the field are either random or semi-random (Eichler et al. 2017; Zheng et al. 2020; Caron et al. 2013). The authors could discuss their results in this context, in particular, their conclusion that 'the distinction between the homotypes is only weakly present in MB calyx'. The structure of the PN-MB clusters might be compared to results from Zheng et al. 2020, who show that the calyx contains a 'fovea' of Kenyon cells oversampling particular PNs, specifically PNs that are from 'food-related' glomeruli. This PN group is probably captured somewhere in the authors' analysis. Interestingly, the authors show that pheromonal largely PNs cluster together in the MB, though they do not draw as much attention to this as the LH pheromonal grouping for some reason – could this be a pheromonal 'fovea'?.

In the LH, the authors focus on their findings of a pheromone-specific cluster within the LH. However, one result they also have is that the other clusters have a mixture of different odour class PNs, so the LH does not easily break down into siloed compartments like the AL. A few studies have tried to break down the LH into different regions that serve different sections of an ethological odour space, based on different analyses (Strutz et al. 2014; Bates et al. 2020; Frechter et al. 2019). If the authors could at least qualitatively compare their groups to the previous groupings, their results and contribution would make more sense in the context of the field.

On C: The authors are interested in looking at 'homotype' uPNs – neurons with dendrites in the same AL glomerulus. This is a novel and interesting approach. The intra-homotypic-PN-distance (a 'bundling' metric) and inter-homotypic-PN-distance scores (a 'packing' metric) they use are insufficiently explained in the text and are better understood looking at figure 4. Figure 4 is a good, concise summary. Because sister PNs are essentially copies of the same cell and occupy the same space – a phenomenon seen across the whole of the *Drosophila* nervous system – it is unsurprising that they arborise similarly in each of the three analysis regions – the authors show, however, that they are 'morphologically similar in different ways' between the AL, the MB and the LH, which is an interesting point. However, the text does not make it very clear to the reader what the authors have shown. On the intra-homotypic-PN-distance: sister neurons are similar in all three regions – though in the AL they seem to tile glomeruli rather than -- intermingle (Figure 4). This last point is not, I think, shown quantitatively, only schematically, – it would be best to formalise and plot somewhere. Considering the inter-homotypic-PN-distance: neurons interdigitate across homotypes in the MB, to a lesser degree in the LH, and barely in the AL, instead of tiling the whole AL.

The authors explore how their morphology findings might correlate with the odour response properties of PNs. They use odour categories from Bates et al. 2020, though in many cases a PN type was given multiple categories (see the supplemental figures), and it is not clear how the authors resolved this conflict. For example, the V glomerulus can also be viewed as 'aversive/bad' as it encodes CO2. Often a neuron cannot be given a singular 'odor scene' label, though they might be clustered using odour response data (Badel et al. 2016).

On D: In the conclusion, the authors do not discuss their own results in relation to the field that much. They only mention (i) one non-novel finding and (ii) one observation. The (i) finding is that certain pheromonal PNs segregate together in the LH (Ruta et al., 2010; Kohl et al., 2013, Frechter et al., 2019; Chakraborty and Sachse, 2021) – although their finding of the same in the MB is perhaps more novel? The authors say that "Our study not only lends support to the existing studies pointing to the labeled-line strategy in the *Drosophila* olfactory system but also suggests that an even more sophisticated level of spatial organization that depends on the homotypes and odor should be present"; it is unclear to me what the second half of the sentence means. The (ii) observation, implicit also in other work (Grabe et al. 2016) – that some PN cell types exist as singletons, often aversive ones. Certainly, in insects, there is some circuit-level difference between cell types that exist in multiples or as singletons – which are often larger, older neurons – but the present paper has no analysis on this point. Perhaps the authors could show how these singletons are morphologically special – e.g. they mention that they are more 'dense' but do not quantify this anywhere in the paper.

One of the authors' key conclusions and a point in their abstract, is that a labeled line strategy is at work in the *Drosophila* olfactory system at the point that second-order projections ramify in the lateral horn (LH). They make expansive statements such as: "Overall, our findings suggest that the *Drosophila* olfactory system leverages the efficiency of the labeled-line design in sensory information processing". A 'labeled line' is generally taken to be a chain of neurons that transfers a message about a single feature, onto higher-order neurons. This message may be modified and transformed along the way, but is generally not directly integrated with information about other features. The very peripheral part of the system in the antennal lobe (AL), ORNs -> PNs, is generally established to be a set of labeled lines (Couto et al., 2005; Fishilevich and Vosshall, 2005; Vosshall et al., 2000). Although significant cross-talk does exist, e.g. through AL local neuron computation, at this stage the PNs can still act as labeled lines (Olsen et al. 2010; Seki et al. 2017), which could support a somewhat labeled synfire chain up to the 3rd order (Jeanne and Wilson 2015).

The authors do not define the term 'labeled line' clearly, or whether the label in question should be glomerular identity, odour identity, or odour scene. They also do not clearly state (a) what neurons they mean to say are in the labeled line, nor (b) whether they think this is largely the case, or only in specific channels – their sweeping statements suggest they mean the former, but their data only weakly support the later. I assume, for (a) they are suggesting that some lateral horn neurons (LHNs) will be part of labeled lines, but not Kenyon cells of the mushroom body, because of the PN morphological properties that the authors quantify in the LH and MB. For (b), established work in the field suggests a murky division within the LH to support odour categorisation (Frechter et al. 2018; Jeanne et al. 2018) – a convergence scheme, and so, not a labeled line system – and the authors' own results show PNs of different, broad odor classes intermingling in 3/4 of their LH clusters. Their analysis, for (b), seems to show that only a small subset of PNs have the appropriate morphology to support labeled line connections in the LH. The main PNs that may form a labeled line, are the pheromonal PNs in the anterior-most region of the LH. This has been noted by the field, in terms of both morphology and connectivity (Ruta et al., 2010; Kohl et al., 2013, Frechter et al., 2019; Chakraborty and Sachse, 2021; Bates et al. 2020; Bates et al. 2020; Jefferis et al. 2007).

However, they do not demonstrate that they act as labeled lines. To make a statement about labeled lines is to make a statement about connectivity. It can be guessed by using morphology when connection data is absent. However, the connection analysis can now be done using data from the hemibrain connectome (Scheffer et al. 2020; Schlegel et al. 2021). Labeled line strategies may exist for some odour channels in *Drosophila* – in particular pheromonal channels such as DA1 (Kohl et al. 2013) or the aversive channels the authors note – but it is not possible to determine this using morphology alone. A preprint that examines DA2 – an aversive PN thought to be part of a labeled line (Stensmyr et al., 2012) – actually saw that in the connectome its targets experienced a lot of convergence from different PNs, only a few preserving a possible labeled line (Huoviala et al. 2020). Labeled lines are probably the exception, not the rule, and with DA2 a strong labeled line organisation seems to become a highly distributed representation at the lateral horn stage. If the authors could compare their work on morphology, to the reality in terms of connectivity, they would be better able to support their ideas and show that the features they quantify correlate with circuit structure.

The authors say that "our analysis for the second-order neurons inside the *Drosophila* olfactory system can be translated to the brain of different organisms including the central nervous system (CNS) of humans". However, I do not think this is strictly true. In *Drosophila*, neurons of the same cell type, the author's 'homotypes', are near isomorphic and occupy a similar space. Therefore, the authors can use, as they did, a metric defined in Euclidean space to compare the neurons, without spatially transforming the cells at all. In mammals, duplicate types can appear across, say, cortical columns and even in the olfactory system be very spatially segregated, since there is not just one glomerulus for each olfactory receptor, as in insects. Therefore, the spatial analysis that would need to be done in mammals would be different.

It is commendable that the authors have made their analysis in python available on GitHub (https://github.com/kirichoi/DrosophilaOlfaction), as well as preprinting their work (https://www.biorxiv.org/content/10.1101/2022.02.23.481655v1). The authors pre-process the skeleton data well, removing registration artefacts. The consensus lateral horn volume (Ito et al. 2014) is somewhat arbitrarily defined as the uPN terminus region, though the MB Calyx is more strongly defined by glial sheathing. The authors wisely do not use this volume but re-define their neuropils – perhaps more accurately regions of interest – by using density estimates built from the neurons' 3D points. It would have been nice to see, in a supplementary figure, how these ROIs differ from the Ito et al. 2014 standard neuropils – since when they use neuropil terms in the text, an informed reader will assume they used the Ito standards. I like the visual segmentation process description in Figure S5.

I think the paper does make many quantitative points on PN morphology well, and could either become a shorter manuscript that shows this information concisely, or a more involved one on 'labelled lines' and the validity of this idea for the olfactory system, where 2nd-order neurons meet 3rd order ones. In the first case, adding thermo sensory PNs, GABAergic uPNs, and mPNs to their analysis could make it more interesting, and produce some new and germane insights for their sub-regional analyses (see below). In the second case, the paper would need to include work that compares across different PN classes and EM datasets, including looking at connectivity in the hemibrain, to make its core claims on labeled lines.

In general, in the present work, the authors do not provide enough biological background information on the olfactory system and what is known about PNs already, the regions they innervate for a naive reader to understand their results, and why their results might be interesting. This could be easily fixed with a little re-writing. The use of mathematical notation within the main text is heavier than needs be and hampers the readers' understanding of what the authors are doing and intend to show. Plainer writing would help. The authors need to communicate their core findings, and their context in the field, more clearly.

Glomeruli identification through the literature can be tricky. There has been some confusion specifically about the identities of VM6 and VC5. The Zheng et al. 2018 paper had a few errors, later rectified by Bates and Schlegel et al. 2020 and then again by Schlegel and Bates et al. 2021. Given the confusion, no one can be faulted for mistakes here, but authors should make sure their labels are the same as in Schlegel and Bates et al. 2021 (that process was a multi-lab debate). I think they used the outdated Zheng et al. 2018 labels.

Suggestions

I have some suggestions for increasing interest in this work, that I humbly submit to the authors. The authors discuss 51 glomeruli of the antennal lobe. Of the olfactory glomeruli, there are 52, but there are actually 58 glomeruli in total in the antennal lobe (e.g. VP1-5), including the thermo-hygrosensory ones. Other work (Schlegel et al. 2021; Bates et al. 2020) has found that some of the most striking differences can be found between olfactory and thermo-sensory glomeruli, as well as some curious associations across the two modalities. Since intellectually this system is very similar – RNs contact PNs that then reach the LH and the zone just ventral to it – but not quite parallel – arbours intermingle with olfactory ones in the ventral LH, and antennal lobe local neurons cross compute between combinations across all 58 glomeruli – I strongly think including these neurons in the author's analysis would increase the interest in this work.

There are a total of 347 PNs in the FAFB dataset that project from the right-side antennal lobe. This is because there are 283 multi-glomerular PNs (mPNs). Bates et al. 2020 also make 58 uPNs from the left side of the brain available. All of the data is open and can be gotten by the authors here: https://fafb.catmaid.virtualflybrain.org/. Similar to the above, I think that including the mPNs as a comparison point, would increase interest in this work.

PNs could also have been clustered by their odour response profiles using data from PN response (Badel et al. 2016) or ORN response (Münch and Galizia 2016), rather than porting labels from Bates et al. 2020 / Mansuorian et al. 2015. The authors might consider whether this could add meaningfully to their analysis.

In addition, as noted in the public review, I think comparing against or using NBLAST could be informative. The authors could run NBLAST on their processed neuron skeletons and discover whether the results differ much from what they have in hand right now. NBLAST is now implemented in the navis python library: https://navis.readthedocs.io/en/latest/source/tutorials/nblast.html.

Lastly, a more out-there suggestion: PN morphology analysis using a skeleton representation has been common in the field. However, volumetric analyses based on neuronal meshes – now available for these neurons through the hemibrain and flywire projects – is almost non-existent. How do the volumes for neurons vary across types, clusters, etc? Using this information could add some simple points, to support the authors in their quantitative description of these neurons.

*Reviewer #3 (Recommendations for the authors):*

This work provides a quantitative evaluation of the spatial organization pattern of olfactory projection neurons (PN) in AL, CA, and LH based on the inter-PN distances using FAFB EM dataset. This NBLAST-based method does provide a simple way to cluster neurons with similar projection patterns and even predict underlying signal processing rules. However, the reliability of the clustering method needs to be improved since the method only got 39 clusters out of 51 well-segregated AL glomeruli serving as the ground truth. This result undermines the conclusions in the higher-brain centers which have a more complex organization. Moreover, the authors defined the neuropils by rotating the neurons along specific axes and then segmenting the dense innervation parts, which may hinder the accuracy of the boundary when the surface is convoluted and does not reveal inner subdomains. Nevertheless, serving as the first step to tackling complex connectomic data, the method is easy and potentially useful.

1. The current study analyzed only 111 uniglomerular PNs instead of the latest-released 164 uniglomerular PNs (Bates et al., 2020). To make the work more valuable, the authors should apply their analysis to the latest dataset. In addition, since the analysis was done from the EM data of a single fly, whether the preferential spatial distribution of PNs and the clustering are consistent in different individuals is unknown. It will be informative and more persuasive if a similar spatial distribution pattern can be observed in another fly EM dataset (Scheffer et al., 2020).

2. The segmentation of neuronal innervation in AL, Calyx, LH is achieved by rotating the neurons along specific axes and then identifying the dense innervation parts as the three neuropils (Figure S5). The methodology is convenient to define the neuropils but will hinder the accuracy of the boundary segmentation when the surface is convoluted (i.e. MB calyx) and does not reveal inner subdomains (i.e. AL glomeruli). As the original EM dataset has already offered neuropil surface point coordinates which can be downloaded from Catmaid website (https://catmaid-fafb.virtualflybrain.org/), attributing the innervation points directly to the defined neuropils should be more accurate.

3. Theoretically, the natural segregation of glomerular structures in AL would serve as the ground truth to test the reliability of the clustering method. Yet, the method only got 39 clusters out of 51 well-segregated glomeruli. This discrepancy undermines the effectiveness of the clustering strategy. The result of silhouette coefficient analysis also suggests that the coefficients are similar for clusters ranging from 30 to 50 in AL (the coefficients are very close almost reaching a plateau) (Figure S6). The author should justify why choosing the number of PN innervation clusters in AL as 39 and provide evidence that it is optimized to find the hidden pattern. Most importantly, if the method fails to reveal the remaining 11 AL glomeruli that can be visually distinguished, it is difficult to see how it can reveal the hidden pattern in MB calyx which is much less well defined.

4. The neuronal distance estimation algorithm may encounter a problem when projection neurons have very different innervation ranges and the total length of the innervation branches. For example, if there is a projection neuron, PNa, only innervates in the entry of Calyx where most PNs pass by, the distance between PNa and other PNs will be very small due to the algorithm only selects the shorter skeleton to evaluate the distance (see method and the symmetric matrices in Figure 3). Thus, the result may not faithfully reflect the fiber distance when two PNs have a drastic difference.

5. The authors estimate λ value which represents the ratio of the mean of intra-homotypic neuron distance and the mean of inter-homotypic neuron distance. (Figure 4, Figure 5, Figure 6, Figure S3). It is a creative way that gives us insight into how PNs overlap with each other in distinct neuropils. However, the volume and the aspect ratio of the three neuropils are different. AL occupies a much larger area compared to calyx and LH. Therefore, the current λ value may simply reflect the spatial distance presented in the neuropil but not the actual degree of overlapping (e.g. in two non-overlapping pairs, the one with long distance will have a low λ value while both pairs have no overlapping at all). To make the λ value reflect more to the degree of overlapping, they might need to normalize it based on the volume, compare the λ values between original data with the data after shuffling PNs into different clusters or take the innervation density of fibers in specific neuropils into account.

6. The authors annotate the reaction profile for a specific glomerulus using a dumb variable to do statistics. The result suggests that pheromone PNs will be clustered together in LH (into 2 clusters) and Calyx. Food-related PNs and aversive PNs also be clustered into different groups in LH but not in Calyx (Figure 7, Table S1). The author should elaborate and discuss more in terms of biology. For example, do food-related odor signals and aversive odor signals converge in MB calyx and diverge in LH? What biological properties may be implicated in the organization?

7. Although the authors have offered the python scripts for researchers to reproduce their results, a comprehensive spreadsheet that contains all the distance results along with functional annotation will greatly improve the accessibility of the analyzed results.

8. The authors should compare their cluster results with Lin et al. 2007 and Bates et al. 2020 and discuss the biological implication.

References:

Bates, A. S., Schlegel, P., Roberts, R. J., Drummond, N., Tamimi, I. F., Turnbull, R., … and Jefferis, G. S. (2020). Complete connectomic reconstruction of olfactory projection neurons in the fly brain. Current Biology, 30(16), 3183-3199.

Jefferis, G. S., Potter, C. J., Chan, A. M., Marin, E. C., Rohlfing, T., Maurer Jr, C. R., and Luo, L. (2007). Comprehensive maps of *Drosophila* higher olfactory centers: spatially segregated fruit and pheromone representation. Cell, 128(6), 1187-1203.

Lin, H. H., Lai, J. S. Y., Chin, A. L., Chen, Y. C., and Chiang, A. S. (2007). A map of olfactory representation in the *Drosophila* mushroom body. Cell, 128(6), 1205-1217.

Nishino, H., Iwasaki, M., Paoli, M., Kamimura, I., Yoritsune, A., and Mizunami, M. (2018). Spatial receptive fields for odor localization. Current biology, 28(4), 600-608.

Zheng, Z., Li, F., Fisher, C., Ali, I. J., Sharifi, N., Calle-Schuler, S., … and Bock, D. D. (2020). Structured sampling of olfactory input by the fly mushroom body. BioRxiv.

---

## [Author Response]

While appreciating the authors' work presented in this manuscript there were two major concerns that came up during the consultations:1. Given that two EM datasets exist for the fly brain, determining the generalisability of what the authors describe should have been done.2. Making claims about labelled line representation of odours or not using morphometric data alone is not justified when connectivity data exists for flies.Based on this, and other comments, the essential revisions that were discussed were the following:The writing: The text needs to be made more accessible to the reader, perhaps by providing more intuitive explanations, biological context, and emphasise their findings more than they currently do (which emphasises methods used more). We would also like to see their analyses in the context of what's been done before.

In our revision, we have made a substantial update to the writing to make our manuscript more accessible to the readers. First, we now provide much more intuitive explanations for the concepts discussed in the manuscript. We have tried to move away from mathematical notations if possible and supplemented our descriptions with intuitive illustrations whenever we thought it was necessary. Therefore, most of the equations and detailed statistics are moved to either the Methods or the appendices, replaced by simple and intuitive descriptions. For example, one of the common comments across the reviewers was that we did not explain what our inter-PN distance meant. Now, the Result section starts with a simple explanation of what we meant by inter-PN distance, along with a demonstrative illustration (Figure 2—figure supplement 1A) and detailed formulations (in the Methods section). On the same note, our figures have been modified to make them more comprehensible, either by adding visual aids, increasing the size of figures and text, or providing a more explicit caption.

The general tone of the manuscript has been shifted as well, moving away from the methodology and statistics to explanations based on biological implications and comparisons to previous literature. Therefore, a substantial portion of the main text has been moved to either the Methods or the appendices. Our Discussion section has been significantly improved, discussing our results by comparing them with Bates et al. (2020), Lin et al. (2007), Strutz et al. (2014), Zheng et al. (2020), etc., covering almost all the references raised by the reviewers. Apart from providing a richer background in the context of what has been done before, we also emphasized some of our novel findings and their biological implications of them.

Reproducibility: Since the EM data in this analysis is based on a single fly, we recommend testing whether the authors' findings will be consistent across individuals. So, could the authors test their analysis on the hemibrain alongside the extant FAFB?

First, we would like to emphasize that our manuscript is now using the latest FAFB dataset by Bates et al., 2020, to answer other requests by the reviewers. Therefore, all our figures in the main text are replaced with the new results based on the updated FAFB dataset.

To test the generalizability of our result, we have performed a reproducibility study using the suggested hemibrain dataset (Scheffer et al., 2020). We have reproduced several key figures in the main text, which were based on the FAFB dataset, using the hemibrain dataset instead, and included them in Figure 13. Comparing the results based on the hemibrain dataset with our new results from the FAFB dataset, we found that the overall results from the two datasets are largely the same and the two datasets are interchangeable in supporting our claims. We also discussed the generality of the results from our analysis by commenting on the similarity and differences between FAFB and hemibrain datasets. Additional details are available in the Methods section and the Appendix 3. On our Github repository, we publicly shared the script we have used to query the neurons from the hemibrain dataset, and a copy of our analysis code tailored for the hemibrain dataset.

Number of uPNs used: The authors use 111 uPNs for their analysis whereas the number of uPNs is 164. (This might be why they obtain only 31 glomeruli). It's not clear why the rest were excluded. Could the authors conduct their analyses by including all cholinergic uPs (including those involved in hygro and thermo sensation)? (We assume the authors have omitted GABAergic neurons from their analyses because they do not innervate the MB? Including it would provide important biological insight, but if not, their reasoning needs to be stated.)

We would like to reiterate our reasoning behind why we chose a subset of uPNs in our original analysis. One of our primary goals in this manuscript was to analyze how the spatial characteristics of homotypic uPNs change as the uPNs innervate different neuropils. Because we wanted to compare the spatial characteristics of uPN innervation at each neuropil, we had to consider uPNs that innervate all three neuropils. Therefore, many GABAergic uPNs ended up not being included in our original analysis. We agree that the description and justification regarding our criterion might have been unclear in the original submission.

That said, the analysis of all uPNs would provide a more comprehensive view of the spatial organization of uPNs, making our study more interesting and satisfactory. Therefore, we have made the following changes to address this issue.

First, as stated before, we have used the latest FAFB dataset (Bates et al., 2020) for analysis, which contains a total of 164 uPNs composing 58 glomeruli. We have re-done all our calculations using the latest FAFB dataset and studied a set of 135 uPNs that innervate all three neuropils, including those involved in hygro and thermo sensation. Based on this result, we have updated the manuscript and the figures, pointing out several newly found properties.

Second, we performed a separate analysis on the ~30 uPNs that were originally left out based on our selection criterion (that they do not innervate all three neuropils). Most of these uPNs are GABAergic, following mlALT instead of mALT. These neurons are still crucial for the olfactory processing in *Drosophila*, but were largely incompatible with our analysis (e.g., we can’t calculate and compare λ across the neuropils). Therefore, we conducted a few compatible analyses that examine (1) the spatial difference between uPNs innervating all three neuropils and uPNs whose innervation is specific to a particular neuropil, and (2) neurotransmitter-based characteristics. The manuscript has been updated accordingly.

Hierarchical clustering: We are concerned that the inferences rely on a few clusterings that don't appear to be very robust. This is usually okay, but a lot of the paper is built around this. So, could the authors reassess their hierarchical clustering parameters and the normalisation for the λ value and state them explicitly for the reader?

We understand the concern regarding the robustness of the clustering result. The output of hierarchical clustering is indeed affected by the cut height and the specific tree-cutting method. However, we would like to point out that this is not a cause of serious concern, as the hierarchical relationships are generated purely from the distance metric we used and are fixed regardless of which tree-cutting method we choose. For example, when looking at the dendrogram for AL (Figure 2—figure supplement 3A), one might notice that the well-segregated glomeruli are expressed through homotypic uPNs under neighboring branches. Therefore, while the final tree-cutting step may affect the number of clusters, increasing the number of clusters will simply break down a large cluster into smaller ones. In fact, if our argument can still be explained by smaller clusters (or a large number of clusters), our argument may be considered better supported.

Nonetheless, we tested a total of four different tree-cutting methods/criteria: elbow method, gap statistics, maximum average silhouette coefficient, and the dynamic hybrid cut tree method (Langfelder et al., 2008) which has been successfully utilized for morphometric classification of mouse V1 neurons by Gouwens et al. (2019). We noticed that using the maximum average silhouette coefficient resulted in the most accurate cluster number in AL (N=54), but it showed a significant discrepancy in cluster number in MB calyx and LH when compared to the other three methods (see Table 1).

After an internal discussion, we have decided to switch our tree-cutting method to the dynamic hybrid cut tree method for the following reasons:

i) Choosing a tree-cutting method that returns a cluster number close to the number of different odor types (which is 10) best serves our goal.

ii) Since we are interested in the clusters of uPN innervations to MB calyx and LH (as the correct labels in AL are already given), it seemed reasonable to choose a method out of the three methods that produced comparable cluster numbers in MB calyx and LH.

iii) The dynamic hybrid cut tree method was proven to give satisfactory results and is methodologically systematic based on our previous experiences and previous literature.

iv) We wanted to test our hypothesis under a harsher condition.

Switching the tree-cutting method resulted in 10 clusters in MB calyx and 11 clusters in LH. We report that our argument still holds despite the higher cluster number. The trend of the uPNs under the same homotype falling into the same cluster became much more apparent (Figures 3, 4, and 8). Our various statistical tests also returned results in line with our findings. Additionally, we have updated our Method section to provide in-depth descriptions of how the hierarchical clustering was done, along with our reasoning behind the choices we made, to help the readers understand our clustering procedure without any ambiguity.

Concerning the issue with λ normalization, we would like to iterate that we devised the λ to be a scale-free ratio that quantifies the relative spatial innervation per homotype. This means that we intended not to scale our λ based on the volume. For example, let us consider a homotype composed of several uPNs with similar d_*intra*_ values for both MB calyx and AL. Indeed, the volume of MB calyx is smaller than that of AL, so the d_*inter*_ will be generally larger in MB calyx, leading to a higher λ value, which is what we intended. Our logic behind this is that in AL, despite all the additional space that the uPNs technically could have innervated, the uPNs ended up localized into a bundle with a small d*_intra_*. In MB calyx, the same uPNs had less space to innervate, to begin with, so the less ‘packing’ observed in MB calyx may not be as novel. We wanted our λ to differentiate the two, which is the reason why our λ is defined as such.

However, it seems like the term ‘overlapping’ is a bit elusive and can be defined in various ways, which we believe is why this issue has been brought up by the reviewers. We feel that our explanation and choice of words (overlapping) were not sufficiently clear. After a lengthy discussion, we have decided to keep the terminology but provide a much more comprehensive explanation of what we mean by the ‘degree of overlapping’ to make sure that the readers won’t get confused.

Labelled line hypothesis: The claims made in this study about the labelled line hypothesis cannot be inferred from morphometrics alone. For this, the authors would need to compare these findings with connectivity. This could even be done at a small scale to understand whether their morphological groups also have more specific or segregated connectivity within the LH / AB.(A possible pipeline for this could be: Use neuprint-python to pull connection data for each chosen PN and subset it to the select brain regions. They could then perform hierarchical clustering on that, and devise groupings that can be compared to the morphological grouping visually using a tanglegram. They might then compare the two using Baker's Γ Index or similar.)

Following the suggestion, we decided to use the connectivity data to conduct an additional small-scale study to check how far the labeled-line principle holds. For a comprehensive connectivity dataset between PNs and higher olfactory neurons such as KCs and LHNs, we utilized the hemibrain dataset. Here, we tried to address the following questions: (1) Are there any KCs and LHNs that carry a specific type of information? If so, how prevalent are they? (2) Are there any insights we can gain from comparing the clustering outputs using spatial proximity (d*_αβ_*) and connectivity?

To answer these questions, we have conducted three additional analyses.

First, we collected a number of KCs and LHNs that synapse only with a single homotype (Figure 10). These ‘homotype-specific’ connections (Nx,spξ), defined as the number of third-order neurons that only synapses with a specific homotype but not with the others (see Figure 9 and Methods for more information), are much more prevalent in LHNs compared to KCs. Certain homotypes (e.g., DA1) have an especially high number of LHNs that only connect to the given homotype. The ‘homotype-specific’ neurons functionally carry a single type of information, thereby may be considered as an extension of the labeled-line strategy.

Second, we collected LHNs connected to a particular homotype and checked which other homotypes these LHNs are also synapsing (thereby analyzing the scope of signal integration happening at LH – see Figure 9 and Methods for more information). Based on this analysis, we found the homotype-specific labeled line predominate until the signal reaches the higher olfactory centers, some of which then transition into odor-specific channels where either a broad or a narrow integration occurs. In MB calyx, no such trend is observed, further supporting the previous literature on randomized connections. Even though we observed a strong per homotype bundling tendency at MB calyx, the high degree of overlapping (denoted by λ) seems to have a bigger impact on connectivity.

Third, we performed a connectivity-based hierarchical clustering and compared the result against our spatial proximity-based clustering result. Tanglegrams are plotted to compare the dendrograms generated from the spatial proximity (d*_αβ_*) and from the connectivity (d*_cos_*) (see Figure 12). To check the relationship between the spatial proximity-based clustering and connectivity-based clustering results, we supplied Baker’s γ index (the ordinal relation), entanglement (transferability), and cophenetic distance correlation (the correlation between two tree structures). We report that the spatially well clustered uPNs at MB calyx do not precisely translate to structured connectivity patterns, consistent with the notion of randomized PN-KC connections, while in LH, spatial and organizational characteristics of uPNs are well-translated to connectivity to LHNs.

In addition to this, please read the other (not essential) recommendations and if any of them are feasible to respond to, please do so.Reviewer #1 (Recommendations for the authors):The olfactory PNs have been studied using numerous approaches. In this study, the authors use a parameter called 'inter PN distance' to assess the relationship between all the uniglomerular PNs in the hemibrain connectome in the AL, MB, and LH. As odour information from the sensory neurons is brought into the AL via OSNs classes that innervate individual glomeruli, PNs that innervate the same glomeruli are considered homotypic. Using the inter PN distance and downstream statistical analyses, the authors infer that odour information is discretely represented in the AL and less so in the LH; however, in the MB, there is considerable overlap in odour representation.The manuscript is well written and presented.- PN information representation across the AL, MB, and LH has been explored before. While the authors do mention this, it would be useful if they discussed what each one of those approaches inferred about olfactory representation, and why they felt their approach was better. In the same vein, it would be nice if they finally concluded by comparing their findings with the others.

We thank the reviewer for the suggestion. We substantially revised our paper, not only discussing how our results are in-line or different from the results by Bates et al. (2020), Lin et al. (2007), Strutz et al. (2014), etc., but also including new topics ranging from spatial innervation patterns to connectivity with third-order neurons. Also, our manuscript now better emphasizes some of the unique findings we made, thereby demonstrating the advantages of our approach.

- The manuscript is statistically dense. It will help readers if the authors provided some intuitive explanations along the way – for example, for inter-PN distance, among others.

Following the reviewer’s comment, we have updated our manuscript to make it more accessible to the readers. For example, we have added a simplified explanation of the distance metric d*_αβ_* to the main text before we discuss the spatial proximity-based clusters, along with a schematic depicting the step-by-step process of calculating d*_αβ_* (see Figure 2—figure supplement 1A). Most of the equations and detailed discussions of statistical test of independence are moved to either the Methods or the appendices, replaced by simple and intuitive descriptions. Whenever we feel our explanation might not be easy to follow, we added illustrations to supplement the text.

- Many of the legends are not very explanatory. Could the authors please elaborate on them? It would also help to have pointers in the figures to focus the attention of the reader. For example, the legend for S2 can be expanded. I can't tell which homotypic PNs are merging into a single cluster. There are similar issues in other figures as well – could the authors please add visual aids such as arrows in the figures?

Following the reviewer’s comment, we have updated our figures and captions to make them easier to read and understand. The size of the figures and texts is increased in general, and we added visual aids whenever we discuss a specific subset of a figure.

Reviewer #2 (Recommendations for the authors):Choi et al. competently quantify known aspects of excitatory, uniglomerular projection neuron (uPN) morphology, in order to demonstrate how neurons of this well-studied cell class differ in the three main brain regions in which they arborise. Classically, these three regions support three different functions: siloed olfactory channel processing (antennal lobe), ethological odour classification (lateral horn), and associative memory (mushroom body). They find that uPNs of the same cell type are morphologically similar in each region, but the nature of that similarity, and their similarity with other uPN cell types, differ as appropriate for the function of the neuron in that region. While not novel, their quantitive description contributes to the field. Their 'major' conclusion on labeled line processing is, however, insufficiently substantiated given that the olfactory system of the fly is post-connectomic and this question is best taken up with connectionist analysis. They also only use a single dataset from a single fly (though a second is available) and do not explore how robust their results are to other clustering approaches/decisions, making it unclear how general their description is.The strength of this work is that it is able to use quantitative methods to demonstrate features of *Drosophila* olfactory projection neuron (PN) anatomy. To date, these features have been known and used in classification work on the PNs (e.g. Tanaka et al. 2012; Bates et al. 2020; Zheng et al. 2018; Bates et al. 2020; Jefferis et al. 2007). They have also been shown through (PN->target) connection analyses (Eichler et al. 2017; Schlegel et al. 2021). The findings are therefore largely not novel but are a good quantitative description and a nice synthesis in one place. This work would be cited in the field for the following point: within a cell type, PNs tile their input glomerulus with their dendrites, intermingle their axons with other PN cell types in the mushroom body, and overlap with each other, and to a lesser degree other PN cell types, in the lateral horn. The authors' other main conclusions would take more to support.In this work, the authors first (A) choose the neurons they will examine, and then perform two main analyses, (B) pair-wise morphology similarity analysis within three major projection regions for these neurons and then (C) an analysis of 'homotypic' PN cell types and how similarity within and between the cell type varies between the three projection areas. They then (D) conclude that their results are indicative of 'form determining function' and 'labelled line' encoding at play in the *Drosophila* olfactory system, where 2nd and 3rd order neurons meet.On A: The authors use data from the FAFB connectome. These are manually reconstructed neuronal skeletons, built using CATMAID. The dataset at present offers a sparse connectome, though a denser one using the same EM data is currently being built by the Flywire project (Dorkenwald et al. 2022), open to anyone for contributions. A dense connectome for the olfactory system already exists in a separate EM dataset, thanks to the hemibrain project (Scheffer et al. 2020). One of the limitations of the present work is that it assumes that the data it uses from FAFB (PNs on a single hemisphere of a single fly) are representative of all flies, and does not discuss this assumption (which is a fair enough assumption given how morphologically stereotyped the system appears to be (Jenett et al. 2012; Schlegel et al. 2021; Jeanne et al. 2018) but should be discussed; the degree of connection stereotypy is still unsettled, and there could be variation in how well neurons of a homotype may fasciculate across brains). The authors could better support their statements by re-running their analysis on the hemibrain data and comparing it to their CATMAID FAFB data. It would greatly strengthen their study to show that the patterns they observe hold across two EM data sets. (Light-level co-registered datasets also exist via the flycircuit project (Shih et al. 2015; Chiang et al. 2011; Costa et al. 2016), though registration offsets likely limit the effectiveness of the authors' present analyses – the authors could consider it).

In the revised manuscript, we have used the latest FAFB dataset for the main analysis (Bates et al., 2020).

All our figures in the main text are replaced with the new results based on the updated FAFB dataset. Additionally, we have performed a reproducibility study using the suggested hemibrain dataset (Scheffer et al., 2020) and validated our results by repeating the calculation. When we compared the results based on the hemibrain dataset with our new results from the FAFB dataset, we found that the overall results from the two datasets are largely the same and the two datasets are interchangeable in supporting our claims. We have reproduced several key figures in the main text, which were based on the FAFB dataset, using the hemibrain dataset instead, and included them in Figure 13. We also discussed the generality of the results from our analysis by commenting on the similarity and differences between FAFB and hemibrain datasets. On our Github repository, we publicly shared the script we have used to query the neurons from the hemibrain dataset and a copy of our analysis code tailored for the hemibrain dataset.

The authors' analysis focuses on 111 PNs. This is close to the number of uniglomerular projection neurons (uPNs) in the right hemisphere. The actual count from Zheng et al. is 114. The updated count for the same dataset is 164 (Bates et al. 2020). The decision to not use a full complement of uPNs is not adequately justified. (The authors state that they dropped three neurons that are not PNs, including the APL – those neurons are not in the numbers I give above, and the Zheng et al. archiver must have given them the wrong label.) In addition, uPNs have two functionally important sub-divisions – there are excitatory uPNs (cholinergic) and inhibitory uPNs (GABAergic). This distinction, and how the two could differ in terms of which clusters they fall into, is significant because it might reveal how different odour channels can synergise with, or antagonise one another. I think the GABAergic uPNs are left out because they have no MB arbors, but the existence of these different PN groups and the choice to exclude some biological classes are not made clear in the main text.Indeed, 6 neurons are said to be dropped because 'they did not project to any of the neuropils'. This does not seem like a great reason for dropping these cells completely from the paper. If the point the authors wish to make is that there are labeled lines among the olfactory PNs, surely the ones that stray off into other brain regions are the most evidential! While they do not say it directly, the authors are looking at cholinergic, olfactory uniglomerular PNs and excluding other PN classes – which is fine, but could be made clear. (There are a total of 347 PNs covering all classes, on the right hemisphere in FAFB).

We thank the reviewer for pointing out this issue. As the reviewer has noted, both cholinergic and GABAergic neurons play important role in olfactory signaling. Our intention was not to ignore GABAergic PNs under some arbitrary conditions. One of our primary goals in this manuscript was to analyze how the spatial characteristics of homotypic uPNs change as the uPNs innervate different neuropils. This made us only consider the uPNs that innervate all three neuropils, which is why many GABAergic uPNs ended up not being included in our original analysis. We agree with the reviewer that the description and justification regarding our criterion might have been unclear. We have updated our manuscript to better reflect the reasoning behind our choice.

That said, we noticed that there are several additional uPNs available from the latest FAFB dataset (Bates et al., 2020) that met our existing criterion, including those involved in hygro and thermo sensation. We considered that analyzing the additional uPNs will provide a more comprehensive view of the spatial organization of uPNs. We have re-done our calculations using the latest FAFB dataset and studied a set of 135 uPNs that innervate all three neuropils. We have updated the manuscript and the figures accordingly, pointing out several newly found properties.

Many GABAergic uPNs that do not innervate all three neuropils (there were 28 of them in the latest FAFB dataset) are still crucial for the olfactory processing in *Drosophila*. These neurons are largely incompatible with our analysis (e.g., we can’t calculate and compare λ across the neuropils), but we conducted a few compatible analyses that examine (1) the difference between uPNs innervating all three neuropils and uPNs whose innervation is specific to a particular neuropil, and (2) neurotransmitter-based characteristics.

There was a total of 137 uPNs innervating MB calyx, indicating our analysis only left two uPNs which constituted VP3. When the hierarchical clustering was performed on the entire 137 uPNs in MB calyx we ended up with an almost identical clustering output. Two missing uPNs were grouped into the clusters C_4_^MB^ and C_6_^MB^, along with other hygro/thermo-sensing homotypes. On the other hand, there was a total of 162 uPNs innervating LH, analyzing the complete uPN innervation in LH more interesting. The addition of 27 uPNs constituting 15 homotypes innervating LH created four new clusters when the hierarchical clustering was performed (see Figure 4—figure supplement 1). The additional 27 uPNs changed the content of the individual clusters; that is, the tree-cutting algorithm broke down a few clusters that became larger due to the additional uPNs. Furthermore, when we calculated d̅_intra_, d̅_inter_, and λ for the 15 homotypes that included the 27 uPNs, we find that the LH d̅_intra_ values increased after adding back 27 uPNs (see Figure 5—figure supplement 2). This suggests that the previously removed uPNs, most of which follow mlALT, are significantly different in terms of spatial and organizational characteristics and thus should be analyzed separately.

Out of 27 additional uPNs in LH, 21 followed mlALT, 5 followed trans-lALT, and 1 followed mALT. Figure 4—figure supplement 2 illustrates how these 27 uPNs spatially innervate LH and demonstrate why d̅_intra_ values were increased, as they are indeed poorly bundled with the existing uPNs under the same homotype. These 27 uPNs are mostly GABAergic: they are composed of 21 GABAergic, 1 cholinergic, and 4 unknown neurotransmitter types, covering 84 % of GABAergic uPNs available in the FAFB dataset. These uPNs innervate LH drastically differently from other uPNs in the same homotype (see homotypes such as DA1, DC4, DL2d, DL2v, DP1l, VA1d, VA1v, VL2a, VL2p, and VP5 in Figure 4—figure supplement 2). Morphologically, inhibitory GABAergic neurons are often considered to be ‘smooth’ and aspiny (Douglas et al., 1989; Bopp et al., 2014; Gouwens et al., 2019), which are discernible from Figure 4—figure supplement 2.

On B: The authors use an all-by-all pair-wise morphology similarity metric to construct a score matrix and then perform hierarchical clustering. The metric is not adequately explained. In reviewing the methods, the metric appears, by the author's own note, to be similar to NBLAST (Costa et al. 2016). There are many different and valid reasons for using different metrics, but it is not clear to me why the authors did not use NBLAST, and how their own method differs. Perhaps results from both would be similar, and it does not really matter – or maybe they found that their own metric is more performant in this situation? The authors, I should be clear, do not claim to have invented a 'better' metric – but some explanation of their choice is warranted. The terms in the author's equation are not adequately explained, but it seems to be a distance score calculated between all points between two neurons in Euclidean space, that decays exponentially with distance, which is common practice (Schlegel et al. 2016; Strutz et al. 2014; Kohl et al. 2013; Masse et al. 2012). Some explanation to that effect should appear in the main text. (I opine that since there are now many studies examining PN morphology with similar metrics, it would be easier if the same algorithms were applied when there is not a good reason to do otherwise, so that results can be directly compared, NBLAST is currently the most popular choice).

First, we have added a simplified explanation of the distance metric d*_αβ_* to the main text before discussing the clusters, along with an illustration depicting the step-by-step process of calculating d*_αβ_* (see Figure 2—figure supplement 1A). Simply put, the distance metric d*_αβ_* only considers the pairwise distance but not the dot product term (which measures the similarity of two neuronal morphologies) used for the NBLAST score. Therefore, d*_αβ_* is computationally comparable to the NBLAST score, but it only measures the spatial proximity between two neurons. The reason why we have decided to use this metric is that we are predominantly interested in the spatial proximity (or co-location) between two uPN innervations but not the structural similarity between them, which the NBLAST score accounts for (a point which is also noted by Zheng et al., 2018). We agree that the NBLAST score is the community standard in terms of quantifying the similarity between two neurons, but we believe the distance metric d*_αβ_* is conceptually more physical and adequate for the primary aim of our study, which is to analyze the spatial and organizational characteristics of uPN innervation in each neuropil. For example, when we calculate the normalized NBLAST distance between uPN innervations and compare it against d*_αβ_*, the two distances are correlated but with significant dispersion, indicating that these two metrics are not exactly the same. There are some cases that inter-PN organization measured by d*_αβ_* can be distorted if NBAST score is used instead (see Figure 2—figure supplement 1B).

In examining their scores, the authors do not give their clustering method, just stating 'a method of hierarchical clustering was used'. Drawing conclusions from hierarchical clustering is very sensitive to the cut height used. It would be nice if the authors showed the result of their clustering as, for example, a dendrogram with the cut height they used indicated by a horizontal line in the main sequence (this is currently S2). Their cut height was determined by a silhouette method for determining optimal clusters. Other common, valid methods include the elbow and gap statistic methods – perhaps in supplement, the authors could give or summarise how the result would have been using those, or in the methods, explain why they could not? How robust is their analysis to different clustering methods, different numbers of clusters, etc? The details of an unsupervised clustering analysis are often skimmed over in papers because we trust the authors were able to tweak things to bring their biological conclusions to the foreground, and that is fine. However, since this paper is all about the results of the authors' clustering analysis, the detail and how robust their analysis is, are very important. Their analysis yields 39 clusters in AL, 3 clusters in MB calyx, and 4 clusters in LH, and there is a nice visualisation in Figure S6 that could use some vertical lines to show the shown cluster numbers. However, those 4 LH clusters could easily be broken down further into smaller groups that look meaningful to a neuroanatomist. The authors use a Pearson's Chi-squared test to observe that their clusters are statistically different from a random sample of PNs, which is great. The authors could re-word this segment though, with simpler language, to let the reader qualitatively know what they are doing and why.

We thank the reviewer for bringing up this topic. First, we admit that the hierarchical clustering was too briefly described. We have updated our Method section to provide in-depth descriptions of how the linkage was constructed and trees are cut. Additionally, we would like to point out that the different colored branches in our dendrograms (Figure 2—figure supplement 3) correspond to different clusters (in Figures 3 and 4).

We also understand the reviewer’s concern about the robustness of our clustering output. However, we do not think this is a cause of serious concern due to the nature of our clustering method. The output of hierarchical clustering is indeed affected by the cut height and the specific tree-cutting method. However, the hierarchical relationships are generated purely from the distance metric we defined, and the tree structure is fixed regardless of which method we choose to cut the leaves. Often, there is no ‘correct’ way to determine the number of clusters or cut height; different methods may offer slightly different suggestions. While we originally reported 39 clusters for AL, 3 clusters for MB calyx, and 4 clusters for LH, we do not consider these cluster numbers to be the ground truth with some biological implications, but rather a result of a particular clustering protocol and parameters we chose. This may be of considerable concern for clustering methods like K-means clustering, but for hierarchical clustering with a fixed distance matrix, the tree structure is retained regardless of the cut height. While the cut height changes the content of clusters, increasing the number of clusters will simply break down a large cluster into smaller ones. In this sense, if we can support our claim with smaller clusters (or a large number of clusters), our arguments might be better supported.

However, the reviewer brought up interesting questions that have made us reflect on our clustering procedure: (1) if we used a different method for tree-cutting, how variable the number of clusters would be? And (2) can our argument hold against smaller clusters (or a large number of clusters)? To answer the first question, we tested a total of four different tree-cutting methods/criteria: elbow method, gap statistics, maximum average silhouette coefficient, and the dynamic hybrid cut tree method (Langfelder et al., 2008) which has been successfully utilized for morphometric classification of mouse V1 neurons by Gouwens et al. (2019). We noticed that using the maximum average silhouette coefficient resulted in the most accurate cluster number in AL (N=54), but it showed a significant discrepancy in cluster number in MB calyx and LH when compared to the other three methods (see Table 1). After an internal discussion, we have decided to switch our tree-cutting method to the dynamic hybrid cut tree method for the following reasons: (i) choosing a tree-cutting method that returns a cluster number close to the number of different odor types (which is 10) best serves our goal, (ii) since we are interested in the clusters of uPN innervations to MB calyx and LH (as the correct labels in AL are already given), it seemed reasonable to choose a method out of the three methods that produced comparable cluster numbers in MB calyx and LH, (iii) the dynamic hybrid cut tree method was proven to give satisfactory results and is methodologically systematic based on our previous experiences and previous literature, and (iv) we wanted to test our hypothesis under a harsher condition to answer question 2.

Switching the tree-cutting method resulted in 10 clusters in MB calyx and 11 clusters in LH. We report that our argument still holds regardless of the higher cluster number. The trend of the uPNs under the same homotype falling into the same cluster became much more apparent (Figures 3, 4, and 8). Our various statistical tests also returned results in line with our findings.

As for the suggestion of a better explanation of Pearson’s Chi-squared test, we report that our manuscript is now significantly more accessible to the readers when it comes to communicating new concepts and introducing statistical quantities. Most of the equations and detailed statistics are moved to either the Methods or the appendices, replaced by simple and intuitive descriptions. Whenever we thought our explanation might be too difficult to follow, we added illustrations to supplement the text.

The authors do not consider their MB clusters in-depth or explain what one would expect from the field. The literature suggests that PN-KC connections in the field are either random or semi-random (Eichler et al. 2017; Zheng et al. 2020; Caron et al. 2013). The authors could discuss their results in this context, in particular, their conclusion that 'the distinction between the homotypes is only weakly present in MB calyx'. The structure of the PN-MB clusters might be compared to results from Zheng et al. 2020, who show that the calyx contains a 'fovea' of Kenyon cells oversampling particular PNs, specifically PNs that are from 'food-related' glomeruli. This PN group is probably captured somewhere in the authors' analysis. Interestingly, the authors show that pheromonal largely PNs cluster together in the MB, though they do not draw as much attention to this as the LH pheromonal grouping for some reason – could this be a pheromonal 'fovea'?.

We thank the reviewer for the suggestion. We are familiar with the literature brought up by the reviewer but actively decided not to pursue the subject in the draft due to the scope of our study. Originally, we did not analyze connectivity between uPNs and KCs, and without this information, we had to be very cautious when making any claims that may require PN-KC information to validate.

That said, we followed the reviewer’s suggestion and performed additional analyses to test the labeledline strategy using the PN-KC connectivity information. We now feel comfortable discussing the spatial properties of uPN innervation in MB calyx and the connectivity between uPNs and KCs. In our connectivity-based clustering, we observed a comparable ‘fovea’ of ‘food-related’ glomeruli seen by Zheng et al. (2020), based on the similarity in the connectivity pattern (see Figure 11). However, this ‘fovea’ does not seem to be completely driven by spatial proximity. We did observe a similarity between spatial proximity-based clustering and connectivity-based clustering for homotypes such as DM1, DM4, DP1m, DP1l, VA2, and VA4, all of which had a high deviation from the random bouton model in Zheng et al. (2020). While many of these homotypes are generally spatially proximal (the vast majority of the uPNs are located in clusters C_6_^MB^ and C_7_^MB^), some homotypes under the food-related ‘fovea’ such as VA2 are sampled from spatially distinct clusters. Therefore, we believe the creation of ‘fovea,’ or the PN-KC connectivity in general, is only partly driven by the spatial properties of uPN in MB calyx, and it appears that other factors play a role. Interestingly, the fact that many pheromone-encoding homotypes are spatially proximal in MB calyx may suggest the existence of pheromone-encoding `fovea,' but most uPNs in these homotypes do not converge in connectivity-based clustering with an exception of VA1d. Therefore, we assume that a pheromonal ‘fovea’ does not exist in MB calyx. In fact, we suspect the spatial organization of pheromone-encoding homotypes in MB calyx, which is placed at the center of the neuropil, to facilitate the observed randomization of connections by increasing the accessibility of KCs to these homotypes. What we found is a potential hygro/thermo ‘fovea,’ where glomeruli such as VP1d and VP2 are both spatially and connectivity-wise clustered together (along with VL1, curiously).

In the LH, the authors focus on their findings of a pheromone-specific cluster within the LH. However, one result they also have is that the other clusters have a mixture of different odour class PNs, so the LH does not easily break down into siloed compartments like the AL. A few studies have tried to break down the LH into different regions that serve different sections of an ethological odour space, based on different analyses (Strutz et al. 2014; Bates et al. 2020; Frechter et al. 2019). If the authors could at least qualitatively compare their groups to the previous groupings, their results and contribution would make more sense in the context of the field.

We thank the reviewer for the valuable information. We have reviewed the references and updated the manuscript to reflect the literature. Bates et al. seem to focus on the axo-axonic PN connections which share certain similarities with our clustering result, presumably driven by the necessity of spatial proximity to form synapses. For example, community 12 by Bates et al. is largely composed of VP1l and DL5, which resembles the cluster C10LH. Community 6 contains a mixture of VA5, VC1, D, DA4l, DC2, DA3, and VA7m, which is reminiscent of the cluster C6LH. However, the results cannot be mapped one-toone, and we suspect many spatially proximate homotypes do not make axo-axonic connections.

The study by Strutz et al. is particularly interesting because their results are comparable to ours. The three domains (LH-PM, LH-AM, and LH-AL) suggested by Strutz et al. seem to be a different combination of our clustering result (LH-PM and LH-AM correspond to the dorsal-posterior region and LH-AL corresponds to a combination of ventral-anterior region and the biforked bundle). The results by Frechter et al. suggest the spatial compartments of uPN innervations in LH cannot be directly translated to chemical feature segregation. However, the spatial concentration of ester-encoding LHNs in the *Drosophila* brain identified by Frechter et al. is intriguing as many homotypes encoding signals originating from esters are integrated by a group of common LHNs (see Figure 11B).

On C: The authors are interested in looking at 'homotype' uPNs – neurons with dendrites in the same AL glomerulus. This is a novel and interesting approach. The intra-homotypic-PN-distance (a 'bundling' metric) and inter-homotypic-PN-distance scores (a 'packing' metric) they use are insufficiently explained in the text and are better understood looking at figure 4. Figure 4 is a good, concise summary. Because sister PNs are essentially copies of the same cell and occupy the same space – a phenomenon seen across the whole of the *Drosophila* nervous system – it is unsurprising that they arborise similarly in each of the three analysis regions – the authors show, however, that they are 'morphologically similar in different ways' between the AL, the MB and the LH, which is an interesting point. However, the text does not make it very clear to the reader what the authors have shown. On the intra-homotypic-PN-distance: sister neurons are similar in all three regions – though in the AL they seem to tile glomeruli rather than -- intermingle (Figure 4). This last point is not, I think, shown quantitatively, only schematically, – it would be best to formalise and plot somewhere. Considering the inter-homotypic-PN-distance: neurons interdigitate across homotypes in the MB, to a lesser degree in the LH, and barely in the AL, instead of tiling the whole AL.

As we understand, the reviewer is pointing out the structural (or morphological) similarity between uPNs under different homotypes within neuropil even though technically, the innervations are parts of the same neurons. We also noticed the difference in the morphology of arborization but decided not to pursue the subject in this manuscript, as our primary goal of this paper is to analyze the spatial and organizational characteristics of uPN innervation in each neuropil. We believe the suggested study may be a bit tangential to the scope of this paper.

We, however, have done a related analysis on the morphological features of PN innervations at each neuropil, and some of the results are currently in press. A preprint is available at

(https://www.biorxiv.org/content/10.1101/2022.04.07.487455v1), should the reviewer wish to see a further discussion on the morphological classification of PNs based on a quantitative measure we devised. We have supplemented our discussion with the reference to our preprint for the readers who might be interested in the subject. We would also like to emphasize that we are very much interested in this subject and are planning on a follow-up paper with the volumetric study of PN innervations included.

The authors explore how their morphology findings might correlate with the odour response properties of PNs. They use odour categories from Bates et al. 2020, though in many cases a PN type was given multiple categories (see the supplemental figures), and it is not clear how the authors resolved this conflict. For example, the V glomerulus can also be viewed as 'aversive/bad' as it encodes CO2. Often a neuron cannot be given a singular 'odor scene' label, though they might be clustered using odour response data (Badel et al. 2016).

As the reviewer has noted, labeling a homotype using a single odor category can be insufficient in describing the wide spectrum of odor molecules it might encode. This can be especially jarring for PNs receiving inputs from ORNs expressing receptors that are broadly tuned. We tried to follow the convention set by Bates et al. as much as possible although we did notice a few homotypes (e.g., VM7d) fall under multiple categories. We noticed that this issue is specifically prevalent in odor categories one might consider ‘food-related,’ somewhat unsurprisingly. Therefore, whenever we make comments on these odor categories, we tried to make qualitative assessments only over the general category of ‘foodrelated’ homotypes. Additionally, we decided to make this detail explicit to the readers by adding the following section to our manuscript (see page 18 in the revision): “The odor type and odor valence information were extracted from various literature (Hallem et al., 2004; Galizia and Sachse, 2010; Stensmyr et al., 2012; Mansourian and Stensmyr, 2015; Badel et al., 2016; Bates et al., 2020) and we closely followed the categorical convention established by Mansourian and Stensmyr (2015) and Bates et al. (2020). However, we note that the categorization of a uPN under a specific odor category may overshadow the complete spectrum of odorants a uPN might encode, especially if the uPN encodes ORs that are broadly tuned. Therefore, we focused on the well-separated pheromone/non-pheromone encoding types and valence information.”

On D: In the conclusion, the authors do not discuss their own results in relation to the field that much. They only mention (i) one non-novel finding and (ii) one observation. The (i) finding is that certain pheromonal PNs segregate together in the LH (Ruta et al., 2010; Kohl et al., 2013, Frechter et al., 2019; Chakraborty and Sachse, 2021) – although their finding of the same in the MB is perhaps more novel? The authors say that "Our study not only lends support to the existing studies pointing to the labeled-line strategy in the *Drosophila* olfactory system but also suggests that an even more sophisticated level of spatial organization that depends on the homotypes and odor should be present"; it is unclear to me what the second half of the sentence means. The (ii) observation, implicit also in other work (Grabe et al. 2016) – that some PN cell types exist as singletons, often aversive ones. Certainly, in insects, there is some circuit-level difference between cell types that exist in multiples or as singletons – which are often larger, older neurons – but the present paper has no analysis on this point. Perhaps the authors could show how these singletons are morphologically special – e.g. they mention that they are more 'dense' but do not quantify this anywhere in the paper.

When we stated that a more sophisticated level of spatial organization is present, we originally meant three things: (1) the spatial organization of uPNs differs greatly in each neuropil which can be quantitatively measured (Figure 5), (2) uPNs in a homotype are tightly bundled in higher olfactory centers (both MB calyx and LH) despite the lack of glomerular structures (Figure 8), and (3) the spatial organization that supersedes the pheromone vs non-pheromone exists and may depend on the odor types (Figures 3, 4, and 6). To address the ambiguity in the conclusion, we have overhauled and greatly enhanced our Discussion section to better discuss our findings in relation to previous literature.

As stated in the previous response, we have done a separate study on the structural aspect of the PN projections to each neuropil, in which the morphological characteristics of ‘singletons’ were touched upon. A more detailed study on this subject is planned.

One of the authors' key conclusions and a point in their abstract, is that a labeled line strategy is at work in the *Drosophila* olfactory system at the point that second-order projections ramify in the lateral horn (LH). They make expansive statements such as: "Overall, our findings suggest that the *Drosophila* olfactory system leverages the efficiency of the labeled-line design in sensory information processing". A 'labeled line' is generally taken to be a chain of neurons that transfers a message about a single feature, onto higher-order neurons. This message may be modified and transformed along the way, but is generally not directly integrated with information about other features. The very peripheral part of the system in the antennal lobe (AL), ORNs -> PNs, is generally established to be a set of labeled lines (Couto et al., 2005; Fishilevich and Vosshall, 2005; Vosshall et al., 2000). Although significant cross-talk does exist, e.g. through AL local neuron computation, at this stage the PNs can still act as labeled lines (Olsen et al. 2010; Seki et al. 2017), which could support a somewhat labeled synfire chain up to the 3rd order (Jeanne and Wilson 2015).The authors do not define the term 'labeled line' clearly, or whether the label in question should be glomerular identity, odour identity, or odour scene. They also do not clearly state (a) what neurons they mean to say are in the labeled line, nor (b) whether they think this is largely the case, or only in specific channels – their sweeping statements suggest they mean the former, but their data only weakly support the later. I assume, for (a) they are suggesting that some lateral horn neurons (LHNs) will be part of labeled lines, but not Kenyon cells of the mushroom body, because of the PN morphological properties that the authors quantify in the LH and MB. For (b), established work in the field suggests a murky division within the LH to support odour categorisation (Frechter et al. 2018; Jeanne et al. 2018) – a convergence scheme, and so, not a labeled line system – and the authors' own results show PNs of different, broad odor classes intermingling in 3/4 of their LH clusters. Their analysis, for (b), seems to show that only a small subset of PNs have the appropriate morphology to support labeled line connections in the LH. The main PNs that may form a labeled line, are the pheromonal PNs in the anterior-most region of the LH. This has been noted by the field, in terms of both morphology and connectivity (Ruta et al., 2010; Kohl et al., 2013, Frechter et al., 2019; Chakraborty and Sachse, 2021; Bates et al. 2020; Bates et al. 2020; Jefferis et al. 2007).However, they do not demonstrate that they act as labeled lines. To make a statement about labeled lines is to make a statement about connectivity. It can be guessed by using morphology when connection data is absent. However, the connection analysis can now be done using data from the hemibrain connectome (Scheffer et al. 2020; Schlegel et al. 2021). Labeled line strategies may exist for some odour channels in *Drosophila* – in particular pheromonal channels such as DA1 (Kohl et al. 2013) or the aversive channels the authors note – but it is not possible to determine this using morphology alone. A preprint that examines DA2 – an aversive PN thought to be part of a labeled line (Stensmyr et al., 2012) – actually saw that in the connectome its targets experienced a lot of convergence from different PNs, only a few preserving a possible labeled line (Huoviala et al. 2020). Labeled lines are probably the exception, not the rule, and with DA2 a strong labeled line organisation seems to become a highly distributed representation at the lateral horn stage. If the authors could compare their work on morphology, to the reality in terms of connectivity, they would be better able to support their ideas and show that the features they quantify correlate with circuit structure.

We thank the reviewer for the valuable opinion. First, we believe it is necessary to clarify what we meant by ‘labeled-line.’ It was not our intention to suggest the PN-KC or the PN-LHN connections follow the ‘labeled-line’ design in general. We used the term ‘labeled-line’ predominately to denote the homotypic bundling that largely extends to MB calyx and LH, instead of ending at the ORN-PN interface (AL). In AL, obvious glomerular structures and ORN convergence to specific glomerulus is already well-known. Our intra-, inter-PN distances, and the clustering results suggest the bundling of PN homotypes is generally well-preserved throughout the neuropils and spatially localized. We were suggesting the ‘labeled-line’ in terms of the glomerular labels (or homotypes) based on this result. Regardless of what the connections to KCs and LHNs are like, PN organization alone could be deemed to express the ‘labeled-line’ principle up until the synaptic interface.

However, the suggestion from the reviewer piqued our interest in this subject and we decided to use the connectivity data to conduct an additional small-scale study to check how far the labeled-line principle holds. For a comprehensive connectivity dataset between PNs and higher olfactory neurons such as KCs and LHNs, we utilized the hemibrain dataset. We queried KCs and LHNs with synaptic connections greater than or equal to 3 for the 120 PNs we tested for the reproducibility study, which resulted in 1754 KCs and 1295 LHNs. Here, we tried to address the following questions: (1) Are there any KCs and LHNs that carry a specific type of information? If so, how prevalent are they? (2) Are there any insights we can gain from comparing the clustering outputs using spatial proximity (d*_αβ_*) and connectivity?

From the connectivity data, we observed that most of the KCs and LHNs integrate information from multiple homotypes but there are also a small number of KCs and LHNs that synapse only with a single homotype (Figure 10). These ‘homotype-specific’ connections (Nx,spξ), defined as the number of thirdorder neurons that only synapses with a specific homotype but not with the others (see Figure 9 and Methods for more information), are much more prevalent in LHNs compared to KCs. Certain homotypes (e.g., DA1) have an especially high number of LHNs that only connect to the given homotype. We also noticed that hygro/thermo-sensing homotypes have a generally higher percentage of ‘homotype-specific’ LHNs. The ‘homotype-specific’ neurons functionally carry a single type of information, thereby may be considered as an extension of the labeled-line strategy.

When we collected LHNs connected to a particular homotype and checked which other homotypes these LHNs are also synapsing (thereby analyzing the scope of signal integration happening at LH – see Figure 9 and Methods for more information), we found a strong tendency of signals from pheromone and hygro/thermo-sensing uPNs to be integrated within the given odor/signal type (Figure 11). For example, signals encoded by pheromonal homotypes share many LHNs, integrating pheromone-related signals and forming odor type-specific ‘channels’ (see purple arrows in Figure 11B). Many food-related homotypes, such as DP1l and DL2v, also share common LHN channels. It seems like the homotype-specific labeled line predominate until the signal reaches the higher olfactory centers, some of which then transition into odor-specific channels where either a broad or a narrow integration occurs. This characteristic aligns with the spatial segregation we observed in our spatial proximity-based clustering study. In MB calyx, no such trend is observed, further supporting the previous literature on randomized connections. Even though we observed a strong per homotype bundling tendency at MB calyx, the high degree of overlapping (denoted by λ) seems to have a bigger impact on connectivity.

To address the second question, we performed a connectivity-based hierarchical clustering and compared the result against our spatial proximity-based clustering result. For the ‘distance’ between the connectivity patterns of two uPNs to third-order olfactory neurons, we used the cosine distance between two vectors, which has been previously used in the field to analyze the connectivity matrices (Bates et al., 2019, Bates et al., 2020, Eschbach et al., 2020). For the hierarchical clustering, we used Ward’s criterion, which minimizes the variance of merged clusters. Tanglegrams are plotted to compare the dendrograms generated from the spatial proximity (d*_αβ_*) and from the connectivity (d*_cos_*) (see Figure 12).

One thing we suspected of certain odor types (or scenes) is that the label extends further to encompass multiple homotypes with common functionality, based on the previous literature commenting on the segregation of pheromone and non-pheromone encoding PNs (Jefferis et al., 2007; Seki et al., 2017; Chakraborty and Sachse, 2021). Our connectivity analysis showed that the connection between PNs and LHNs is highly systematic (see Figure 12B). We quantitatively studied the output by (1) analyzing the relationship between neurons based on connectivity-based hierarchical clustering and (2) comparing the relationship between two hierarchical clustering outputs. We ran statistical tests on the glomerular labels, odor types, and odor valence against the connectivity-based clustering output. First, we found a statistically significant association between the glomerular labels/odor types/odor valence and the PNLHN connectivity (p ≪ 0.001). This is visually observable through the tanglegram, where PNs under the same homotypes are within small cosine distances and thus are neighboring in the dendrogram (Figure 12B).

PN-KC connectivity (see Figure 12A), on the other hand, has a much weaker association overall, with many homotypes that used to be spatially grouped into a cluster now distributed across multiple clusters. This is also in line with previous literature. We do see, however, a small subset of homotypes that are spatially clustered and carry similar connectivity patterns, such as D, V, and VM7d.

The two tree structures (one for spatial proximity and another for connectivity) are compared using several different metrics. First, we calculated Baker’s γ index, which is a measure of the rank correlation between two lists (corresponding to the leaves in a dendrogram) spanning from -1 to 1 (where 0 indicates the ordering of two trees are completely dissimilar and 1 or -1 indicate the ordering of two trees match). We found that G_Baker_^MB^ = 0.286 and G_Baker_^LH^ = 0.219. Baker’s γ index in LH is in line with the previous literature by Bates et al. (2020), who conducted a similar study using the NBLAST scores and reported G_Baker_^LH^ = 0.21. What is surprising is G_Baker_^MB^, which had a higher value compared to that of LH. We assume that this is due to Baker’s γ index only considering the ordinal relation between leaves. In our opinion, to provide a comprehensive description of the tanglegram, the ordinal relation (quantified by Baker’s γ index) should be supplemented by additional metrics, such as transferability (quantified by entanglement) and the correlation between two tree structures (quantified by cophenetic distance correlation).

We, therefore, supplied two additional measures that provide a complete picture of the tanglegram. The first is the entanglement, a measure ranging from 0 to 1, which quantifies the number of lines crossing in a tanglegram. We report the entanglement at MB calyx to be 0.35 and at LH to be 0.26 (a lower entanglement score suggests a possibility of block-to-block mapping without disturbing the tree structure). Next, we computed the cophenetic distance correlation between two trees. The cophenetic distance between two leaves measures the minimum height of a tree that contains both leaves. We calculated the Pearson’s correlation coefficient using the pairwise cophenetic distance between all leaves.

We report r = −0.032 (p > 0.001) for MB calyx and r = 0.236 (p << 0.001) for LH.

The authors say that "our analysis for the second-order neurons inside the *Drosophila* olfactory system can be translated to the brain of different organisms including the central nervous system (CNS) of humans". However, I do not think this is strictly true. In *Drosophila*, neurons of the same cell type, the author's 'homotypes', are near isomorphic and occupy a similar space. Therefore, the authors can use, as they did, a metric defined in Euclidean space to compare the neurons, without spatially transforming the cells at all. In mammals, duplicate types can appear across, say, cortical columns and even in the olfactory system be very spatially segregated, since there is not just one glomerulus for each olfactory receptor, as in insects. Therefore, the spatial analysis that would need to be done in mammals would be different.

While we understand that duplicate but spatially segregated homotypes exist in mammals, we still believe a distance metric in Euclidean space can still work. Instead of per homotype categorization, the calculation can be done over each glomerulus. Different glomeruli under the same homotype can be compared, or if one wishes to consider features per homotype, glomeruli under the same types can be averaged. The detailed methodology must differ, but we believe a similar distance metric can be applied to the mammalian olfactory system in principle.

It is commendable that the authors have made their analysis in python available on GitHub (https://github.com/kirichoi/DrosophilaOlfaction), as well as preprinting their work (https://www.biorxiv.org/content/10.1101/2022.02.23.481655v1). The authors pre-process the skeleton data well, removing registration artefacts. The consensus lateral horn volume (Ito et al. 2014) is somewhat arbitrarily defined as the uPN terminus region, though the MB Calyx is more strongly defined by glial sheathing. The authors wisely do not use this volume but re-define their neuropils – perhaps more accurately regions of interest – by using density estimates built from the neurons' 3D points. It would have been nice to see, in a supplementary figure, how these ROIs differ from the Ito et al. 2014 standard neuropils – since when they use neuropil terms in the text, an informed reader will assume they used the Ito standards. I like the visual segmentation process description in Figure S5.

Unfortunately, we were unable to recover the surface point coordinates from the paper by Ito et al. (2014) as suggested. Instead, we present the comparison between our segmentation outputs to that provided in CATMAID. Our segmentations are largely within the boundary given by CATMAID. A slight difference is observed for MB calyx at the medial and posterior region of the neuropil. However, we do not think this is a serious issue since we are segmenting neuropils from a bundle composed solely of uPNs, where to goal is to simply remove the rigid PN backbone from the synaptically dense neuropils. Therefore, the accuracy of our boundaries is not as influential compared to segmenting neuropils out of a dense mixture of different types of neurons. The minor difference in the density we observed might be due to the lack of uPNs that does not innervate all three neuropils and mPNs (which were dropped when generating the boundaries) when we segmented the neuropils.

**Author response image 1. sa2fig1:** Comparison of volumes defined in CATMAID (blue) and boundaries defined by our segmentation method (black) for MB calyx (top) and LH (bottom).

I think the paper does make many quantitative points on PN morphology well, and could either become a shorter manuscript that shows this information concisely, or a more involved one on 'labelled lines' and the validity of this idea for the olfactory system, where 2nd-order neurons meet 3rd order ones. In the first case, adding thermo sensory PNs, GABAergic uPNs, and mPNs to their analysis could make it more interesting, and produce some new and germane insights for their sub-regional analyses (see below). In the second case, the paper would need to include work that compares across different PN classes and EM datasets, including looking at connectivity in the hemibrain, to make its core claims on labeled lines.In general, in the present work, the authors do not provide enough biological background information on the olfactory system and what is known about PNs already, the regions they innervate for a naive reader to understand their results, and why their results might be interesting. This could be easily fixed with a little re-writing. The use of mathematical notation within the main text is heavier than needs be and hampers the readers' understanding of what the authors are doing and intend to show. Plainer writing would help. The authors need to communicate their core findings, and their context in the field, more clearly.

We thank the reviewer for the suggestion. In the end, we decided to implement parts of both cases that were suggested by the reviewer. First, we have re-done our calculation using the latest FAFB dataset, including hygro/thermos-sensing uPNs. We have performed a comprehensive study on every uPNs that innervate all three neuropils and a separate analysis including uPNs that does not innervate all three neuropils. Many GABAergic uPNs that were originally left out are now incorporated into the manuscript. Additionally, we performed an extensive study on the labeled-line hypothesis using connectivity data with third-order olfactory neurons. Our connectivity-based studies examined the ‘homotype-specific’ KCs/LHNs, signal integration across different homotypes done by KCs/LHNs, and the comparison between spatial proximity-based and connectivity-based clustering.

We have updated our manuscript, along with its figures and captions, so that readers have less trouble digesting the content. We have considerably updated the manuscript by adding various biological backgrounds and previous literature the reviewers have mentioned.

Glomeruli identification through the literature can be tricky. There has been some confusion specifically about the identities of VM6 and VC5. The Zheng et al. 2018 paper had a few errors, later rectified by Bates and Schlegel et al. 2020 and then again by Schlegel and Bates et al. 2021. Given the confusion, no one can be faulted for mistakes here, but authors should make sure their labels are the same as in Schlegel and Bates et al. 2021 (that process was a multi-lab debate). I think they used the outdated Zheng et al. 2018 labels.

We thank the reviewer for the valuable information. Since we have re-done our calculation based on the latest FAFB dataset (Bates et al., 2020), we ended up predominantly using the labels by Bates et al. We have further updated our labels according to Schlegel et al. (2021), regarding glomeruli VC3m, VC3l, and VC5.

SuggestionsI have some suggestions for increasing interest in this work, that I humbly submit to the authors. The authors discuss 51 glomeruli of the antennal lobe. Of the olfactory glomeruli, there are 52, but there are actually 58 glomeruli in total in the antennal lobe (e.g. VP1-5), including the thermo-hygrosensory ones. Other work (Schlegel et al. 2021; Bates et al. 2020) has found that some of the most striking differences can be found between olfactory and thermo-sensory glomeruli, as well as some curious associations across the two modalities. Since intellectually this system is very similar – RNs contact PNs that then reach the LH and the zone just ventral to it – but not quite parallel – arbours intermingle with olfactory ones in the ventral LH, and antennal lobe local neurons cross compute between combinations across all 58 glomeruli – I strongly think including these neurons in the author's analysis would increase the interest in this work.There are a total of 347 PNs in the FAFB dataset that project from the right-side antennal lobe. This is because there are 283 multi-glomerular PNs (mPNs). Bates et al. 2020 also make 58 uPNs from the left side of the brain available. All of the data is open and can be gotten by the authors here: https://fafb.catmaid.virtualflybrain.org/. Similar to the above, I think that including the mPNs as a comparison point, would increase interest in this work.

We thank the reviewer for the suggestion. As noted in previous responses, we have re-done our calculation using the latest FAFB dataset, from which we found 57 glomeruli (we couldn’t find the.swc file associated with neuron ID = 1356477 forming VP3 in the dataset and other uPNs in VP3 does not innervate all three neuropils) including those involved in hygro/thermos-sensing, and the hemibrain dataset, from which we recovered all 58 glomeruli. As pointed out by the reviewer, we observed hygro/thermos-sensing uPNs to innervate significantly different regions of MB calyx and LH. uPNs forming these glomeruli are spatially segregated from the odor/pheromone encoding uPNs. In MB calyx, we found these uPNs to rarely project ventrally, generally clustered to the base of the neuropil. In LH, we found these neurons to be clustered in the dorsal-ventral region of the neuropil, hardly innervating the neuropil but covering the medial side of LH. The manuscript has been updated to include this information. Additionally, as described in previous responses, we have updated the manuscript with the additional analyses of many GABAergic uPNs that were originally left out of our study.

PNs could also have been clustered by their odour response profiles using data from PN response (Badel et al. 2016) or ORN response (Münch and Galizia 2016), rather than porting labels from Bates et al. 2020 / Mansuorian et al. 2015. The authors might consider whether this could add meaningfully to their analysis.

We thank the reviewer for an interesting suggestion. The odor response/dose-response profiles are critical in functional and information-theoretic analyses of the olfactory system. A related study would be a valuable addition to our odor type-dependent analyses by providing a continuous physiological variable to study against the spatial characteristics of uPN innervations. Unfortunately, we have a limited time for the revision of this manuscript, and a satisfactory study involving the odor response profile may not be possible within this time frame.

In addition, as noted in the public review, I think comparing against or using NBLAST could be informative. The authors could run NBLAST on their processed neuron skeletons and discover whether the results differ much from what they have in hand right now. NBLAST is now implemented in the navis python library: https://navis.readthedocs.io/en/latest/source/tutorials/nblast.html.

We have calculated the normalized NBLAST distance between uPN innervations in each neuropil and compared it against our d*_αβ_* (see Figure 2—figure supplement 1B). While the two distances are highly correlated, there are cases with the same NBLAST distance with different d*_αβ_*, as d*_αβ_* only measures the spatial proximity but not the morphological similarity between two neurons. As we stated before, the primary goal of this paper is to analyze the spatial and organizational characteristics of uPN innervation in each neuropil, and we believe the distance metric d*_αβ_* is conceptually more physical and adequate for the primary aim of our study.

Lastly, a more out-there suggestion: PN morphology analysis using a skeleton representation has been common in the field. However, volumetric analyses based on neuronal meshes – now available for these neurons through the hemibrain and flywire projects – is almost non-existent. How do the volumes for neurons vary across types, clusters, etc? Using this information could add some simple points, to support the authors in their quantitative description of these neurons.

As the reviewer pointed out, neuronal volumetric analyses of the brain are relatively rare, largely due to the lack of data to perform said analyses until now. We agree this is indeed a very interesting topic, but due to the limited time we have for this manuscript revision, we are afraid that a comprehensive study of the volumetric characteristics may not be possible at this time. However, we are quite enthusiastic about this subject, and we are eager to perform a follow-up study in the near future to answer this question.

Reviewer #3 (Recommendations for the authors):This work provides a quantitative evaluation of the spatial organization pattern of olfactory projection neurons (PN) in AL, CA, and LH based on the inter-PN distances using FAFB EM dataset. This NBLAST-based method does provide a simple way to cluster neurons with similar projection patterns and even predict underlying signal processing rules. However, the reliability of the clustering method needs to be improved since the method only got 39 clusters out of 51 well-segregated AL glomeruli serving as the ground truth. This result undermines the conclusions in the higher-brain centers which have a more complex organization. Moreover, the authors defined the neuropils by rotating the neurons along specific axes and then segmenting the dense innervation parts, which may hinder the accuracy of the boundary when the surface is convoluted and does not reveal inner subdomains. Nevertheless, serving as the first step to tackling complex connectomic data, the method is easy and potentially useful.1. The current study analyzed only 111 uniglomerular PNs instead of the latest-released 164 uniglomerular PNs (Bates et al., 2020). To make the work more valuable, the authors should apply their analysis to the latest dataset. In addition, since the analysis was done from the EM data of a single fly, whether the preferential spatial distribution of PNs and the clustering are consistent in different individuals is unknown. It will be informative and more persuasive if a similar spatial distribution pattern can be observed in another fly EM dataset (Scheffer et al., 2020).

Following the reviewer’s comment, we have re-done all our calculations using the PNs that meet our existing criterion in the latest FAFB dataset (Bates et al., 2020). We ended up analyzing 135 PNs that innervate all three neuropils in the latest FAFB dataset, including those involved in hygro and thermo sensation. Also, we have performed a separate analysis on the rest of the uPNs that did not innervate all three neuropils (28 uPNs), most of which are GABAergic. We have updated the manuscript and the figures, pointing out several newly found characteristics. Additionally, we have applied our calculation to the hemibrain dataset (Scheffer et al., 2020) to make sure that our results are reproducible and generalizable in different individuals. We generated several figures in the main text using the hemibrain dataset and compared them against the figures generated from the FAFB dataset. The results from both datasets are in support of our arguments (Figure 13).

2. The segmentation of neuronal innervation in AL, Calyx, LH is achieved by rotating the neurons along specific axes and then identifying the dense innervation parts as the three neuropils (Figure S5). The methodology is convenient to define the neuropils but will hinder the accuracy of the boundary segmentation when the surface is convoluted (i.e. MB calyx) and does not reveal inner subdomains (i.e. AL glomeruli). As the original EM dataset has already offered neuropil surface point coordinates which can be downloaded from Catmaid website (https://catmaid-fafb.virtualflybrain.org/), attributing the innervation points directly to the defined neuropils should be more accurate.

We appreciate the reviewer’s concern since systematically segmenting different parts of a brain is indeed a surprisingly challenging problem. However, we do not believe this issue will cause a serious problem for our study. Our segmentation process was done only on uPN bundles to separate neuropils from rigid PN backbones. This means that an accurate boundary is preferable but not strictly necessary since we do not have to worry about incorporating non-PNs into our study. We are also not too concerned about identifying the inner subdomains since our analyses do not necessitate to re-segment the AL but instead use the pre-determined labels offered by the original authors of the dataset. We have compared our boundaries with the volume surface provided in CATMAID and found that our methodology seems to be good enough for the purpose of detaching synaptically dense regions from the PN backbones.

3. Theoretically, the natural segregation of glomerular structures in AL would serve as the ground truth to test the reliability of the clustering method. Yet, the method only got 39 clusters out of 51 well-segregated glomeruli. This discrepancy undermines the effectiveness of the clustering strategy. The result of silhouette coefficient analysis also suggests that the coefficients are similar for clusters ranging from 30 to 50 in AL (the coefficients are very close almost reaching a plateau) (Figure S6). The author should justify why choosing the number of PN innervation clusters in AL as 39 and provide evidence that it is optimized to find the hidden pattern. Most importantly, if the method fails to reveal the remaining 11 AL glomeruli that can be visually distinguished, it is difficult to see how it can reveal the hidden pattern in MB calyx which is much less well defined.

As the reviewer pointed out, we have fewer clusters for AL than the known number of glomeruli available in the dataset. However, it should be noted that this is primarily an issue of a particular tree-cutting method we used, not the hierarchical clustering per se. When looking at the dendrogram for AL, one might notice that the well-segregated glomeruli are indeed expressed through uPNs forming the same glomerulus grouped under a common branch. We could arbitrarily choose a cut height that correctly recovers all known glomeruli available in the latest FAFB dataset, suggesting that our distance metric and the clustering algorithm are appropriate to show the hidden pattern in the spatial organization. If the tree structure correctly produces the expected grouping of uPNs in AL based on the glomerular labels, we believe the actual number of clusters formed through a specific tree-cutting method is less relevant – a smaller number of clusters is simply a manifestation of grouping several glomeruli together. In our application, clustering on AL is not necessary since we already know the correct labels for each uPN. On the other hand, we do not know the correct labeling for uPNs at MB calyx and LH, so we had to resort to some systematic approach to figure out the appropriate number of clusters. For this purpose, we decided to use a method that returns a cluster number close to the number of different odor types (which is 10), as we believe that would best serve our primary goal of this paper.

4. The neuronal distance estimation algorithm may encounter a problem when projection neurons have very different innervation ranges and the total length of the innervation branches. For example, if there is a projection neuron, PNa, only innervates in the entry of Calyx where most PNs pass by, the distance between PNa and other PNs will be very small due to the algorithm only selects the shorter skeleton to evaluate the distance (see method and the symmetric matrices in Figure 3). Thus, the result may not faithfully reflect the fiber distance when two PNs have a drastic difference.

We thank the reviewer for pointing out a valid concern. We believe the normalization term (the division by the number of coordinates in the neuron N_*α*_) should alleviate this issue. This term will make sure that two long branches that are spatially well-bundled have a smaller value of d*_αβ_* than a short branch and a long branch. It is possible to have two long branches that are far apart having a similar d*_αβ_* to that of a short branch and a long branch, but we believe both cases should be treated as poorly bundled.

5. The authors estimate λ value which represents the ratio of the mean of intra-homotypic neuron distance and the mean of inter-homotypic neuron distance. (Figure 4, Figure 5, Figure 6, Figure S3). It is a creative way that gives us insight into how PNs overlap with each other in distinct neuropils. However, the volume and the aspect ratio of the three neuropils are different. AL occupies a much larger area compared to calyx and LH. Therefore, the current λ value may simply reflect the spatial distance presented in the neuropil but not the actual degree of overlapping (e.g. in two non-overlapping pairs, the one with long distance will have a low λ value while both pairs have no overlapping at all). To make the λ value reflect more to the degree of overlapping, they might need to normalize it based on the volume, compare the λ values between original data with the data after shuffling PNs into different clusters or take the innervation density of fibers in specific neuropils into account.

We thank the reviewer for their valuable opinion. First, we would like to iterate that we devised the λ to be a scale-free ratio that quantifies the relative spatial innervation per homotype. This means that we intended to not scale our λ based on the volume. Let us consider a homotype composed of several uPNs and has similar d*_intra_* values for both MB calyx and AL. Indeed, the volume of MB calyx is smaller than that of AL, so the d_inter_ will be generally larger in MB calyx, leading to a higher λ value. This is what we intended and our logic behind this is that in AL, despite all the additional space that the uPNs technically could have innervated, the uPNs ended up localized into a bundle with a small d*_intra_*. In MB calyx, the same uPNs had less space to innervate, to begin with, so the less ‘packing’ observed in MB calyx may not be as novel. We wanted our λ to differentiate the two, which is the reason why our λ is defined as such.

We believe both definitions provide valuable information on spatial organization and therefore important. However, it seems like the term ‘overlapping’ is a bit elusive and can be defined in various ways, which we believe is why the reviewer might have been confused by our definition. We feel that our explanation and choice of words (overlapping) were not sufficiently clear. After a lengthy discussion, we have decided to keep the terminology but provide a much more comprehensive explanation of what we mean by the ‘degree of overlapping’ to make sure that the readers won’t get confused.

6. The authors annotate the reaction profile for a specific glomerulus using a dumb variable to do statistics. The result suggests that pheromone PNs will be clustered together in LH (into 2 clusters) and Calyx. Food-related PNs and aversive PNs also be clustered into different groups in LH but not in Calyx (Figure 7, Table S1). The author should elaborate and discuss more in terms of biology. For example, do food-related odor signals and aversive odor signals converge in MB calyx and diverge in LH? What biological properties may be implicated in the organization?

We believe there is a much more subtle uPN organization going on at each neuropil. First, our inter-PN distances and clustering results suggest the bundling of uPN homotypes is generally well-preserved throughout the neuropils despite the visual lack of glomerulus at MB calyx and LH. However, in MB calyx, the spatial segregation between different homotypes is practically non-existent, leading to a high degree of overlapping. In LH, the spatial segregation between homotypes becomes much more pronounced, leading to a reduction in the degree of overlapping. Therefore, in terms of connectivity, the surface presented by MB calyx is much more diverse, perhaps to assist the randomized sampling known to exist at the PN-KC interface. In LH, we find four stereotyped conformations (both spatially and morphologically) overall: (1) dorsal posterior region largely occupied by food-encoding uPNs, (2) ventralanterior region largely occupied by pheromone-encoding uPNs, (3) biforked bundle surrounding dorsalposterior region largely occupied by food-encoding uPNs with an aversive response, and (4) dorsalanterior-medial region largely occupied by hygro/thermo-sensing uPNs. The spatial localization seems to be related to the sensory signal encoded by the uPNs, as suggested in previous literature.

To gain insight into the biological implications of the spatial properties we observed, analysis based on the connectivity to high-order olfactory neurons is necessary. Our connectivity-based analyses observed a substantial number of LHNs synapsing only with particular homotypic uPNs, which seems to be correlated with the spatial segregation of uPN homotypes in LH. Additionally, LHNs tend to be much more specific about signal integration at LH compared to KCs. That is, LHNs tend to be selective about which homotypic uPNs to synapse with and the connectivity is correlated with the odor types. The thirdorder olfactory neurons that take inputs only from a specific homotype are indeed exceptions in the PNKC or the PN-LHN interface, as most of the KCs and LHNs tend to integrate signals from diverse homotypic PNs. But these special cases may be considered as an extension of the ‘labeled-line’ design seen in the second-order neurons (PNs). These ‘labeled-line’ designs have significant biological implications. The olfactory information encoded by neurons part of the ‘labeled-line’ design may be processed under a different principle from other odors. We have updated the manuscript with the above information.

7. Although the authors have offered the python scripts for researchers to reproduce their results, a comprehensive spreadsheet that contains all the distance results along with functional annotation will greatly improve the accessibility of the analyzed results.

We have shared the pre-computed pairwise distance matrices for PNs in every neuropil with the respective neuron IDs annotated for both the FAFB and the hemibrain dataset in.csv files. Furthermore, we have created spreadsheets containing some of the functional information (e.g., glomerulus labels, clustering labels, etc.), all of which are available in our repository.

8. The authors should compare their cluster results with Lin et al. 2007 and Bates et al. 2020 and discuss the biological implication.

Following the reviewer’s suggestion, we have added a paragraph in the Discussion section comparing our cluster results with that of Lin et al. (2007) and Bates et al. (2020) Our results are generally consistent with the previous studies that have clustered olfactory neurons using various forms of data. For example, in our study, homotypes DL2v and DL2d constitute a bilateral cluster in MB calyx (C_3_^MB^), and the dual organization of uPNs is present in MB calyx and LH, such that homotypes DC2, DL1, and VA5 are sorted into the same cluster in LH while sharing similar innervation pattern in MB calyx, all of which are in line with Lin et al. (2007)

The uPNs under the same cluster or nearby clusters in our study are clustered together in the NBLAST score-based clustering analysis by Bates et al. (2020). The uPNs that ended up in the same cluster or nearby clusters, such as homotypes DM1, DM3, DM4, VA4, and VM3 in the cluster C_3_^LH^, are also grouped in the NBLAST score-based clustering analysis by Bates et al. (2020). This suggests that a level of stereotypy of uPN organization in MB calyx and LH that is universal, which can be captured through different metrics and methodologies.

References:

1. Badel, L., Ohta, K., Tsuchimoto, Y., and Kazama, H. (2016). Decoding of context-dependent olfactory behavior in *Drosophila*. Neuron, 91(1), 155-167.

2. Bates, A. S., Janssens, J., Jefferis, G. S., and Aerts, S. (2019). Neuronal cell types in the fly: singlecell anatomy meets single-cell genomics. Current opinion in neurobiology, 56, 125-134.

3. Bates, A. S., Schlegel, P., Roberts, R. J., Drummond, N., Tamimi, I. F., Turnbull, R., ... and Jefferis, G. S. (2020). Complete connectomic reconstruction of olfactory projection neurons in the fly brain. Current Biology, 30(16), 3183-3199.

4. Bopp, R., Maçarico da Costa, N., Kampa, B. M., Martin, K. A., and Roth, M. M. (2014). Pyramidal cells make specific connections onto smooth (GABAergic) neurons in mouse visual cortex. PLoS biology, 12(8), e1001932.

5. Das Chakraborty, S., and Sachse, S. (2021). Olfactory processing in the lateral horn of *Drosophila*. Cell and Tissue Research, 383(1), 113-123.

6. Choi, K., Kim, W. K., and Hyeon, C. (2022). Polymer physics-based classification of neurons. bioRxiv.

7. Douglas, R. J., Martin, K. A., and Whitteridge, D. (1989). A canonical microcircuit for neocortex. Neural computation, 1(4), 480-488.

8. Eschbach, C., Fushiki, A., Winding, M., Schneider-Mizell, C. M., Shao, M., Arruda, R., ... and Zlatic, M. (2020). Recurrent architecture for adaptive regulation of learning in the insect brain. Nature Neuroscience, 23(4), 544-555.

9. Galizia, C. G., and Sachse, S. (2010). Odor coding in insects. The neurobiology of olfaction, 35, 70.

10. Gouwens, N. W., Sorensen, S. A., Berg, J., Lee, C., Jarsky, T., Ting, J., ... and Koch, C. (2019). Classification of electrophysiological and morphological neuron types in the mouse visual cortex. *Nature neuroscience*, 22(7), 1182-1195.

11. Hallem, E. A., Ho, M. G., and Carlson, J. R. (2004). The molecular basis of odor coding in the *Drosophila* antenna. Cell, 117(7), 965-979.

12. Ito, K., Shinomiya, K., Ito, M., Armstrong, J. D., Boyan, G., Hartenstein, V., ... and Insect Brain Name Working Group. (2014). A systematic nomenclature for the insect brain. Neuron, 81(4), 755-765.

13. Jefferis, G. S., Potter, C. J., Chan, A. M., Marin, E. C., Rohlfing, T., Maurer Jr, C. R., and Luo, L. (2007). Comprehensive maps of *Drosophila* higher olfactory centers: spatially segregated fruit and pheromone representation. Cell, 128(6), 1187-1203.

14. Langfelder, P., Zhang, B., and Horvath, S. (2008). Defining clusters from a hierarchical cluster tree: the Dynamic Tree Cut package for R. *Bioinformatics*, 24(5), 719-720.

15. Lin, H. H., Lai, J. S. Y., Chin, A. L., Chen, Y. C., and Chiang, A. S. (2007). A map of olfactory representation in the *Drosophila* mushroom body. Cell, 128(6), 1205-1217.

16. Mansourian, S., and Stensmyr, M. C. (2015). The chemical ecology of the fly. Current opinion in neurobiology, 34, 95-102.

17. Scheffer, L. K., Xu, C. S., Januszewski, M., Lu, Z., Takemura, S. Y., Hayworth, K. J., ... and Plaza, S. M. (2020). A connectome and analysis of the adult *Drosophila* central brain. Elife, 9.

18. Seki, Y., Dweck, H. K., Rybak, J., Wicher, D., Sachse, S., and Hansson, B. S. (2017). Olfactory coding from the periphery to higher brain centers in the *Drosophila* brain. BMC biology, 15(1), 1-20.

19. Stensmyr, M. C., Dweck, H. K., Farhan, A., Ibba, I., Strutz, A., Mukunda, L., ... and Hansson, B. S. (2012). A conserved dedicated olfactory circuit for detecting harmful microbes in *Drosophila*. Cell, 151(6), 1345-1357.

20. Zheng, Z., Li, F., Fisher, C., Ali, I. J., Sharifi, N., Calle-Schuler, S., ... and Bock, D. D. (2022). Structured sampling of olfactory input by the fly mushroom body. Current Biology.

21. Zheng, Z., Lauritzen, J. S., Perlman, E., Robinson, C. G., Nichols, M., Milkie, D., ... and Bock, D. D. (2018). A complete electron microscopy volume of the brain of adult *Drosophila melanogaster*. Cell, 174(3), 730-743.